# Near-Optimal Bounds for Learning Gaussian Halfspaces with Random Classification Noise

**Ilias Diakonikolas**
University of Wisconsin, Madison
ilias@cs.wisc.edu

**Jelena Diakonikolas**
University of Wisconsin, Madison
jelena@cs.wisc.edu

**Daniel M. Kane**
University of California, San Diego
dakane@ucsd.edu

**Puqian Wang**
University of Wisconsin, Madison
pwang333@wisc.edu

**Nikos Zarifis**
University of Wisconsin, Madison
zarifis@wisc.edu

## Abstract

We study the problem of learning general (i.e., not necessarily homogeneous) halfspaces with Random Classification Noise under the Gaussian distribution. We establish nearly-matching algorithmic and Statistical Query (SQ) lower bound results revealing a surprising information-computation gap for this basic problem. Specifically, the sample complexity of this learning problem is $\widetilde{\Theta}(d/\epsilon)$, where $d$ is the dimension and $\epsilon$ is the excess error. Our positive result is a computationally efficient learning algorithm with sample complexity $\tilde{O}(d/\epsilon + d/(\max\{p, \epsilon\})^2)$, where $p$ quantifies the bias of the target halfspace. On the lower bound side, we show that any efficient SQ algorithm (or low-degree test) for the problem requires sample complexity at least $\Omega(d^{1/2}/(\max\{p, \epsilon\})^2)$. Our lower bound suggests that this quadratic dependence on $1/\epsilon$ is inherent for efficient algorithms.

## 1 Introduction

A halfspace or Linear Threshold Function (LTF) is any Boolean function $h : \mathbb{R}^d \to \{\pm 1\}$ of the form $h(\mathbf{x}) = \text{sign}(\mathbf{w} \cdot \mathbf{x} + t)$, where $\mathbf{w} \in \mathbb{R}^d$ is the weight vector and $t \in \mathbb{R}$ is the threshold. The function $\text{sign} : \mathbb{R} \to \{\pm 1\}$ is defined as $\text{sign}(u) = 1$ if $u \geq 0$ and $\text{sign}(u) = -1$ otherwise. The problem of learning halfspaces is a classical problem in machine learning, going back to the Perceptron algorithm [Ros58] and has had a big impact in both the theory and the practice of the field [Vap98, FS97]. Here we study the problem of PAC learning halfspaces in the distribution-specific setting in the presence of Random Classification Noise (RCN) [AL88]. Specifically, we focus on the basic case in which the marginal distribution on examples is the standard Gaussian — one of the simplest and most extensively studied distributional assumptions.

In the realizable PAC model [Val84b] (i.e., when the labels are consistent with a concept in the class), the class of halfspaces on $\mathbb{R}^d$ is efficiently learnable to 0-1 error $\epsilon$ using $\widetilde{O}(d/\epsilon)$ samples via linear programming (even in the distribution-free setting). This sample complexity upper bound is information-theoretically optimal, even if we know a priori that the distribution on examples is well-behaved (e.g., Gaussian or uniform). That is, in the realizable setting, there is an efficient algorithm for halfspaces achieving the optimal sample complexity (within logarithmic factors).

37th Conference on Neural Information Processing Systems (NeurIPS 2023).

**Learning Gaussian Halfspaces with RCN.** The RCN model [AL88] is the most basic model of random noise. In this model, the label of each example is independently flipped with probability exactly $\eta$, for some noise parameter $0 < \eta < 1/2$. One of the classical results on PAC learning with RCN [Kea98] states that any Statistical Query (SQ) algorithm can be transformed into an RCN noise-tolerant PAC learner with at most a polynomial complexity blowup. Halfspaces are known to be efficiently PAC learnable in the presence of RCN, even in the distribution-free setting [BFKV97, Coh97, DKT21, DTK22]. Alas, all these efficient algorithms require sample complexity that is suboptimal within polynomial factors in $d$ and $1/\epsilon$.

The sample complexity of PAC learning Gaussian halfspaces with RCN is $\widetilde{\Theta}(d/((1-2\eta)\epsilon))$. This bound can be derived, e.g., from [MN06], and the lower bound essentially matches the realizable case, up to a necessary scaling of $(1-2\eta)$.[1] Given the fundamental nature of this learning problem, it is natural to ask whether a computationally efficient algorithm with (near-) optimal sample complexity (i.e., within logarithmic factors of the optimal) exists. That is, we are interested in a fine-grained sample size versus computational complexity analysis of the problem. This leads us to the following question:

> *Is there a sample near-optimal and polynomial-time algorithm*
> *for learning Gaussian halfspaces with RCN?*

In this paper, we explore the above question and provide two main contributions — essentially resolving the question within logarithmic factors. On the positive side, we give an efficient algorithm with sample complexity $\tilde{O}_\eta(d/\epsilon + d/(\max\{p,\epsilon\})^2)$ for the problem. Here the parameter $p \in [0, 1/2]$ (Definition 1.2) quantifies the *bias* of the target function; a "balanced" function has $p = 1/2$ and a constant function has $p = 0$. The worst-case upper bound arises when $p = \Theta(\epsilon)$, in which case our algorithm has sample complexity of $\tilde{O}_\eta(d/\epsilon^2)$. Perhaps surprisingly, *we provide formal evidence that the quadratic dependence on the quantity $1/\max\{p, \epsilon\}$ in the sample complexity cannot be improved for computationally efficient algorithms.* Our lower bounds apply for two restricted yet powerful models of computation, namely Statistical Query algorithms and low-degree polynomial tests. Our lower bounds suggest an inherent *statistical-computational tradeoff* for this problem.

## 1.1 Our Results

We study the complexity of learning halfspaces with RCN under the Gaussian distribution. Let $\mathcal{C} = \{f : \mathbb{R}^d \to \{\pm 1\} \mid f(\mathbf{x}) = \mathrm{sign}(\mathbf{w} \cdot \mathbf{x} + t)\}$ be the class of general (i.e., not necessarily homogeneous) halfspaces in $\mathbb{R}^d$. The following definition summarizes our learning problem.

**Definition 1.1** (Learning Gaussian Halfspaces with RCN). *Let $\mathcal{D}$ be a distribution on $(\mathbf{x}, y) \in \mathbb{R}^d \times \{\pm 1\}$ whose $\mathbf{x}$-marginal $\mathcal{D}_\mathbf{x}$ is the standard Gaussian. Moreover, there exists $\eta \in (0, 1/2)$ and a target $f \in \mathcal{C}$ such that the label $y$ of example $\mathbf{x}$ satisfies $y = f(\mathbf{x})$ with probability $1 - \eta$ and $y = -f(\mathbf{x})$ otherwise. Given $\epsilon > 0$ and sample access to $\mathcal{D}$, the goal is to output a hypothesis $h$ that with high probability satisfies $\mathrm{err}_{0-1}^\mathcal{D}(h) := \mathbf{Pr}_\mathcal{D}[h(\mathbf{x}) \neq y] \leq \eta + \epsilon$.*

Our main contribution is a sample near-optimal efficient algorithm for this problem coupled with a matching statistical-computational tradeoff for SQ algorithms and low-degree polynomial tests. It turns out that the sample complexity of our algorithm depends on the bias of the target halfspace, defined below.

**Definition 1.2** (*p*-biased function). *For $p \in [0, 1/2]$, we say that a Boolean function $f : \mathbb{R}^d \to \{\pm 1\}$ is p-biased with respect to the distribution $\mathcal{D}_\mathbf{x}$, if $\min\{\mathbf{Pr}_{\mathbf{x} \sim \mathcal{D}_\mathbf{x}}[f(\mathbf{x}) = 1], \mathbf{Pr}_{\mathbf{x} \sim \mathcal{D}_\mathbf{x}}[f(\mathbf{x}) = -1]\} = p$.*

For example, a homogeneous halfspace $f(\mathbf{x}) = \mathrm{sign}(\mathbf{w} \cdot \mathbf{x})$ under the standard Gaussian distribution $\mathcal{D}_\mathbf{x} = \mathcal{N}(\mathbf{0}, \mathbf{I})$ satisfies $\mathbf{E}_{\mathbf{x} \sim \mathcal{D}_\mathbf{x}}[f(\mathbf{x})] = 0$, and therefore has bias $p = 1/2$. For a general halfspace $f(\mathbf{x}) = \mathrm{sign}(\mathbf{w} \cdot \mathbf{x} + t)$ with $\|\mathbf{w}\|_2 = 1$, it is not difficult to see that its bias under the standard Gaussian is approximately $p \sim (1/t) \exp(-t^2/2)$ (see Fact B.1).

We can now state our algorithmic contribution.

**Theorem 1.3.** (Main Algorithmic Result) *There exists an algorithm that, given $\epsilon, \delta \in (0, 1/2)$ and $N$ samples from a distribution $\mathcal{D}$ satisfying Definition 1.1, runs in time $O(dN/\epsilon^2)$ and returns a*

---

[1]Throughout this introduction, it will be convenient to view $\eta$ as a constant bounded away from $1/2$.

*hypothesis $h \in \mathcal{C}$ such that with probability at least $1 - \delta$, it holds $\mathrm{err}_{0-1}^{\mathcal{D}}(h) \leq \eta + \epsilon$. The sample complexity of the algorithm is $N = \widetilde{O}\big(\frac{d}{(1-2\eta)\epsilon} + \frac{d}{\max(p(1-2\eta),\epsilon)^2}\big)\log(1/\delta)$.*

Some comments are in order. We note that the first term in the sample complexity matches the information-theoretic lower bound (within a logarithmic factor), even for homogeneous halfspaces ($p = 1/2$); see, e.g., [MN06, HY15]. The second term — scaling quadratically with $1/\max\{(1 - 2\eta)p, \epsilon\}$ — is not information-theoretically necessary and dominates the sample complexity when $p = O_\eta(\sqrt{\epsilon})$. In the worst-case, i.e., when $p = O_\eta(\epsilon)$, our algorithm has sample complexity $\widetilde{\Theta}_\eta(d/\epsilon^2)$. Perhaps surprisingly, we show in Theorem 1.5 that this quadratic dependence is required for any computationally efficient SQ algorithm; and, via [BBH+20], for any low-degree polynomial test.

**Basics on SQ Model.** SQ algorithms are a broad class of algorithms that, instead of having direct access to samples, are allowed to query expectations of bounded functions of the distribution.

**Definition 1.4** (SQ algorithms). *Let $D$ be a distribution on $\mathbb{R}^d$. A statistical query is a bounded function $q : \mathbb{R}^d \to [-1, 1]$. For $u > 0$, the $\mathrm{VSTAT}(u)$ oracle responds to the query $q$ with a value $v$ such that $|v - \mathbf{E}_{\mathbf{x} \sim D}[q(\mathbf{x})]| \leq \tau$, where $\tau = \max(1/u, \sqrt{\mathrm{Var}_{\mathbf{x} \sim D}[q(\mathbf{x})]/u})$. We call $\tau$ the* tolerance *of the statistical query. A* Statistical Query algorithm *is an algorithm whose objective is to learn some information about an unknown distribution $D$ by making adaptive calls to the corresponding oracle.*

The SQ model was introduced in [Kea98] as a natural restriction of the PAC model [Val84a]. Subsequently, the model has been extensively studied in a range of contexts, see, e.g., [Fel16]. The class of SQ algorithms is broad and captures a range of known supervised learning algorithms. More broadly, several known algorithmic techniques in machine learning are known to be implementable using SQs (see, e.g., [FGR+17, FGV17]).

We can now state our SQ lower bound result.

**Theorem 1.5** (SQ Lower Bound). *Fix any constant $c \in (0, 1/2)$ and let $d$ be sufficiently large. For any $p \geq 2^{-O(d^c)}$, any SQ algorithm that learns the class of $p$-biased halfspaces on $\mathbb{R}^d$ with Gaussian marginals in the presence of RCN with $\eta = 1/3$ to error less than $\eta + p/3$ either requires queries of accuracy better than $\widetilde{O}(pd^{c/2-1/4})$, i.e., queries to $\mathrm{VSTAT}(\widetilde{O}(d^{1/2-c}/p^2))$, or needs to make at least $2^{\Omega(d^c)}$ statistical queries.*

Informally speaking, Theorem 1.5 shows that no SQ algorithm can learn $p$-biased halfspaces in the presence of RCN (with $\eta = 1/3$) to accuracy $\eta + O(\epsilon)$ (considering $p > \epsilon/2$) with a sub-exponential in $d^{\Omega(1)}$ many queries, unless using queries of small tolerance — that would require at least $\Omega(\sqrt{d}/p^2)$ samples to simulate. This result can be viewed as a near-optimal information-computation tradeoff for the problem, within the class of SQ algorithms. When $p = 2\epsilon$, the computational sample complexity lower bound we obtain is $\Omega(\sqrt{d}/\epsilon^2)$. That is, for sufficiently small $\epsilon$, the computational sample complexity of the problem (in the SQ model) is polynomially higher than its information-theoretic sample complexity.

Via [BBH+20], we obtain a qualitatively similar lower bound in the low-degree polynomial testing model; see Appendix D.

## 1.2 Our Techniques

**Upper Bound.** At a high level, our main algorithm consists of three main subroutines. We start with a simple Initialization (warm-start) subroutine which ensures that we can choose a weight vector $\mathbf{w}_0$ with sufficiently small angle to the target vector $\mathbf{w}^*$. This subroutine essentially amounts to estimating the degree-one Chow parameters of the target function and incurs sample complexity $\widetilde{O}(d/(\max(p(1-2\eta), \epsilon)^2))$. We emphasize that our procedure does not require knowing the bias $p$ of the target halfspace; instead, it estimates this parameter to a constant factor.

Our next (and main) subroutine is an optimization procedure that is run for $\widetilde{O}(1/\epsilon^2)$ different guesses of the threshold $t$. At a high level, our optimization subroutine can be seen as a variant of Riemannian (sub)gradient descent on the unit sphere, applied to the empirical LeakyReLU loss — defined as $\mathrm{LeakyReLU}_\lambda(u) = (1 - \lambda)u\mathbb{1}\{u \geq 0\} + \lambda u\mathbb{1}\{u < 0\}$ — with parameter $\lambda$ set to $\eta$, $u = \mathbf{w} \cdot \mathbf{x}$,

and *with samples restricted to a band*, namely $a < |\mathbf{w} \cdot \mathbf{x}| < b$ — with $a$ and $b$ chosen as functions of the guess for the threshold $t$. The band restriction is key in avoiding $\Omega(d/\epsilon^2)$ dependence in the sample complexity; instead, we only require order-$(d/\epsilon)$ samples to be drawn for the empirical LeakyReLU loss subgradient estimate. Using the band, the objective is restricted to a region where the current hypothesis incorrectly classifies a constant fraction of the mass from which we can perform "denoising" with constantly many samples.

For a sufficiently accurate estimate $\hat{t}$ of $t$ (which is satisfied by at least one of the guesses for which our optimization procedure is run), we argue that there is a sufficiently negative correlation between the empirical subgradient and the target weight vector $\mathbf{w}^*$. This result, combined with our initialization, enables us to inductively argue that the distance between the weight vector constructed by the optimization procedure and the target vector $\mathbf{w}^*$ contracts and becomes smaller than $\epsilon$ within order-$\log(1/\epsilon)$ iterations. This result is quite surprising, since the LeakyReLU loss is nonsmooth (it is, in fact, piecewise linear) and we do not explicitly bound its growth outside the set of its minima (i.e., we do not prove a local error bound, which would typically be used to prove linear convergence). Thus, the result we establish is impossible to obtain using black-box results for nonsmooth optimization. Additionally, we never explicitly use the LeakyReLU loss function or prove that it is minimized by $\mathbf{w}^*$; instead, we directly prove that the vectors $\mathbf{w}$ constructed by our procedure converge to the target vector $\mathbf{w}^*$. At a technical level, our result is enabled by a novel inductive argument, which we believe may be of independent interest (see Lemma 2.8 for more details).

Since each run of our optimization subroutine returns a different hypothesis, at least one of which is accurate (the one using the "correct" guess of the threshold $t$), we need an efficient way to select a hypothesis with the desired error guarantee. This is achieved via our third subroutine — a simple hypothesis testing procedure, which draws a fresh sample and selects a hypothesis with the lowest test error. By standard results [MN06], such a hypothesis satisfies our target error guarantee.

**SQ Lower Bound.** To prove our SQ lower bound, it suffices to establish the existence of a large set of distributions whose pairwise correlations are small [FGR+17]. Inspired by the methodology of [DKS17], we achieve this by selecting our distributions on labeled examples $(\mathbf{x}, y)$ to be random rotations of a single one-dimensional distribution that nearly matches low-order Gaussian moments, and embedding this in a hidden random direction. Our hard distributions are as follows: We define the halfspaces $f_{\mathbf{v}}(\mathbf{x}) = \mathrm{sign}(\mathbf{v} \cdot \mathbf{x} - t)$, where $\mathbf{v}$ is a randomly chosen unit vector and the threshold $t$ is chosen such that $\mathbf{Pr}_{\mathbf{x} \sim \mathcal{N}}[f_{\mathbf{v}}(\mathbf{x}) = 1] = p$. We then let $y = f_{\mathbf{v}}(\mathbf{x})$ with probability $2/3$, and $-f_{\mathbf{v}}(\mathbf{x})$ otherwise. By picking a packing of nearly orthogonal vectors $\mathbf{v}$ on the unit sphere (i.e., set of vectors with pairwise small inner product), we show that each pair of these $f_{\mathbf{v}}$'s corresponding to distinct vectors in the packing have very small pairwise correlations (with respect to the distribution where $\mathbf{x}$ is a standard Gaussian and $y$ is independent of $\mathbf{x}$). While the results of [DKS17] cannot be directly applied to give our desired corerlation bounds, the Hermite analytic ideas behind them are useful in this context. In particular, the correlation between two such distributions can be computed in terms of their angle and the Hermite spectrum. A careful analysis (Lemma 3.3) gives an inner product that is $\widetilde{O}(\cos(\theta)p)$, where $\theta$ is the angle between the corresponding vectors. Combined with our packing bound, this is sufficient to obtain our final SQ lower bound result.

## 1.3 Related and Prior Work

A long line of work in theoretical machine learning has focused on developing computationally efficient algorithms for learning halfspaces under natural distributional assumptions in the presence of RCN and related semi-random noise models; see, e.g., [ABHU15, ABHZ16, YZ17, ZLC17, DKTZ20a, DKTZ20b, DKK+20, DKK+21, DKK+22]. Interestingly, the majority of these works focused on the special case of homogeneous halfspaces. We next describe in detail the most relevant prior work.

Prior work [YZ17, ZSA20, ZL21] gave sample near-optimal and computationally efficient learners for *homogeneous* halfspaces with RCN (and, more generally, bounded noise). Specifically, these works developed algorithms using near-optimal sample complexity of $\widetilde{O}_{\eta}(d/\epsilon)$. However, their algorithms and analyses are customized to the homogeneous case, and it is not clear how to extend them for general halfspaces. In fact, since all of these algorithms are easily implementable in the SQ model, our SQ lower bound (Theorem 1.5) implies that these prior algorithms *cannot* be adapted to handle the general case without an increase in sample complexity. Finally, [DKTZ22] gave an algorithm with sample complexity $\widetilde{O}(d/\epsilon^2)$ to learn general Gaussian halfspaces with adversarial

label noise to error $O(\mathrm{OPT}) + \epsilon$, where OPT is the optimal misclassification error. Unfortunately, this algorithm does not suffice for our RCN setting (where $\mathrm{OPT} = \eta$), since its error guarantee is significantly weaker than ours.

Very recent work [DDK$^+$23] gave an SQ lower bound for $\gamma$-margin halfspaces with RCN, which has some similarities to ours. Specifically, [DDK$^+$23] showed that any efficient SQ algorithm for that problem requires sample complexity $\Omega(1/(\gamma^{1/2}\epsilon^2))$. Intuitively, the margin assumption allows for a much more general family of distributions compared to our Gaussian assumption here. In particular, the SQ construction of that work does not have any implications in our setting. Even though the Gaussian distribution does not have a margin, it is easy to see that it satisfies an approximate margin property for $\gamma \sim 1/\sqrt{d}$. In fact, using an adaptation of our construction, we believe we can quantitatively strengthen the lower bound of [DDK$^+$23] to $\Omega(1/(\gamma\epsilon^2))$. For more details, see Appendix A.

## 1.4 Preliminaries

For $n \in \mathbb{Z}_+$, we define $[n] \coloneqq \{1, \ldots, n\}$. We use lowercase bold characters for vectors and uppercase bold characters for matrices. For $\mathbf{x} \in \mathbb{R}^d$ and $i \in [d]$, $\mathbf{x}_i$ denotes the $i$-th coordinate of $\mathbf{x}$, and $\|\mathbf{x}\|_2 \coloneqq (\sum_{i=1}^d \mathbf{x}_i^2)^{1/2}$ denotes the $\ell_2$-norm of $\mathbf{x}$. We use $\mathbf{x} \cdot \mathbf{y}$ for the inner product of $\mathbf{x}, \mathbf{y} \in \mathbb{R}^d$ and $\theta(\mathbf{x}, \mathbf{y})$ for the angle between $\mathbf{x}$ and $\mathbf{y}$. We slightly abuse notation and denote by $\mathbf{e}_i$ the $i^{\text{th}}$ standard basis vector in $\mathbb{R}^d$. We further use $\mathbb{1}_A$ to denote the characteristic function of the set $A$, i.e., $\mathbb{1}_A(\mathbf{x}) = 1$ if $\mathbf{x} \in A$ and $\mathbb{1}_A(\mathbf{x}) = 0$ if $\mathbf{x} \notin A$. We use the standard $O(\cdot), \Theta(\cdot), \Omega(\cdot)$ asymptotic notation. We also use $\widetilde{O}(\cdot)$ to omit poly-logarithmic factors in the argument. We use $\mathbf{E}_{x \sim \mathcal{D}}[x]$ for the expectation of the random variable $x$ according to the distribution $\mathcal{D}$ and $\mathbf{Pr}[\mathcal{E}]$ for the probability of event $\mathcal{E}$. For simplicity of notation, we omit the distribution when it is clear from the context. For $(\mathbf{x}, y)$ distributed according to $\mathcal{D}$, we denote by $\mathcal{D}_{\mathbf{x}}$ the distribution of $\mathbf{x}$. As is standard, we use $\mathcal{N}$ to denote the standard normal distribution in $d$ dimensions; i.e., with its mean being the zero vector and its covariance being the identity matrix.

## 2 Efficiently Learning Gaussian Halfspaces

In this section, we prove Theorem 1.3 by analyzing Algorithm 1. As discussed in the introduction and shown in Algorithm 1, there are three main procedures in our algorithm. The guarantees of our Initialization (warm start) procedure, which ensures sufficient correlation between the initial weight vector $\mathbf{w}_0$ and the target vector $\mathbf{w}^*$, are stated in Section 2.1, while the proofs and pseudocode are in Appendix B.1. Our main results for this section, including the Optimization procedure and associated analysis, are in Section 2.2. The Testing procedure is standard and deferred to Appendix B.2, together with most of the technical details from this section.

Throughout this section, we assume that the parameter $\eta$ (RCN parameter) is known. As will become clear from our analysis, a constant factor approximation to the value of $1 - 2\eta$ is sufficient to obtain our results. For completeness, we show how to obtain such an approximation in Appendix B.3. For simplicity, we present the results for $t \geq 0$ and $t \leq \sqrt{2\log((1-2\eta)/\epsilon)}$. This is without loss of

---

**Algorithm 1** Main Algorithm

1: **Input:** $\delta, \eta, \epsilon$, sample access to distribution $\mathcal{D}$
2: $[\mathbf{w}_0, \hat{p}] = \mathrm{Initialization}(\delta, \eta, \epsilon)$; $\epsilon' = \epsilon/(1 - 2\eta)$
3: $t_0 = \sqrt{2\log(1/\hat{p})}$, $M = 8\lceil \frac{\sqrt{2(\log(4/\hat{p}))} - \sqrt{2\log(1/\hat{p})}}{(\epsilon')^2} \rceil + 1$
4: Draw $N_2 = O(\frac{d\log(1/\delta)\log(1/\epsilon')}{(1-2\eta)^2\epsilon'})$ samples $\{(\mathbf{x}^{(i)}, y^{(i)})\}_{i=1}^{N_2}$ from $\mathcal{D}$
5: **for** $m = 1 : M$ **do**
6: $\quad$ $t_m = t_0 + (m-1)\frac{(\epsilon')^2}{8}$, $\gamma_m = \frac{\epsilon'}{2}\exp(t_m^2/2)$
7: $\quad$ $\widehat{\mathbf{w}}_m = \mathrm{Optimization}(\mathbf{w}_0, t_m, \gamma_m, \eta, \{(\mathbf{x}^{(i)}, y^{(i)})\}_{i=1}^{N_2})$
8: **end for**
9: $[\widehat{\mathbf{w}}_{\mathrm{out}}, t_{\mathrm{out}}] = \mathrm{Testing}((\widehat{\mathbf{w}}_1, t_1), (\widehat{\mathbf{w}}_2, t_2), \ldots, (\widehat{\mathbf{w}}_M, t_M))$
10: **return** $\widehat{\mathbf{w}}_{\mathrm{out}}, t_{\mathrm{out}}$

---

---

**Algorithm 2** Optimization

---

1: **Input:** $\mathbf{w}_0, \hat{t}, \hat{\gamma}, \eta, N_2$ i.i.d. samples $(\mathbf{x}^{(i)}, y^{(i)})$ from $\mathcal{D}$
2: $\mu_0 \leftarrow \frac{(1-4\rho)\sqrt{2\pi}}{16(1-2\eta)}; \rho \leftarrow 0.00098; P(\hat{t}, \hat{\gamma}) \leftarrow \mathbf{Pr}_{z \sim \mathcal{N}}[-\hat{t} \leq z \leq -\hat{t} + \hat{\gamma}]$
3: **for** $k = 0$ **to** $K$ **do**
4: $\quad$ Let $\mathcal{E}(\mathbf{w}_k, \hat{t}) := \{\mathbf{x} : -\hat{t} \leq \mathbf{w}_k \cdot \mathbf{x} \leq -\hat{t} + \hat{\gamma}\}$
5: $\quad$ Let $\mathbf{g}(\mathbf{w}_k; \mathbf{x}^{(i)}, y^{(i)}) = \frac{1}{2}((1 - 2\eta)\mathrm{sign}(\mathbf{w}_k \cdot \mathbf{x}^{(i)} + \hat{t}) - y^{(i)})\mathrm{proj}_{\mathbf{w}_k^{\perp}}(\mathbf{x}^{(i)})$
6: $\quad$ $\widehat{\mathbf{g}}(\mathbf{w}_k) \leftarrow \frac{1}{N_2} \sum_{i=1}^{N_2} \mathbf{g}(\mathbf{w}_k; \mathbf{x}^{(i)}, y^{(i)})\frac{\mathbb{1}\{\mathbf{x}^{(i)} \in \mathcal{E}(\mathbf{w}_k, \hat{t})\}}{P(\hat{t}, \hat{\gamma})}$
7: $\quad$ $\mu_k \leftarrow \mu_{k-1}(1 - \rho)$
8: $\quad$ $\mathbf{w}_{k+1} \leftarrow \frac{\mathbf{w}_k - \mu_k \widehat{\mathbf{g}}(\mathbf{w}_k)}{\|\mathbf{w}_k - \mu_k \widehat{\mathbf{g}}(\mathbf{w}_k)\|_2}$
9: **end for**
10: **return** $\mathbf{w}_{K+1}$

---

generality. For the former, it is by the simple symmetry of the standard normal distribution that the entire argument translates into the case $t < 0$, possibly by exchanging the meaning of '+1' and '-1' labels. For the latter, we note that when the bias is small, i.e., for $p \leq \epsilon/(2(1 - 2\eta))$, a constant hypothesis suffices.

## 2.1 Initialization Procedure

We begin this section with Lemma 2.1, which shows that given $N_1 = \widetilde{O}(d/(\kappa^4 p^2(1-2\eta)^2)\log(1/\delta))$ i.i.d. samples from $\mathcal{D}$, we can construct a good initial point $\mathbf{w}_0$ that forms an angle at most $\kappa$ with the target weight vector $\mathbf{w}^*$. For our purposes, $\kappa$ should be of the order $1/t$. For $t \leq \sqrt{2\log(1/\epsilon')}$, where $\epsilon' = \epsilon/(1 - 2\eta)$, we can ensure that $N_1 = \widetilde{O}(d/(p^2(1-2\eta)^2)\log(1/\delta))$. The downside of the lemma, however, is that the number of samples $N_1$ requires at least approximate knowledge of the bias parameter $p$ (or, more accurately, of $e^{-t^2/2}$). We address this challenge by arguing (in Lemma 2.2) that we can estimate $p$ using the procedure described in Algorithm 4, without increasing the total number of drawn samples by a factor larger than order-$\log(1/\epsilon')$.

**Lemma 2.1** (Initialization via Chow Parameters). *Given $\kappa > 0$, define $p_t = e^{-t^2/2}$, $N_1 = O(d/(\kappa^4 p_t^2(1-2\eta)^2)\log(1/\delta))$ and let $(\mathbf{x}^{(i)}, y^{(i)})$ for $i \in [N_1]$ be i.i.d. samples drawn from $\mathcal{D}$. Let $\mathbf{u} = \frac{1}{N_1}\sum_{i=1}^{N_1}\mathbf{x}^{(i)}y^{(i)}$ and $\mathbf{w}_0 = \mathbf{u}/\|\mathbf{u}\|_2$. Then, with probability $1 - \delta$, we have $\theta(\mathbf{w}_0, \mathbf{w}^*) \leq \kappa$.*

We now leverage Lemma 2.1 to argue about the correctness of implementable Initialization procedure, stated as Algorithm 4 in Appendix B.1, where the proofs for this subsection can be found.

**Lemma 2.2.** *Consider the Initialization procedure described by Algorithm 4 in Appendix B.1. If $0 \leq t \leq \sqrt{2\log((1-2\eta)/\epsilon)}$, then with probability at least $1 - \delta$, $\exp(-t^2/2) \leq \hat{p} \leq 4\exp(-t^2/2)$. The algorithm draws a total of $\widetilde{O}\left(\frac{d\log(1/\delta)}{\max\{(1-2\eta)p, \epsilon\}^2}\right)$ samples and ensures that $\theta(\mathbf{w}_0, \mathbf{w}^*) \leq \min\{\frac{1}{5t}, \frac{\pi}{2}\}$.*

## 2.2 Optimization

As discussed before, our Optimization procedure (Algorithm 2) can be seen as Riemannian subgradient descent on the unit sphere. Crucial to our analysis is the use of subgradient estimates from Line 5 and Line 6, where we condition on the event that the samples come from a thin band, defined in Line 4. Without this conditioning, the algorithm would correspond to projected subgradient descent of the LeakyReLU loss on the unit sphere. The conditioning effectively changes the landscape of the loss function being optimized, which cannot be argued anymore to even be convex, as the definition of the band depends on the weight vector $\mathbf{w}$ at which the vector $\widehat{\mathbf{g}}(\mathbf{w})$ is evaluated. Nevertheless, as we argue in this section, the optimization procedure can be carried out very efficiently, even exhibiting a linear convergence rate. To simplify the notation, in this section we denote the conditioned distribution $\mathcal{D}|_{\mathcal{E}(\mathbf{w}, \hat{t})}$ by $\mathcal{D}(\mathbf{w}, \hat{t})$. We carry out the analysis assuming the estimate $\hat{t}$ is within additive $\epsilon^2$ of the true threshold value $t$; as argued before, this has to be true for at least one estimate $\hat{t}$ for which the Optimization procedure is invoked.

In the following lemma, we show that if the angle between a weight vector $\mathbf{w}$ and the target vector $\mathbf{w}^*$ is from a certain range, we can guarantee that $\mathbf{g}(\mathbf{w})$ is sufficiently negatively correlated with $\mathbf{w}^*$. This condition is then used to argue about progress of our algorithm. The upper bound on $\theta$ will hold initially, by our initialization procedure, and we will inductively argue that it holds for all iterations. The lower bound, when violated, will imply that the distance between $\mathbf{w}$ and $\mathbf{w}^*$ is small, in which case we would have converged to a sufficiently good solution $\mathbf{w}$.

**Lemma 2.3.** *Fix any $\epsilon' \in (0,1)$. Suppose that $0 \le t \le \sqrt{2\log(1/\epsilon')}$ and $\mathbf{w} \in \mathbb{R}^d$ is such that $\|\mathbf{w}\|_2 = 1$, and $\theta = \theta(\mathbf{w}, \mathbf{w}^*)$ satisfies the inequality $\epsilon' \exp(t^2/2) \le \theta \le 1/(5t)$. If $|\hat{t} - t| \le \epsilon'^2/8$ and $\hat{\gamma} = (1/2)\epsilon' \exp(\hat{t}^2/2)$, then $\mathbf{E}_{(\mathbf{x},y)\sim\mathcal{D}(\mathbf{w},\hat{t})}[\mathbf{g}(\mathbf{w};\mathbf{x},y) \cdot \mathbf{w}^*] \le -(1-2\eta)\sin\theta/(2\sqrt{2\pi})$.*

Since, by construction, $\mathbf{g}(\mathbf{w})$ is orthogonal to $\mathbf{w}$ (see Line 5 in Algorithm 2), we can bound the norm of the expected gradient vector by bounding $\mathbf{g}(\mathbf{w}) \cdot \mathbf{u}$ for some unit vectors $\mathbf{u}$ that are orthogonal to $\mathbf{w}$ using similar techniques as in Lemma 2.3. To be specific, we have the following lemma.

**Lemma 2.4.** *Under the assumptions of Lemma 2.3, $\big\| \mathbf{E}_{(\mathbf{x},y)\sim\mathcal{D}(\mathbf{w},\hat{t})}[\mathbf{g}(\mathbf{w};\mathbf{x},y)]\big\|_2 \le \frac{(1-2\eta)}{\sqrt{2\pi}}$.*

The last technical ingredient that we need is the following lemma which shows a uniform bound on the difference between the empirical gradient $\widehat{\mathbf{g}}(\mathbf{w})$ and its expectation (for more details, see Lemma B.6 and Corollary B.8 in Appendix B).

**Lemma 2.5.** *Consider the learning problem from Definition 1.1. Let $\epsilon', \hat{t}, \hat{\gamma}$ be parameters satisfying the conditions of Lemma 2.3. Let $\delta \in (0,1)$. Then using $\widetilde{O}(d\log(1/\delta)/((1-2\eta)^2\epsilon'))$ samples to construct $\widehat{\mathbf{g}}$, for any unit vector $\mathbf{w}$ such that $\epsilon' \exp(t^2/2) \le \theta(\mathbf{w},\mathbf{w}^*) \le 1/(5t)$, it holds with probability at least $1 - \delta$: $\|\widehat{\mathbf{g}}(\mathbf{w}) - \mathbf{E}_{(\mathbf{x},y)\sim\mathcal{D}(\mathbf{w},\hat{t})}[\mathbf{g}(\mathbf{w})]\|_2 \le (1/4)\| \mathbf{E}_{(\mathbf{x},y)\sim\mathcal{D}(\mathbf{w},\hat{t})}[\mathbf{g}(\mathbf{w})]\|_2$.*

We are now ready to present and prove our main algorithm-related result. A short roadmap for our proof is as follows. Since Algorithm 1 constructs a grid with grid-width $\epsilon'^2/8$ that covers all possible values of the true threshold $t$, there exists at least one guess $\hat{t}$ that is $\epsilon'^2$-close to the true threshold $t$. We first show that to get a halfspace with error at most $\epsilon'$, it suffices to use this $\hat{t}$ as the threshold and find a weight vector $\mathbf{w}$ such that the angle $\theta(\mathbf{w}, \mathbf{w}^*)$ is of the order $\epsilon'$, which is exactly what Algorithm 2 does. The connection between $\theta(\mathbf{w}, \mathbf{w}^*)$ and the error is conveyed by the inequality $\mathbf{Pr}[\text{sign}(\mathbf{w} \cdot \mathbf{x} + t) \ne \text{sign}(\mathbf{w}^* \cdot \mathbf{x} + t)] \le (\theta(\mathbf{w}, \mathbf{w}^*)/\pi)\exp(-t^2/2)$; see Appendix B. Let $\mathbf{w}_k$ be the parameter generated by Algorithm 2 at iteration $k$ for threshold $\hat{t}$. We show that $\theta(\mathbf{w}_k, \mathbf{w}^*)$ converges to zero at a linear rate. To this end, we prove that under our carefully devised step size $\mu_k$, there exists an upper bound on $\|\mathbf{w}_k - \mathbf{w}^*\|_2$, which contracts at each iteration. Note that since both $\mathbf{w}_k$ and $\mathbf{w}^*$ are on the unit sphere, we have $\|\mathbf{w}_k - \mathbf{w}^*\|_2 = 2\sin(\theta(\mathbf{w}_k, \mathbf{w}^*)/2)$. Essentially, this implies that Algorithm 2 produces a sequence of parameters $\mathbf{w}_k$ such that $\theta(\mathbf{w}_k, \mathbf{w}^*)$ converges to 0 linearly, under this threshold $\hat{t}$. Thus, we can conclude that there exists a halfspace among all halfspaces generated by Algorithm 1 that achieves $\epsilon'$ error with high probability.

**Theorem 2.6.** *Consider the learning problem from Definition 1.1. Fix any unit vector $\mathbf{w}_0 \in \mathbb{R}^d$ such that $\theta(\mathbf{w}_0, \mathbf{w}^*) \le \min(1/(5t), \pi/2)$. Fix any $\epsilon, \delta > 0$. Let $\hat{t} > 0$ be a threshold such that $|\hat{t} - t| \le \epsilon^2/(8(1-2\eta)^2)$, and let $\hat{\gamma} = \epsilon/(2(1-2\eta))\exp(\hat{t}^2/2)$. Then Algorithm 2 uses $N_2 = \widetilde{O}\big(d/((1-2\eta)\epsilon)\log(1/\delta)\big)$ samples from $\mathcal{D}$, has runtime $\widetilde{O}(N_2 d)$, and outputs a weight vector $\mathbf{w}$ such that $h(\mathbf{x}) = \text{sign}(\mathbf{w} \cdot \mathbf{x} + \hat{t})$ satisfies $\mathbf{Pr}[h(\mathbf{x}) \ne y] \le \eta + \epsilon$ with probability at least $1 - \delta$.*

*Proof.* Let $\epsilon' = (\epsilon/1 - 2\eta)$, and denote by $\mathbf{w}_k$ the vector produced by the algorithm at $k^{\text{th}}$ iteration for threshold $\hat{t}$. For any unit vector $\mathbf{w}$ and $|\hat{t} - t| \le \epsilon'^2/8$, it holds $\mathbf{Pr}[\text{sign}(\mathbf{w} \cdot \mathbf{x} + \hat{t}) \ne \text{sign}(\mathbf{w}^* \cdot \mathbf{x} + t)] \le \epsilon'^2/(4\sqrt{2\pi}) + \theta(\mathbf{w}, \mathbf{w}^*)/\pi \exp(-t^2/2)$ (see Appendix B.2 for more details). Therefore, it suffices to find a parameter $\mathbf{w}$ such that $\theta(\mathbf{w}, \mathbf{w}^*) \le \pi\epsilon' \exp(t^2/2)$. Note that since both $\mathbf{w}$ and $\mathbf{w}^*$ are unit vectors, we have $\|\mathbf{w} - \mathbf{w}^*\|_2 = 2\sin(\theta/2)$, indicating that it suffices to minimize $\|\mathbf{w} - \mathbf{w}^*\|_2$ efficiently. As proved in Section 2.1, we can start with an initial vector $\mathbf{w}_0$ such that $\theta(\mathbf{w}_0, \mathbf{w}^*) \le 1/(5t)$ by calling Algorithm 4 (in Appendix B.1). Denote $\theta_k = \theta(\mathbf{w}_k, \mathbf{w}^*)$ and consider the case when $\theta_k \ge \epsilon' \exp(t^2/2)$. We establish the following claim:

**Claim 2.7.** *Let $C_1 := (1-2\eta)/\sqrt{2\pi}$. Drawing $N_2 = \widetilde{O}(d\log(1/\delta)/((1-2\eta)^2\epsilon'))$ samples from distribution $\mathcal{D}$, we have that if $\theta_k \ge \epsilon' \exp(t^2/2)$ then with probability at least $1 - \delta$: $\|\mathbf{w}_{k+1} - \mathbf{w}^*\|_2^2 \le \|\mathbf{w} - \mathbf{w}^*\|_2^2 - (C_1/2)\mu_k \sin\theta_k + 4C_1^2\mu_k^2$.*

It remains to choose the step size $\mu_k$ properly to get linear convergence. By carefully designing a shrinking step size, we are able to construct an upper bound $\phi_k$ on the distance of $\|\mathbf{w}_{k+1} - \mathbf{w}_k\|_2$ using Claim 2.7. Importantly, by exploiting the property that both $\mathbf{w}$ and $\mathbf{w}^*$ are on the unit sphere, we show that the upper bound is contracting at each step, even though the distance $\|\mathbf{w}_{k+1} - \mathbf{w}_k\|_2$ could be increasing. Concretely, we have the following lemma.

**Lemma 2.8.** *Let $\rho = 0.00098$ and $\phi_k = (1 - \rho)^k$. Then, setting $\mu_k = (1 - 4\rho)\phi_k/(16C_1)$ it holds $\sin(\theta_k/2) \leq \phi_k$ for $k = 1, \cdots, K$.*

*Proof.* Let $\phi_k = (1 - \rho)^k$ where $\rho = 0.00098$. This choice of $\rho$ ensures that $32\rho^2 + 1020\rho - 1 \leq 0$. We show by induction that choosing $\mu_k = (1 - 4\rho)\phi_k/(16C_1) = (1 - \rho)^k(1 - 4\rho)/(16C_1)$, it holds $\sin(\theta_k/2) \leq \phi_k$. The condition certainly holds for $k = 1$ since $\theta_1 \in [0, \pi/2]$. Now suppose that $\sin(\theta_k/2) \leq \phi_k$ for some $k \geq 1$. We discuss the following 2 cases: $\phi_k \geq \sin(\theta_k/2) \geq \frac{3}{4}\phi_k$ and $\sin(\theta_k/2) \leq \frac{3}{4}\phi_k$. First, suppose $\phi_k \geq \sin(\theta_k/2) \geq \frac{3}{4}\phi_k$. Since $\sin(\theta_k/2) \leq \sin\theta_k$, it also holds $\sin\theta_k \geq \frac{3}{4}\phi_k$. Bringing in the fact that $\|\mathbf{w}_{k+1} - \mathbf{w}^*\|_2 = 2\sin(\theta_{k+1}/2)$ and $\|\mathbf{w}_k - \mathbf{w}^*\|_2 = 2\sin(\theta_k/2)$, as well as the definition of $\mu_k$, the conclusion of Claim 2.7 becomes:

$$(2\sin(\theta_{k+1}/2))^2 \leq (2\sin(\theta_k/2))^2 - (C_1/2)\mu_k \sin\theta_k + 4C_1^2(1 - 4\rho)\phi_k\mu_k/(16C_1)$$
$$\leq 4\phi_k^2 - 3C_1\mu_k\phi_k/8 + C_1(1 - 4\rho)\mu_k\phi_k/4 = 4\phi_k^2(1 - (1 + 8\rho)(1 - 4\rho)/512),$$

where in the second inequality we used $\sin\theta_k \geq \frac{3}{4}\phi_k$ and in the last equality we used the definition of $\mu_k$ by which $\mu_k = (1 - 4\rho)\phi_k/(16C_1)$. Since $\rho$ is chosen so that $32\rho^2 + 1020\rho - 1 \leq 0$, we have:

$$\sin(\theta_{k+1}/2) \leq \phi_k\sqrt{1 - (1 + 8\rho)(1 - 4\rho)/512} \leq (1 - \rho)\phi_k = (1 - \rho)^{k+1},$$

as desired. Next, consider $\sin(\theta_k/2) \leq (3/4)\phi_k$. Recall that $\mathbf{w}_{k+1} = \text{proj}_{\mathbb{B}}(\mathbf{w}_k - \mu_k\widehat{\mathbf{g}}(\mathbf{w}_k))$ and $\mathbf{w}_k \in \mathbb{B}$, where $\mathbb{B}$ is the unit ball[2]; therefore, $\|\mathbf{w}_{k+1} - \mathbf{w}_k\|_2 \leq \|\mathbf{w}_k - \mu_k\widehat{\mathbf{g}}(\mathbf{w}) - \mathbf{w}_k\|_2 = \mu_k\|\widehat{\mathbf{g}}(\mathbf{w}_k)\|_2$ by the non-expansiveness of the projection operator. Furthermore, applying Lemma 2.5 and Lemma 2.4, it holds that $\|\widehat{\mathbf{g}}(\mathbf{w}_k)\|_2 \leq (5/4)\|\mathbf{E}_{(\mathbf{x},y)\sim\mathcal{D}(\mathbf{w},\hat{t})}[\mathbf{g}(\mathbf{w})]\|_2 \leq 2(1 - 2\eta)/\sqrt{2\pi}$, i.e., we have $\|\widehat{\mathbf{g}}(\mathbf{w}_k)\|_2 \leq 2C_1$; therefore, $\|\mathbf{w}_{k+1} - \mathbf{w}_k\|_2 \leq 2\mu_kC_1$, which indicates that:

$$2(\sin(\theta_{k+1}/2) - \sin(\theta_k/2)) = \|\mathbf{w}_{k+1} - \mathbf{w}^*\|_2 - \|\mathbf{w}_k - \mathbf{w}^*\|_2 \leq \|\mathbf{w}_{k+1} - \mathbf{w}_k\|_2 \leq 2\mu_kC_1.$$

Since we have assumed $\sin(\theta_k/2) \leq (3/4)\phi_k$, then it holds:

$$\phi_{k+1} - \sin(\theta_{k+1}/2) \geq (1 - \rho)\phi_k - \phi_k + \phi_k - \sin(\theta_k/2) - (1 - 4\rho)\phi_k/16 \geq 3(1 - 4\rho)\phi_k/16 > 0,$$

since we have chosen $\mu_k = (1 - 4\rho)\phi_k/(16C_1)$. Hence, it also holds that $\sin(\theta_{k+1}/2) \leq \phi_{k+1}$. $\square$

Lemma 2.8 shows that $\sin(\theta_k/2)$ converges to 0 linearly. Therefore, using $N_2 = \widetilde{O}(d\log(1/\delta)/((1 - 2\eta)^2\epsilon'))$ samples, after $K = O((1/\rho)\log(1/(\exp(t^2/2)\epsilon'))) = O(\log(1/\epsilon'))$ iterations, we get a $\mathbf{w}_K$ such that $\theta_K \leq 2\sin(\theta_K/2) \leq \epsilon'\exp(t^2/2)$. Let $h(\mathbf{x}) := \text{sign}(\mathbf{w}_K \cdot \mathbf{x} + \hat{t})$. Then it holds that the disagreement of $h(\mathbf{x})$ and $f(\mathbf{x})$ is bounded by $\mathbf{Pr}[h(\mathbf{x}) \neq f(\mathbf{x})] \leq \epsilon'$ (see Appendix B for more details). Finally, since $\text{err}_{0-1}^{\mathcal{D}}(h) = \mathbf{Pr}_{(\mathbf{x},y)\sim\mathcal{D}}[h(\mathbf{x}) \neq y] = \eta + (1 - 2\eta)\mathbf{Pr}_{\mathbf{x}\sim\mathcal{D}_\mathbf{x}}[h(\mathbf{x}) \neq \text{sign}(\mathbf{w}^* \cdot \mathbf{x} + t)]$, for any $h : \mathbb{R}^d \mapsto \{\pm 1\}$, to get misclassification error at most $\eta + \epsilon$ (with respect to the $y$), it suffices to use $\epsilon' = \epsilon/(1 - 2\eta)$. Therefore, we get $\mathbf{Pr}_{(\mathbf{x},y)\sim\mathcal{D}}[\text{sign}(\mathbf{w}_K \cdot \mathbf{x} + \hat{t}) \neq y] \leq \eta + \epsilon$, using $N_2 = \widetilde{O}(d\log(1/\delta)/((1 - 2\eta)\epsilon))$ samples. Since the algorithm runs for $O(\log(1/\epsilon))$ iterations, the overall runtime is $\widetilde{O}(N_2d)$. This completes the proof of Theorem 2.6. $\square$

*Proof Sketch of Theorem 1.3.* From Lemma 2.2, we get that with $\widetilde{O}(d/((1 - 2\eta)^2p^2))$ samples our Initialization procedure (Algorithm 4) produces a unit vector $\mathbf{w}_0$ so that $\theta(\mathbf{w}_0, \mathbf{w}^*) \leq \min(1/(5t), \pi/2)$ with high probability. We construct a grid of $(\epsilon^2/(8(1 - 2\eta)^2))$-separated values, containing all the possible values of the threshold $t$ of size roughly $\sim 1/\epsilon^2$. We run Algorithm 2 for each possible choice of the threshold $t$. Conditioned on the choice of $\hat{t}$ and $\mathbf{w}_0$ that satisfies the assumptions of Theorem 2.6, Algorithm 2 outputs a weight vector $\widehat{\mathbf{w}}$ so that $\mathbf{Pr}_{(\mathbf{x},y)\sim\mathcal{D}}[\text{sign}(\widehat{\mathbf{w}} \cdot \mathbf{x} + \hat{t}) \neq y] \leq \eta + \epsilon$. Using standard concentration facts, we have that with a sample size of order $\widetilde{O}(d/((1 - 2\eta)\epsilon))$ from $\mathcal{D}$, we can output the hypothesis with the minimum empirical error with high probability. $\square$

---

[2]This is true because $\widehat{\mathbf{g}}(\mathbf{w}_k)$ is orthogonal to $\mathbf{w}_k$, and thus $\|\mathbf{w}_k - \mu_k\widehat{\mathbf{g}}(\mathbf{w}_k)\|_2 > 1$, meaning that projections onto the unit ball and the unit sphere are the same in this case.

# 3 SQ Lower Bound for Learning Gaussian Halfspaces with RCN

To state our SQ lower bound theorem, we require the following standard definition.

**Definition 3.1** (Decision/Testing Problem over Distributions). *Let $D$ be a distribution and $\mathfrak{D}$ be a family of distributions over $\mathbb{R}^d$. We denote by $\mathcal{B}(\mathfrak{D}, D)$ the decision (or hypothesis testing) problem in which the input distribution $D'$ is promised to satisfy either (a) $D' = D$ or (b) $D' \in \mathfrak{D}$, and the goal of the algorithm is to distinguish between these two cases.*

**Theorem 3.2** (SQ Lower Bound for Testing RCN Halfspaces). *Fix $c \in (0, 1/2)$ and let $d \in \mathbb{N}$ be sufficiently large. For any $p \geq 2^{-O(d^c)}$, any SQ algorithm that learns the class of (at most) $p$-biased Gaussian halfspaces on $\mathbb{R}^d$ in the presence of RCN with $\eta = 1/3$ to error less than $\eta + p/3$ either requires queries to $\mathrm{VSTAT}(\widetilde{O}(d^{1/2-c}/p^2))$, or needs to make at least $2^{\Omega(d^c)}$ statistical queries.*

We note that our SQ lower bound applies to a natural testing version of our learning problem. By a standard reduction (see Lemma C.9), it follows that any learning algorithm for the problem requires either $2^{\Omega(d^c)}$ many queries or at least one query to $\mathrm{VSTAT}(\widetilde{O}(d^{2c-1/2}/p^2))$. We also note that the established bound is tight for the corresponding testing problem (see Appendix C.6).

*Proof of Theorem 3.2.* For any unit vector $\mathbf{v} \in \mathbb{R}^d$, we define the LTF $f_{\mathbf{v}}(\mathbf{x}) = \mathrm{sign}(\mathbf{v} \cdot \mathbf{x} - t)$, where $t > 0$ and denote $p = \mathbf{Pr}_{\mathbf{x} \sim \mathcal{N}}[f_{\mathbf{v}}(\mathbf{x}) = 1]$. Let $D_{\mathbf{v}}$ be the distribution on $(\mathbf{x}, y)$ with respect to $f_{\mathbf{v}}$ with the random variable $y$ supported on $\{\pm 1\}$ as follows: $\mathbf{Pr}[y = f_{\mathbf{v}}(\mathbf{x}) \mid \mathbf{x}] = 1 - \eta$ and $\mathbf{x}$ is distributed as standard normal. Denote by $A_{\mathbf{v}}$ the distribution $D_{\mathbf{v}}$ conditioned on $y = 1$ and by $B_{\mathbf{v}}$ the distribution $D_{\mathbf{v}}$ conditioned on $y = -1$. It is easy to see that

$$A_{\mathbf{v}}(\mathbf{x}) = G(\mathbf{x})(\eta + (1 - 2\eta)\mathbb{1}\{f_{\mathbf{v}}(\mathbf{x}) > 0\})/(\eta + (1 - 2\eta)p)$$

and

$$B_{\mathbf{v}}(\mathbf{x}) = G(\mathbf{x})(1 - \eta - (1 - 2\eta)\mathbb{1}\{f_{\mathbf{v}}(\mathbf{x}) > 0\})/(1 - \eta - (1 - 2\eta)p) .$$

Fix unit vectors $\mathbf{v}, \mathbf{u} \in \mathbb{R}^d$ and let $\theta$ be the angle between them. We bound from above the correlation between $f_{\mathbf{v}}(\mathbf{x})$ and $f_{\mathbf{u}}(\mathbf{x})$. Our main technical lemma is the following:

**Lemma 3.3.** *Let $f_{\mathbf{v}}(\mathbf{x})$ and $f_{\mathbf{u}}(\mathbf{x})$ defined as above. Then it holds*

$$\Big| \mathop{\mathbf{E}}_{\mathbf{x} \sim \mathcal{N}}[f_{\mathbf{v}}(\mathbf{x})f_{\mathbf{u}}(\mathbf{x})] - \mathop{\mathbf{E}}_{\mathbf{x} \sim \mathcal{N}}[f_{\mathbf{v}}(\mathbf{x})] \mathop{\mathbf{E}}_{\mathbf{x} \sim \mathcal{N}}[f_{\mathbf{u}}(\mathbf{x})]\Big| \leq 4|\cot(\theta)| \exp(-t^2) \exp\left(|\cos(\theta)|t^2\right) .$$

*Proof of Lemma 3.3.* We start by calculating the Hermite coefficients of the univariate function $\mathrm{sign}(z - t)$. We will use the fact that $\mathbf{E}_{z \sim \mathcal{N}}[\mathrm{sign}(z - t)\mathrm{He}_i(z)] = 2i^{-1/2}\mathrm{He}_{i-1}(t) \exp(-t^2/2)$ (see Claim C.7). Let $c_i$ be the Hermite coefficient of degree $i$. Without loss of generality (due to the rotational invariance of the Gaussian distribution), we can assume that $\mathbf{v} = \mathbf{e}_1$ and $\mathbf{u} = \cos\theta\mathbf{e}_1 + \sin\theta\mathbf{e}_2$. Using standard algebraic manipulations and orthogonality arguments (see Claim C.5 for more details), we have that

$$\mathop{\mathbf{E}}_{\mathbf{x} \sim \mathcal{N}}[f_{\mathbf{v}}(\mathbf{x})f_{\mathbf{u}}(\mathbf{x})] = \mathop{\mathbf{E}}_{\mathbf{x}_1, \mathbf{x}_2 \sim \mathcal{N}}[\mathrm{sign}(\mathbf{x}_1 - t)\mathrm{sign}(\cos\theta\mathbf{x}_1 + \sin\theta\mathbf{x}_2 - t)] = \sum_{i \geq 0} \cos^i\theta\, c_i^2 .$$

Note that $\mathrm{He}_0(\mathbf{x}) = 1$, therefore $c_0 = \mathbf{E}_{\mathbf{x} \sim \mathcal{N}}[f_{\mathbf{v}}(\mathbf{x})]$. Therefore, we have that

$$\mathop{\mathbf{E}}_{\mathbf{x} \sim \mathcal{N}}[f_{\mathbf{v}}(\mathbf{x})f_{\mathbf{u}}(\mathbf{x})] = \sum_{i \geq 1} \cos^i\theta c_i^2 + \mathop{\mathbf{E}}_{\mathbf{x} \sim \mathcal{N}}[f_{\mathbf{v}}(\mathbf{x})] \mathop{\mathbf{E}}_{\mathbf{x} \sim \mathcal{N}}[f_{\mathbf{u}}(\mathbf{x})] .$$

Let $J = \sum_{i \geq 1} \cos^i\theta c_i^2$. To complete the proof, it remains to bound the $|J|$. We show the following:

**Claim 3.4.** *It holds that $|J| \leq 4|\cot(\theta)| \exp(-t^2) \exp\left(|\cos(\theta)|t^2\right)$.*

*Proof of Claim 3.4.* Note that from Claim C.7, we have that $c_i = 2\mathrm{He}_{i-1}(t) \exp(-t^2/2)/\sqrt{i}$, hence, it holds that $J = 4\cos(\theta) \exp(-t^2) \sum_{i=1}^{\infty} i^{-1}\mathrm{He}_{i-1}^2(t) \cos^{i-1}\theta$. We use the following fact.

**Fact 3.5** (Mehler Formula, see, e.g. [Foa78]). *For $|\rho| < 1$ and $x, y \in \mathbb{R}$, it holds that*

$$\exp\left(-\frac{1}{2}\frac{\rho}{1 - \rho^2}(x - y)^2\right) = \sqrt{1 - \rho^2} \sum_{k \geq 0} \rho^k \mathrm{He}_k(x)\mathrm{He}_k(y) \exp\left(-\frac{1}{2}\frac{\rho}{1 + \rho^2}(x^2 + y^2)\right) .$$

Applying Fact 3.5 for $\rho = \cos(\theta)$ and $x = y = t$, we get that

$$|J| = 4\big|\cos\theta \exp(-t^2) \sum_{i \geq 1} i^{-1} \mathrm{He}_{i-1}^2(t) \cos^{i-1}\theta\big| \leq 4|\cos\theta| \exp(-t^2) \sum_{i \geq 0} \mathrm{He}_i^2(t)|\cos\theta|^i$$

$$= 4|\cot\theta| \exp(-t^2) \exp\left(\frac{|\cos(\theta)|t^2}{(1+\cos^2\theta)}\right).$$

This completes the proof of Claim 3.4. $\hfill\square$

Using Claim 3.4, we get that

$$\big| \mathop{\mathbf{E}}_{\mathbf{x}\sim\mathcal{N}}[f_{\mathbf{v}}(\mathbf{x})f_{\mathbf{u}}(\mathbf{x})] - \mathop{\mathbf{E}}_{\mathbf{x}\sim\mathcal{N}}[f_{\mathbf{v}}(\mathbf{x})]\mathop{\mathbf{E}}_{\mathbf{x}\sim\mathcal{N}}[f_{\mathbf{u}}(\mathbf{x})]\big| \leq 4|\cot(\theta)|\exp(-t^2)\exp\left(|\cos(\theta)|t^2\right),$$

completing the proof of Lemma 3.3. $\hfill\square$

We associate each $\mathbf{v}$ and $\mathbf{u}$ to a distribution $D_{\mathbf{v}}$ and $D_{\mathbf{u}}$, constructed as above. The following lemma provides explicit bounds on the correlation between the distributions $D_{\mathbf{v}}$ and $D_{\mathbf{u}}$. Recall that the pairwise correlation of two distributions with cdfs $D_1, D_2$ with respect to a distribution with cdf $D$ is defined as $\chi_D(D_1, D_2) + 1 := \int_{x \in \mathcal{X}} D_1(x)D_2(x)/D(x)$ (see Definition C.1). We have the following lemma (see Appendix C.4 for its proof):

**Lemma 3.6.** *Let $D_0$ be a product distribution distributed as $\mathcal{N} \times \{\pm 1\}$, where $\mathbf{Pr}_{(\mathbf{x},y)\sim D_0}[y = 1] = \mathbf{Pr}_{(\mathbf{x},y)\sim D_{\mathbf{v}}}[y = 1] = p$. We have $\chi_{D_0}(D_{\mathbf{v}}, D_{\mathbf{u}}) \leq 2(1-2\eta)(\mathbf{E}[f_{\mathbf{v}}(\mathbf{x})f_{\mathbf{u}}(\mathbf{x})] - \mathbf{E}[f_{\mathbf{v}}(\mathbf{x})]\mathbf{E}[f_{\mathbf{u}}(\mathbf{x})])$ and $\chi^2(D_{\mathbf{v}}, D_0) \leq (1-2\eta)(\mathbf{E}[f_{\mathbf{v}}(\mathbf{x})] - \mathbf{E}[f_{\mathbf{v}}(\mathbf{x})]^2)$.*

For any $c \in (0, 1/2)$, there exists a set $\mathcal{S}$ of $2^{\Omega(d^c)}$ unit vectors in $\mathbb{R}^d$ such that for any pair $\mathbf{v} \neq \mathbf{u} \in \mathcal{S}$ satisfies $|\mathbf{v} \cdot \mathbf{u}| < d^{-1/2+c}$ (Fact C.4). We associate each $\mathbf{v} \in \mathcal{S}$ with $f_{\mathbf{v}}$ and a distribution $D_{\mathbf{v}}$ and denote $\mathfrak{D} = \{D_{\mathbf{v}}, \mathbf{v} \in \mathcal{S}\}$. By the definition of $\mathcal{S}$ and Lemma 3.3, for any $\mathbf{v}, \mathbf{u} \in \mathcal{S}$, we have that $\mathbf{E}_{\mathbf{x}\sim\mathcal{N}}[f_{\mathbf{v}}(\mathbf{x})f_{\mathbf{u}}(\mathbf{x})] \leq 4d^{-1/2+c}\exp(-t^2)\exp\left((t/d^{1/4-c/2})^2\right) + \mathbf{E}_{\mathbf{x}\sim\mathcal{N}}[f_{\mathbf{v}}(\mathbf{x})]\mathbf{E}_{\mathbf{x}\sim\mathcal{N}}[f_{\mathbf{u}}(\mathbf{x})]$. Since $t/d^{1/4-c/2} \leq 1/2$ by assumption, we get that $|\mathbf{E}_{\mathbf{x}\sim\mathcal{N}}[f_{\mathbf{v}}(\mathbf{x})f_{\mathbf{u}}(\mathbf{x})] - \mathbf{E}_{\mathbf{x}\sim\mathcal{N}}[f_{\mathbf{v}}(\mathbf{x})]\mathbf{E}_{\mathbf{x}\sim\mathcal{N}}[f_{\mathbf{u}}(\mathbf{x})]| \leq d^{-1/2+c}\exp(-t^2)$. By Lemma 3.6, it follows that $\chi_{D_0}(D_{\mathbf{v}}, D_{\mathbf{u}}) \leq C(1 - 2\eta)\exp(-t^2)d^{-1/2+c}$ and $\chi^2(D_{\mathbf{v}}, D_0) \leq C(1 - 2\eta)\exp(-t^2/2)$, where $C > 0$ is an absolute constant. From standard SQ machinery (see, e.g., Lemma C.3), we have that any SQ algorithm that solves the decision problem $\mathcal{B}(\mathfrak{D}, D_0)$, requires either $2^{\Omega(d^c)}$ queries, or at least one query to $\mathrm{VSTAT}(\exp(t^2)d^{1/2+c})$. Noting that $p = O(\exp(-t^2/2)/t)$ (by Fact B.1) completes the proof of Theorem 3.2. $\hfill\square$

## 4 Acknowledgements

ID was supported by NSF Medium Award CCF-2107079, NSF Award CCF-1652862 (CAREER), and a DARPA Learning with Less Labels (LwLL) grant. JD was supported by NSF Award CCF-2007757 and by the U. S. Office of Naval Research under award number N00014-22-1-2348. DK was supported by NSF Medium Award CCF-2107547 and NSF Award CCF-1553288 (CAREER). PW was supported in part by NSF Award CCF-2007757. NZ was supported in part by NSF award 2023239, NSF Medium Award CCF-2107079, and a DARPA Learning with Less Labels (LwLL) grant.

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

# Appendix

**Organization** The appendix is organized as follows: In Appendix A, we provide additional comparison to prior work. In Appendix B, we present the full version of Section 2, completing proofs and providing supplementary lemmas omitted in the main body. In Appendix C, we start with background on Hermite polynomials and the SQ model, followed by omitted proofs from Section 3. Finally, in Appendix D, we prove our lower bound for low-degree polynomial testing.

## A   Comparison with Previous Work

Here we provide a more detailed comparison with the very recent work of [DDK$^+$23], which gives an algorithm and an SQ lower bound for learning $\gamma$-margin halfspaces in the presence of RCN. We start by noting that the SQ-hard instance of [DDK$^+$23] is based on a discrete distribution on the hypercube. Consequently, neither the construction nor its analysis have any implications on the Gaussian setting studied here. More generally, the margin assumption intuitively captures a much more general family of distributions than the Gaussian distribution (though, formally speaking, the two assumptions are incomparable, as the Gaussian only exhibits an approximate margin property). On the positive side, [DDK$^+$23] gives an efficient algorithm for learning margin halfspaces in the presence of RCN with sample complexity $\widetilde{O}(1/(\gamma^2\epsilon^2))$. It is important to note that the homogeneity (i.e., origin-centered) assumption provably does not help for the margin case — i.e., the case of homogeneous halfspaces with a margin is as hard as the case of general halfspaces with a margin. In sharp contrast, our algorithm in this work has sample complexity that crucially depends on the unknown bias $p$ of the target halfspace — interpolating between $\widetilde{O}(d/\epsilon)$ and $\widetilde{O}(d/\epsilon^2)$.

## B   Full Version of Section 2

Throughout this paper, we frequently use the following fact. For $\mathbf{x}$ drawn from the standard normal distribution, the threshold $t$ in the definition of the halfspace can be related to the bias of $f(\mathbf{x}) = \mathrm{sign}(\mathbf{w}^* \cdot \mathbf{x} + t)$, as follows.

**Fact B.1** (Komatsu's Inequality). *For any $t \in \mathbb{R}$, the bias $p$ of a halfspace described by $f(\mathbf{x}) = \mathrm{sign}(\mathbf{w}^* \cdot \mathbf{x} + t)$ can be bounded as:*

$$\sqrt{\frac{2}{\pi}}\frac{\exp(-t^2/2)}{t + \sqrt{t^2 + 4}} \leq p \leq \sqrt{\frac{2}{\pi}}\frac{\exp(-t^2/2)}{t + \sqrt{t^2 + 2}}.$$

We begin by stating our main result, in Theorem 1.3 below, and providing a high-level summary of the main algorithm, in Algorithm 3. We note that both the assumption that $t \geq 0$ and that $t \leq \sqrt{2\log((1 - 2\eta)/\epsilon)}$ are without loss of generality. For the former, it is by the simple symmetry of the standard normal distribution that the entire argument translates into the case $t < 0$, possibly by exchanging the meaning of '+1' and '-1' labels. For the latter, we note that when the bias is small, i.e., for $p \leq \epsilon/(2(1 - 2\eta))$, a constant hypothesis suffices. Thus, only the cases covered by Theorem 1.3 are of interest to us. For the rest of the section, we assume that $\eta$ is known a priori; we show in Appendix B.3 an efficient way to estimate it without increasing the sample complexity.

**Theorem 1.3.** (Main Algorithmic Result) *There exists an algorithm that, given $\epsilon, \delta \in (0, 1/2)$ and $N$ samples from a distribution $\mathcal{D}$ satisfying Definition 1.1, runs in time $O(dN/\epsilon^2)$ and returns a hypothesis $h \in \mathcal{C}$ such that with probability at least $1 - \delta$, it holds $\mathrm{err}_{0-1}^{\mathcal{D}}(h) \leq \eta + \epsilon$. The sample complexity of the algorithm is $N = \widetilde{O}\big(\frac{d}{(1-2\eta)\epsilon} + \frac{d}{\max(p(1-2\eta),\epsilon)^2}\big)\log(1/\delta)$.*

**Remark B.2.** The sample complexity stated in Theorem 1.3 has two components: the first depends on $\epsilon$, and the second depends on the bias $p$. As we demonstrate in Section 3, the $1/p^2$ term is required for computationally efficient SQ algorithms and low-degree tests. Moreover, the term $\widetilde{O}(d\log(1/\delta)/((1 - 2\eta)\epsilon))$ is information-theoretically optimal, even when $p = 1/2$, as shown, e.g., in [HY15].

At a high level, our main algorithm consists of three main subroutines, as summarized in Algorithm 3. The subroutines are specified in the rest of the section, where we analyze the sample complexity and the runtime of our algorithm, and, as a consequence, prove Theorem 1.3. In more detail, the

---

**Algorithm 3** Main Algorithm

---

1: **Input:** $\delta$, $\eta$, $\epsilon$, sample access to distribution $\mathcal{D}$
2: $[\mathbf{w}_0, \hat{p}] = \text{Initialization}(\delta, \eta, \epsilon)$; $\epsilon' = \epsilon/(1 - 2\eta)$
3: $t_0 = \sqrt{2\log(1/\hat{p})}$, $M = 8\lceil \frac{\sqrt{2(\log(4/\hat{p}))} - \sqrt{2\log(1/\hat{p})}}{(\epsilon')^2} \rceil + 1$
4: Draw $N_2 = O(\frac{d\log(1/\delta)\log(1/\epsilon')}{(1-2\eta)^2\epsilon'})$ samples $\{(\mathbf{x}^{(i)}, y^{(i)})\}_{i=1}^{N_2}$ from $\mathcal{D}$
5: **for** $m = 1 : M$ **do**
6:     $t_m = t_0 + (m-1)\frac{(\epsilon')^2}{8}$, $\gamma_m = \frac{\epsilon'}{2}\exp(t_m^2/2)$
7:     $\widehat{\mathbf{w}}_m = \text{Optimization}(\mathbf{w}_0, t_m, \gamma_m, \eta, \{(\mathbf{x}^{(i)}, y^{(i)})\}_{i=1}^{N_2})$
8: **end for**
9: $[\widehat{\mathbf{w}}_{\text{out}}, t_{\text{out}}] = \text{Testing}((\widehat{\mathbf{w}}_1, t_1), (\widehat{\mathbf{w}}_2, t_2), \ldots, (\widehat{\mathbf{w}}_M, t_M))$
10: **return** $\widehat{\mathbf{w}}_{\text{out}}, t_{\text{out}}$

---

Initialization subroutine (specified in Algorithm 4) ensures that we can choose a vector $\mathbf{w}_0$ from the unit sphere that forms a sufficiently small angle with the target vector $\mathbf{w}^*$, using $\widetilde{O}(d\log(1/\delta)/((1 - 2\eta)^2 p^2))$ samples. This can be seen as warm-start for the main optimization procedure (specified in Algorithm 5). Crucially, the Initialization procedure does not require knowing the bias $p$ of the target halfspace; instead, it estimates this parameter to a constant factor.

The Optimization procedure is run for different guesses of the threshold $t$ ($O(\log(1/\epsilon)/\epsilon^2)$ of them). At a high level, it can be seen as a variant of Riemannian (sub)gradient descent on the unit sphere, applied to the empirical LeakyReLU loss — defined as $\text{LeakyReLU}_\lambda(u) = (1 - \lambda)u\mathbb{1}\{u \geq 0\} + \lambda u\mathbb{1}\{u < 0\}$ — with parameter $\lambda$ set to $\eta$, $u = \mathbf{w} \cdot \mathbf{x}$, and *with samples restricted to a band*, namely $a < |\mathbf{w} \cdot \mathbf{x}| < b$ — with $a$ and $b$ chosen as functions of the guess for the threshold $t$. The band restriction is key in avoiding $d/\epsilon^2$ dependence in the sample complexity, and instead only requiring order-$(d/\epsilon)$ samples to be drawn for the empirical LeakyReLU loss subgradient estimate. For a sufficiently accurate estimate $\hat{t}$ of $t$ (which is satisfied by at least one of the guesses for which the Optimization procedure is run), we argue that there is a sufficiently negative correlation between the empirical subgradient and the target weight vector $\mathbf{w}^*$. This result, combined with the Initialization result, then enables us to inductively argue that the distance between the weight vector constructed by the Optimization procedure and the target vector $\mathbf{w}^*$ contracts and becomes smaller than $\epsilon$ within order-$\log(1/\epsilon)$ iterations. This result is quite surprising, since the LeakyReLU loss is nonsmooth (it is, in fact, piecewise linear) and we do not explicitly bound its growth outside the set of its minima (i.e., we do not prove a local error bound, which would typically be used to prove linear convergence). Thus, the result as ours is impossible using black-box results for nonsmooth optimization. Additionally, we never even explicitly use the LeakyReLU loss function or prove that it is minimized by $\mathbf{w}^*$; instead, we directly prove that the vectors $\mathbf{w}$ constructed by our procedure converge to the target vector $\mathbf{w}^*$. On a technical level, our result is enabled by a novel inductive argument, which we believe may be of independent interest.

Finally, since each run of the Optimization subroutine returns a different hypothesis, with only the one(s) with the guess of $t$ being sufficiently accurate satisfying our theoretical guarantee, we need a principled approach to selecting a hypothesis with the target error guarantee. This is achieved in the Testing procedure, which simply draws a fresh sample and selects a hypothesis with the lowest test error. By a standard result due to [MN06], such a hypothesis satisfies our target error guarantee.

### B.1 Initialization Procedure

We begin this section with Lemma 2.1, which shows that for any $\kappa > 0$, given $N_1 = \widetilde{O}(d/(\kappa^4 p^2(1 - 2\eta)^2)\log(1/\delta))$ i.i.d. samples from $\mathcal{D}$, we can construct a good initial point $\mathbf{w}_0$ that forms an angle at most $\kappa$ with the target weight vector $\mathbf{w}^*$. For our purposes, $\kappa$ should be of the order $\frac{1}{t}$. For $t \leq 2\sqrt{\log(1/\epsilon')}$, where $\epsilon' = \epsilon/(1 - 2\eta)$, we can ensure that $N_1 = \widetilde{O}(d/(p^2(1 - 2\eta)^2)\log(1/\delta))$. The downside of the lemma, however, is that the number of samples $N_1$ requires at least approximate knowledge of the bias parameter $p$ (or, more accurately, of $e^{-t^2/2}$). We address this challenge by arguing (in Lemma 2.2) that we can estimate $p$ using the procedure described in Algorithm 4, without increasing the total number of drawn samples by a factor larger than order-$\log(1/\epsilon')$.

---

**Algorithm 4** Initialization

---

1: **Input:** $\delta, \eta, \epsilon > 0$
2: Let $j = 0$
3: **repeat**
4:     $j \leftarrow j + 1$
5:     $p_j \leftarrow 1/2^j$
6:     Draw $n_j = \lceil \frac{32\pi d(\log(1/\delta) + \log\log((1-2\eta)/\epsilon))}{(1-2\eta)^2 p_j^2} \rceil$ new samples from $\mathcal{D}$
7:     $\mathbf{u}_j \leftarrow \frac{1}{n_j} \sum_{i=1}^{n_j} y^{(i)} \mathbf{x}^{(i)}$
8: **until** $\|\mathbf{u}_j\|_2 \geq \frac{3}{4} \sqrt{\frac{2}{\pi}} (1 - 2\eta) p_j$ **or** $(1 - 2\eta) p_j \leq \epsilon$
9: Let $\hat{p} = 2 p_j$, $\kappa = 1/(5\sqrt{2(\log(4) + \log(1/\hat{p}))})$
10: Draw $N_1 = \lceil \frac{64\pi d \log(2/\delta)}{\kappa^4 (1-2\eta)^2 \hat{p}^2} \rceil$ new samples from $\mathcal{D}$
11: $\mathbf{u} \leftarrow \frac{1}{N_1} \sum_{i=1}^{N_1} \mathbf{x}^{(i)} y^{(i)}$
12: $\mathbf{w}_0 \leftarrow \mathbf{u}/\|\mathbf{u}\|_2$
13: **return** $\mathbf{w}_0, \hat{p}$

---

**Lemma 2.1** (Initialization via Chow Parameters). *Given $\kappa > 0$, define $p_t = e^{-t^2/2}$, $N_1 = O(d/(\kappa^4 p_t^2 (1-2\eta)^2)) \log(1/\delta))$ and let $(\mathbf{x}^{(i)}, y^{(i)})$ for $i \in [N_1]$ be i.i.d. samples drawn from $\mathcal{D}$. Let $\mathbf{u} = \frac{1}{N_1} \sum_{i=1}^{N_1} \mathbf{x}^{(i)} y^{(i)}$ and $\mathbf{w}_0 = \mathbf{u}/\|\mathbf{u}\|_2$. Then, with probability $1 - \delta$, we have $\theta(\mathbf{w}_0, \mathbf{w}^*) \leq \kappa$.*

*Proof.* We start by showing that $\mathbf{E}_{(\mathbf{x},y)\sim\mathcal{D}}[y\mathbf{x}]$ is parallel to $\mathbf{w}^*$ and has nontrivial magnitude, and then draw our conclusions from there. In particular, we prove that

$$\mathbf{E}_{(\mathbf{x},y)\sim\mathcal{D}}[y\mathbf{x}] = \sqrt{\frac{2}{\pi}}(1 - 2\eta) p_t \mathbf{w}^*. \tag{1}$$

To do so, observe first that for any vector $\mathbf{v}$ in the orthogonal complement of $\mathbf{w}^*$ (i.e., such that $\mathbf{v} \cdot \mathbf{w}^* = 0$), $\mathbf{x} \cdot \mathbf{v}$ (projection of $\mathbf{x}$ onto $\mathbf{v}$) is independent of $\mathbf{x} \cdot \mathbf{w}^*$, as $\mathbf{x}$ is drawn from $\mathcal{N}$. Thus, $\mathbf{E}_{(\mathbf{x},y)\sim\mathcal{D}}[y\mathbf{x}] \cdot \mathbf{v} = 0$, which means that $\mathbf{E}_{(\mathbf{x},y)\sim\mathcal{D}}[y\mathbf{x}]$ is either a zero vector or parallel to $\mathbf{w}^*$. To determine the magnitude of this vector, we next look at the projection of $\mathbf{E}_{(\mathbf{x},y)\sim\mathcal{D}}[y\mathbf{x}]$ onto $\mathbf{w}^*$. Using that $\mathbf{w}^* \cdot \mathbf{x}$ is a one-dimensional standard normal random variable, we have

$$\begin{aligned}
\mathbf{E}_{\mathbf{x}\sim\mathcal{N}}[\text{sign}(\mathbf{w}^* \cdot \mathbf{x} + t)\mathbf{w}^* \cdot \mathbf{x}] &= \mathbf{E}_{z\sim\mathcal{N}}[\mathbb{1}\{z \geq -t\}z - \mathbb{1}\{z < -t\}z] \\
&= \mathbf{E}_{z\sim\mathcal{N}}[\mathbb{1}\{-t < z < t\}z] + 2\mathbf{E}_{z\sim\mathcal{N}}[\mathbb{1}\{z \geq t\}z] \\
&= 2\mathbf{E}_{z\sim\mathcal{N}}[\mathbb{1}\{z \geq t\}z] \\
&= \frac{2}{\sqrt{2\pi}} \int_t^{+\infty} z e^{-z^2/2} \mathrm{d}z = -\frac{2}{\sqrt{2\pi}} \int_t^{+\infty} \mathrm{d}\left(e^{-z^2/2}\right) \\
&= \sqrt{\frac{2}{\pi}} e^{-t^2/2} = \sqrt{\frac{2}{\pi}} p_t.
\end{aligned}$$

Recalling the definition of $y$, we now obtain Equation (1). We then conclude from Equation (1) that $\|\mathbf{E}_{(\mathbf{x},y)\sim\mathcal{D}}[y\mathbf{x}]\|_2 = \sqrt{\frac{2}{\pi}}(1 - 2\eta) p_t$, as $\|\mathbf{w}^*\|_2 = 1$, by assumption.

The next step is to show that $\mathbf{u} = \frac{1}{N_1} \sum_{i=1}^{N_1} y^{(i)} \mathbf{x}^{(i)}$ concentrates near its expectation. Note that $y\mathbf{x}$ is $\sqrt{2}$-sub-Gaussian as $\mathbf{x}$ is standard normal, therefore, by the multiplicative Hoeffding bound,

$$\begin{aligned}
\mathbf{Pr}\left[ \left\| \frac{1}{N_1} \sum_{i=1}^{N_1} y^{(i)} \mathbf{x}^{(i)} - \mathbf{E}_{(\mathbf{x},y)\sim\mathcal{D}}[y\mathbf{x}] \right\|_2 \geq \frac{\kappa^2}{8} \|\mathbf{E}_{(\mathbf{x},y)\sim\mathcal{D}}[y\mathbf{x}]\|_2 \right] &\leq 2\exp\left( -\frac{N_1 \kappa^4 \|\mathbf{E}_{(\mathbf{x},y)\sim\mathcal{D}}[y\mathbf{x}]\|_2^2}{128 d} \right) \\
&\leq 2\exp\left( -\frac{N_1 \kappa^4 (1-2\eta)^2 p_t^2}{64\pi d} \right).
\end{aligned}$$

Thus, choosing $N_1 = \lceil \frac{64\pi d \log(2/\delta)}{\kappa^4 (1-2\eta)^2 p_t^2} \rceil$ suffices to guarantee that $\|\mathbf{u} - \mathbf{E}_{(\mathbf{x},y)\sim\mathcal{D}}[y\mathbf{x}]\|_2 \leq \frac{\kappa^2}{8} \|\mathbf{E}_{(\mathbf{x},y)\sim\mathcal{D}}[y\mathbf{x}]\|_2$, with probability at least $1 - \delta$.

It remains to bound $\theta(\mathbf{w}_0, \mathbf{w}^*)$, where $\mathbf{w}_0 = \mathbf{u}/\|\mathbf{u}\|_2$. Since $\mathbf{w}^*$ is a unit vector, we have that with probability at least $1 - \delta$,

$$
\cos\theta(\mathbf{w}_0, \mathbf{w}^*) = \frac{\mathbf{u} \cdot \mathbf{w}^*}{\|\mathbf{u}\|_2} \geq \frac{(\mathbf{u} - \mathbf{E}_{(\mathbf{x},y)\sim\mathcal{D}}[y\mathbf{x}]) \cdot \mathbf{w}^* + \mathbf{E}_{(\mathbf{x},y)\sim\mathcal{D}}[y\mathbf{x}] \cdot \mathbf{w}^*}{\|\mathbf{u} - \mathbf{E}_{(\mathbf{x},y)\sim\mathcal{D}}[y\mathbf{x}]\|_2 + \|\mathbf{E}_{(\mathbf{x},y)\sim\mathcal{D}}[y\mathbf{x}]\|_2}
$$

$$
\geq \frac{(1 - \kappa^2/8)\|\mathbf{E}_{(\mathbf{x},y)\sim\mathcal{D}}[y\mathbf{x}]\|_2}{(1 + \kappa^2/8)\|\mathbf{E}_{(\mathbf{x},y)\sim\mathcal{D}}[y\mathbf{x}]\|_2} = 1 - \frac{\kappa^2/4}{1 + \kappa^2/8},
$$

where we have used that $\mathbf{E}_{(\mathbf{x},y)\sim\mathcal{D}}[y\mathbf{x}] \cdot \mathbf{w}^* = \|\mathbf{E}_{(\mathbf{x},y)\sim\mathcal{D}}[y\mathbf{x}]\|_2$ (by Equation (1)) and $(\mathbf{u} - \mathbf{E}_{(\mathbf{x},y)\sim\mathcal{D}}[y\mathbf{x}]) \cdot \mathbf{w}^* \geq -\sup_{\mathbf{w}:\|\mathbf{w}\|_2=1}(\mathbf{u} - \mathbf{E}_{(\mathbf{x},y)\sim\mathcal{D}}[y\mathbf{x}]) \cdot \mathbf{w}$, which is greater than or equal to $-\frac{\kappa^2}{8}\|\mathbf{E}_{(\mathbf{x},y)\sim\mathcal{D}}[y\mathbf{x}]\|_2$, by the concentration argument used above.

As $\cos\theta(\mathbf{w}_0, \mathbf{w}^*) \leq 1 - \theta(\mathbf{w}_0, \mathbf{w}^*)^2/4$, we further have

$$
\theta(\mathbf{w}_0, \mathbf{w}^*)^2/4 \leq \frac{\kappa^2/4}{1 + \kappa^2/8} \leq \kappa^2/4,
$$

yielding the desired result that $\theta(\mathbf{w}_0, \mathbf{w}^*) \leq \kappa$. $\qquad\square$

We now leverage Lemma 2.1 to argue about correctness of Algorithm 4, which provides an implementable initialization procedure.

**Lemma 2.2.** *Consider the Initialization procedure described by Algorithm 4 in Appendix B.1. If $0 \leq t \leq \sqrt{2\log((1-2\eta)/\epsilon)}$, then with probability at least $1-\delta$, $\exp(-t^2/2) \leq \hat{p} \leq 4\exp(-t^2/2)$. The algorithm draws a total of $\widetilde{O}\big(\frac{d\log(1/\delta)}{\max\{(1-2\eta)p, \epsilon\}^2}\big)$ samples and ensures that $\theta(\mathbf{w}_0, \mathbf{w}^*) \leq \min\{\frac{1}{5t}, \frac{\pi}{2}\}$.*

*Proof.* Let $\epsilon' = \epsilon/(1-2\eta)$ and $J = \lceil\log_2(1/\epsilon')\rceil$. As we have shown in the proof of Lemma 2.1, vector $\mathbf{u}_j$ satisfies

$$
\mathbf{Pr}\left[\|\mathbf{u}_j - \mathbf{E}[y\mathbf{x}]\|_2 \geq \frac{1}{4}\sqrt{\frac{2}{\pi}}(1-2\eta)p_j\right] \leq 2\exp\left(-\frac{n_j(1-2\eta)^2 p_j^2}{32\pi d}\right).
$$

Since the algorithm runs for at most $J = \lceil\log_2(1/\epsilon')\rceil$ iterations, applying the union bound yields:

$$
\mathbf{Pr}\left[\|\mathbf{u}_j - \mathbf{E}[y\mathbf{x}]\|_2 \geq \frac{1}{4}\sqrt{\frac{2}{\pi}}(1-2\eta)p_j, \ \forall j = 1, \cdots, J\right] \leq 2J\exp\left(-\frac{n_j(1-2\eta)^2 p_j^2}{32\pi d}\right) \leq \delta,
$$

where we plugged in $n_j = \lceil\frac{32\pi d(\log(1/\delta)+\log\log(1/\epsilon'))}{(1-2\eta)^2 p_j^2}\rceil$. Furthermore, recall that we have proved in Lemma 2.1 that $\|\mathbf{E}[y\mathbf{x}]\|_2 = \sqrt{2/\pi}(1-2\eta)\exp(-t^2/2)$. Hence, with probability at least $1-\delta$, we have that for all $j = 1, \cdots, J$,

$$
\sqrt{\frac{2}{\pi}}(1-2\eta)(\exp(-t^2/2) - p_j/4) \leq \|\mathbf{u}_j\|_2 \leq \sqrt{\frac{2}{\pi}}(1-2\eta)(\exp(-t^2/2) + p_j/4). \qquad (2)
$$

Since we have assumed $0 \leq t \leq \sqrt{2\log(1/\epsilon')}$, it must be $\exp(-t^2/2) \geq \epsilon'$, thus the claimed bound on $\hat{p}$ holds if the algorithm loop ends because $j = \lceil\log_2(1/\epsilon')\rceil$. Consider now the case that the loop ends before reaching the upper bound on the number of iterations. Observe that when $p_j$ is still far away from $\exp(-t^2/2)$, i.e., when $p_j > 2\exp(-t^2/2)$, Equation (2) shows that with probability at least $1-\delta$, $\|\mathbf{u}_j\|_2 < (3/4)\sqrt{2/\pi}(1-2\eta)p_j$; thus, the algorithm will continue to decrease our guess $p_j$. On the other hand, if $p_j$ is already small, i.e., if $p_j \leq \exp(-t^2/2)$, Equation (2) implies that $\|\mathbf{u}_j\|_2 \geq (3/4)\sqrt{2/\pi}(1-2\eta)p_j$, reaching the repeat-until loop termination condition. As any iteration of the algorithm reduces the value of $p_j$ by a factor of 2, we conclude that the loop ends with $p_j$ that satisfies $\frac{1}{2}\exp(-t^2/2) \leq p_j \leq 2\exp(-t^2/2)$, hence the bound on $\hat{p}$ follows as $\hat{p} = 2p_j$.

The bound on the total number of samples drawn by the algorithm follows by observing that for each iteration $j$ of the repeat-until loop, $n_j = O(N_1)$, while there are $O(\log(1/\epsilon'))$ total loop iterations.

To complete the proof, it remains to note that the bound on $\hat{p}$ implies

$$
\sqrt{2\log(1/\hat{p})} \leq t \leq \sqrt{2(\log(4) + \log(1/\hat{p}))}
$$

Hence $\kappa$ selected in the algorithm satisfies $\kappa \leq \frac{1}{5t}$. Furthermore, since the algorithm runs for at least one iteration, it holds that $\hat{p} \leq 2p_1 = 1$. Thus, our choice of $\kappa$ also guarantees that $\kappa \leq 1/(5\sqrt{2\log(4)}) \leq \frac{\pi}{2}$. It remains to apply Lemma 2.1. $\qquad\square$

**Algorithm 5** Optimization

---

1: **Input:** $\mathbf{w}_0, \hat{t}, \hat{\gamma}, \eta, N_2$ i.i.d. samples $(\mathbf{x}^{(i)}, y^{(i)})$ from $\mathcal{D}$
2: $\mu_0 \leftarrow \frac{(1-4\rho)\sqrt{2\pi}}{16(1-2\eta)}; \rho \leftarrow 0.00098; P(\hat{t}, \hat{\gamma}) \leftarrow \mathbf{Pr}_{z\sim\mathcal{N}}[-\hat{t} \leq z \leq -\hat{t} + \hat{\gamma}]$
3: **for** $k = 0$ **to** $K$ **do**
4:     Let $\mathcal{E}(\mathbf{w}_k, \hat{t}) := \{\mathbf{x} : -\hat{t} \leq \mathbf{w}_k \cdot \mathbf{x} \leq -\hat{t} + \hat{\gamma}\}$
5:     Let $\mathbf{g}(\mathbf{w}_k; \mathbf{x}^{(i)}, y^{(i)}) = \frac{1}{2}((1 - 2\eta)\mathrm{sign}(\mathbf{w}_k \cdot \mathbf{x}^{(i)} + \hat{t}) - y^{(i)})\mathrm{proj}_{\mathbf{w}_k^\perp}(\mathbf{x}^{(i)})$
6:     $\widehat{\mathbf{g}}(\mathbf{w}_k) \leftarrow \frac{1}{N_2} \sum_{i=1}^{N_2} \mathbf{g}(\mathbf{w}_k; \mathbf{x}^{(i)}, y^{(i)}) \frac{\mathbb{1}\{\mathbf{x}^{(i)} \in \mathcal{E}(\mathbf{w}_k, \hat{t})\}}{P(\hat{t}, \hat{\gamma})}$
7:     $\mu_k \leftarrow \mu_{k-1}(1 - \rho)$
8:     $\mathbf{w}_{k+1} \leftarrow \frac{\mathbf{w}_k - \mu_k \widehat{\mathbf{g}}(\mathbf{w}_k)}{\|\mathbf{w}_k - \mu_k \widehat{\mathbf{g}}(\mathbf{w}_k)\|_2}$
9: **end for**
10: **return** $\mathbf{w}_{K+1}$

---

## B.2 Optimization

As discussed before, our Optimization procedure (Algorithm 5) can be seen as Riemannian subgradient descent on the unit sphere. Crucial to our analysis is the use of subgradient estimates from Line 5 and Line 6, where we condition on the event that the samples come from a thin band, defined in Line 4. Without this conditioning, the algorithm would correspond to projected subgradient descent of the LeakyReLU loss on the unit sphere. The conditioning effectively changes the landscape of the loss function being optimized, which cannot be argued anymore to even be convex, as the definition of the band depends on the weight vector $\mathbf{w}$ at which the vector $\widehat{\mathbf{g}}(\mathbf{w})$ is evaluated. Nevertheless, as we argue in this section, the optimization procedure can be carried out very efficiently, even exhibiting a linear convergence rate. To simplify the notation, in this section we denote the conditioned distribution $\mathcal{D}|_{\mathcal{E}(\mathbf{w},\hat{t})}$ by $\mathcal{D}(\mathbf{w}, \hat{t})$. We carry out the analysis assuming the estimate $\hat{t}$ is within additive $\epsilon^2$ of the true threshold value $t$; as argued before, this has to be true for at least one estimate $\hat{t}$ for which the Optimization procedure is invoked.

In the following lemma, we show that if the angle between a weight vector $\mathbf{w}$ and the target vector $\mathbf{w}^*$ is from a certain range, we can guarantee that $\mathbf{g}(\mathbf{w})$ is sufficiently negatively correlated with $\mathbf{w}^*$. This condition is then used to argue about progress of our algorithm. The upper bound on $\theta$ will hold initially, by our initialization procedure, and we will inductively argue that it holds for most iterations. The lower bound, when violated, will imply that the distance between $\mathbf{w}$ and $\mathbf{w}^*$ is small, in which case we would have converged to a sufficiently good solution $\mathbf{w}$.

**Lemma 2.3.** *Fix any* $\epsilon' \in (0, 1)$. *Suppose that* $0 \leq t \leq \sqrt{2\log(1/\epsilon')}$ *and* $\mathbf{w} \in \mathbb{R}^d$ *is such that* $\|\mathbf{w}\|_2 = 1$, *and* $\theta = \theta(\mathbf{w}, \mathbf{w}^*)$ *satisfies the inequality* $\epsilon' \exp(t^2/2) \leq \theta \leq 1/(5t)$. *If* $|\hat{t} - t| \leq \epsilon'^2/8$ *and* $\hat{\gamma} = (1/2)\epsilon' \exp(\hat{t}^2/2)$, *then* $\mathbf{E}_{(\mathbf{x},y)\sim\mathcal{D}(\mathbf{w},\hat{t})}[\mathbf{g}(\mathbf{w}; \mathbf{x}, y) \cdot \mathbf{w}^*] \leq -(1 - 2\eta)\sin\theta/(2\sqrt{2\pi})$.

*Proof.* To simplify the notation, in the following we write $\mathbf{g}(\mathbf{w}) = \mathbf{g}(\mathbf{w}; \mathbf{x}, y)$, as $(\mathbf{x}, y)$ is clear from the context. Using the definition of conditional expectations as well as the definition of $\mathcal{D}(\mathbf{w}, \hat{t})$, we have

$$\mathbf{E}_{(\mathbf{x},y)\sim\mathcal{D}(\mathbf{w},\hat{t})}[\mathbf{g}(\mathbf{w}) \cdot \mathbf{w}^*] = \mathbf{E}_{(\mathbf{x},y)\sim\mathcal{D}}[\mathbf{g}(\mathbf{w}) \cdot \mathbf{w}^* | \mathcal{E}(\mathbf{w}, \hat{t})] = \frac{\mathbf{E}_{(\mathbf{x},y)\sim\mathcal{D}}[\mathbf{g}(\mathbf{w}) \cdot \mathbf{w}^* \mathbb{1}\{\mathcal{E}(\mathbf{w}, \hat{t})\}]}{\mathbf{Pr}[\mathcal{E}(\mathbf{w}, \hat{t})]}. \quad (3)$$

We carry out the proof by bounding the numerator $\mathbf{E}_{(\mathbf{x},y)\sim\mathcal{D}}[\mathbf{g}(\mathbf{w}) \cdot \mathbf{w}^* \mathbb{1}\{\mathcal{E}(\mathbf{w}, \hat{t})\}]$. Recall that $\mathbf{E}_{(\mathbf{x},y)\sim\mathcal{D}}[y|\mathbf{x}] = (1-2\eta)\mathrm{sign}(\mathbf{w}^* \cdot \mathbf{x} + t)$. Furthermore, since the Gaussian distribution is rotationally invariant, we can assume without loss of generality that $\mathbf{w} = \mathbf{e}_1$ and $\mathbf{w}^* = \cos\theta\mathbf{e}_1 + \sin\theta\mathbf{e}_2$.

Therefore,

$$\mathop{\mathbf{E}}_{(\mathbf{x},y)\sim\mathcal{D}}[\mathbf{g}(\mathbf{w})\cdot\mathbf{w}^*\mathbb{1}\{\mathcal{E}(\mathbf{w},\hat{t})\}]$$

$$=\frac{1-2\eta}{2}\mathop{\mathbf{E}}_{(\mathbf{x},y)\sim\mathcal{D}}[(\mathrm{sign}(\mathbf{w}\cdot\mathbf{x}+\hat{t})-\mathrm{sign}(\mathbf{w}^*\cdot\mathbf{x}+t))\mathrm{proj}_{\mathbf{w}^\perp}(\mathbf{x})\cdot\mathbf{w}^*\mathbb{1}\{\mathcal{E}(\mathbf{w},\hat{t})\}]$$

$$=\frac{1-2\eta}{2}\mathop{\mathbf{E}}_{\mathbf{x}\sim\mathcal{D}_{\mathbf{x}}}\big[(\mathrm{sign}(\mathbf{x}_1+\hat{t})-\mathrm{sign}(\cos\theta\mathbf{x}_1+\sin\theta\mathbf{x}_2+t))(\mathbf{x}_2\mathbf{e}_2)$$
$$\cdot(\cos\theta\mathbf{e}_1+\sin\theta\mathbf{e}_2)\mathbb{1}\{\mathcal{E}(\mathbf{w},\hat{t})\}\big]$$

$$=(1-2\eta)\sin\theta\mathop{\mathbf{E}}_{\mathbf{x}\sim\mathcal{D}_{\mathbf{x}}}\big[\mathbb{1}\{\mathcal{E}(\mathbf{w},\hat{t}),\mathbf{x}_2\sin\theta\le-t-\mathbf{x}_1\cos\theta\}\mathbf{x}_2\big],\tag{4}$$

where in the final equality, we used the fact that under the event $\mathcal{E}(\mathbf{w},\hat{t})$, $\mathbf{x}_1=\mathbf{w}\cdot\mathbf{x}$ is greater than or equal to $-\hat{t}$ and consequently, the expression $\mathrm{sign}(\mathbf{x}_1+\hat{t})-\mathrm{sign}(\cos\theta\mathbf{x}_1+\sin\theta\mathbf{x}_2+t)$ equals 2 when $\cos\theta\mathbf{x}_1+\sin\theta\mathbf{x}_2+t$ is less than or equal to zero, and it is zero in all other cases.

Recall that under the assumptions of the lemma, $|t-\hat{t}|\le\epsilon'^2/8$. To bound the expectation in Equation (4), we consider two possible cases: $t\ge1$ and $t\le1$. The reason that we discuss these two cases is that when $t$ is large, under the condition that $-\hat{t}\le\mathbf{x}_1\le-\hat{t}+\hat{\gamma}$ and $\mathbf{x}_2\sin\theta\le-t-\mathbf{x}_1\cos\theta$, it is guaranteed that $\mathbf{x}_2\le0$. On the other hand, when $t$ is small, it is possible that $\mathbf{x}_2\ge0$. However, we can show that even though $\mathbf{x}_2\ge0$ in some area of the band, the expectation $\mathbf{E}[\mathbf{x}_2\mathbb{1}\{\mathcal{E}(\mathbf{w},\hat{t}),0\le\mathbf{x}_2\sin\theta\le-t-\mathbf{x}_1\cos\theta\}]$ is small since $t$ is very small. This is handled in the following two claims, under the same assumptions as in the statement of the lemma.

**Claim B.3.** *If $t\ge1$, then*

$$\mathop{\mathbf{E}}_{\mathbf{x}\sim\mathcal{D}_{\mathbf{x}}}[\mathbf{x}_2\mathbb{1}\{\mathcal{E}(\mathbf{w},\hat{t}),\mathbf{x}_2\sin\theta\le-t-\mathbf{x}_1\cos\theta\}]\le-\frac{2\mathbf{Pr}[\mathcal{E}(\mathbf{w},\hat{t})]}{3\sqrt{2\pi}}.$$

*Proof.* When $t\ge1$, under event $\mathcal{E}(\mathbf{w},\hat{t})$, we have

$$\mathbf{x}_2\sin\theta\le-t-\mathbf{x}_1\cos\theta\le-t+\hat{t}\cos\theta$$
$$\le-t+t\cos(\theta)+\epsilon'^2\cos(\theta)/8$$
$$\le-2\sin^2(\theta/2)t+\epsilon'^2/8.$$

Since we have assumed $t\ge1$ and $\theta\ge\epsilon'\exp(t^2/2)$, we further have

$$\mathbf{x}_2\sin\theta\le-2\sin^2(\theta/2)t+\epsilon'^2/8\le-\frac{\epsilon'^2}{8}(\exp(t^2)-1)\le0,$$

where we used that $\sin(\theta/2)\ge\theta/4$, which holds for any $\theta\le\pi$. Therefore, $\mathbf{x}_2\le0$ when $\mathbb{1}\{\mathcal{E}(\mathbf{w},\hat{t}),\mathbf{x}_2\sin\theta\le-t-\mathbf{x}_1\cos\theta\}=1$.

Note that conditioning on $\mathcal{E}(\mathbf{w},\hat{t})$ we have $\mathbf{x}_1\le-\hat{t}+\hat{\gamma}$, hence it holds $\mathbb{1}\{\mathcal{E}(\mathbf{w},\hat{t}),\mathbf{x}_2\sin\theta\le-t-\mathbf{x}_1\cos\theta\}\ge\mathbb{1}\{\mathcal{E}(\mathbf{w},\hat{t}),\mathbf{x}_2\sin\theta\le-t-(-\hat{t}+\hat{\gamma})\cos\theta\}$. In addition, since $\hat{t}\ge t-\epsilon'^2/8$, we have $\mathbb{1}\{\mathcal{E}(\mathbf{w},\hat{t}),\mathbf{x}_2\sin\theta\le-t-\mathbf{x}_1\cos\theta\}\ge\mathbb{1}\{\mathcal{E}(\mathbf{w},\hat{t}),\mathbf{x}_2\sin\theta\le-(1-\cos\theta)t-(\epsilon'^2/8+\hat{\gamma})\cos\theta\}$. Therefore, we have the following upper bound:

$$\mathop{\mathbf{E}}_{\mathbf{x}\sim\mathcal{D}_{\mathbf{x}}}[\mathbf{x}_2\mathbb{1}\{\mathcal{E}(\mathbf{w},\hat{t}),\mathbf{x}_2\sin\theta\le-t-\mathbf{x}_1\cos\theta\}]$$
$$\le\mathop{\mathbf{E}}_{\mathbf{x}\sim\mathcal{D}_{\mathbf{x}}}[\mathbf{x}_2\mathbb{1}\{\mathcal{E}(\mathbf{w},\hat{t}),\mathbf{x}_2\le-t\tan(\theta/2)-(\epsilon'^2/8+\hat{\gamma})\cot\theta\}],\tag{5}$$

where we used the trigonometric identity $(1-\cos\theta)/\sin\theta=\tan(\theta/2)$. Since $\mathbf{x}_1$ and $\mathbf{x}_2$ are independent standard normal random variables, the expectation on the right-hand side of Equation (5) has the following closed form expression:

$$\mathop{\mathbf{E}}_{\mathbf{x}\sim\mathcal{D}_{\mathbf{x}}}[\mathbf{x}_2\mathbb{1}\{\mathcal{E}(\mathbf{w},\hat{t}),\mathbf{x}_2\le-t\tan(\theta/2)-(\epsilon'^2/8+\hat{\gamma})\cot\theta\}]$$
$$=\mathbf{Pr}[\mathcal{E}(\mathbf{w},\hat{t})]\int_{-\infty}^{-t\tan(\theta/2)-(\epsilon'^2/8+\hat{\gamma})\cot\theta}\frac{x}{\sqrt{2\pi}}\exp(-x^2/2)\,\mathrm{d}x$$
$$=-\frac{\mathbf{Pr}[\mathcal{E}(\mathbf{w},\hat{t})]}{\sqrt{2\pi}}\exp\left(-\frac{1}{2}\left(t\tan\left(\frac{\theta}{2}\right)+\frac{\epsilon'^2}{8}\cot\theta+\hat{\gamma}\cot\theta\right)^2\right).\tag{6}$$

Let us now bound $t \tan\left(\frac{\theta}{2}\right) + \frac{\epsilon'^2}{8} \cot\theta + \hat{\gamma}\cot\theta$. Using the trigonometric inequalities $\tan(\theta/2) \leq \theta/2$ and $\cot\theta \leq 1/\theta$, which hold for $\theta \in [0, \pi/2]$, and recalling that $\epsilon' \exp(t^2/2) \leq \theta \leq 1/(5t)$, we have

$$
\begin{aligned}
t \tan\left(\frac{\theta}{2}\right) + \frac{\epsilon'^2}{8}\cot\theta + \hat{\gamma}\cot\theta &\leq \frac{t\theta}{2} + \frac{\epsilon'^2}{8\theta} + \frac{\hat{\gamma}}{\theta} \\
&\leq \frac{1}{10} + \frac{1}{8} + \frac{1}{2}e^{\frac{\hat{t}^2 - t^2}{2}} \\
&\leq \frac{9}{40} + \frac{1}{2}e^{\frac{\epsilon'^2}{8}}\sqrt{2\log(1/\epsilon')} \leq \frac{7}{8},
\end{aligned}
\tag{7}
$$

where the second inequality is by the definition of $\hat{\gamma}$ and the bounds on $\theta$ and the third inequality is by $\hat{t} - t \leq \frac{\epsilon'^2}{8}$ and $\hat{t}, t \leq \sqrt{2\log(1/\epsilon')}$. Hence, combining Equation (5)–Equation (7), we get

$$
\begin{aligned}
\mathop{\mathbf{E}}_{\mathbf{x} \sim \mathcal{D}_\mathbf{x}}[\mathbf{x}_2 \mathbb{1}\{\mathcal{E}(\mathbf{w}, \hat{t}), \mathbf{x}_2 \sin\theta \leq -t - \mathbf{x}_1 \cos\theta\}] &\leq -\frac{\mathbf{Pr}[\mathcal{E}(\mathbf{w}, \hat{t})]}{\sqrt{2\pi}}\exp(-(7/8)^2/2) \\
&\leq -\frac{2\mathbf{Pr}[\mathcal{E}(\mathbf{w}, \hat{t})]}{3\sqrt{2\pi}},
\end{aligned}
\tag{8}
$$

as claimed. $\qquad\square$

We now proceed to the case where $t < 1$.

**Claim B.4.** *If $t < 1$, then*

$$
\mathop{\mathbf{E}}_{\mathbf{x} \sim \mathcal{D}_\mathbf{x}}[\mathbf{x}_2 \mathbb{1}\{\mathcal{E}(\mathbf{w}, \hat{t}), \mathbf{x}_2 \sin\theta \leq -t - \mathbf{x}_1 \cos\theta\}] \leq -\frac{\mathbf{Pr}[\mathcal{E}(\mathbf{w}, \hat{t})]}{2\sqrt{2\pi}}.
$$

*Proof.* In this case, it is possible that $\mathbf{x}_2 \geq 0$ when $\mathbb{1}\{\mathcal{E}(\mathbf{w}), \mathbf{x}_2 \sin\theta \leq -t - \mathbf{x}_1 \cos\theta\} = 1$. However, since $\hat{\gamma} = \epsilon'\exp(\hat{t}^2/2)/2 \geq \epsilon'^2/8$, it must be $-\hat{t} + \hat{\gamma} \geq -t - \epsilon'^2/8 + \hat{\gamma} \geq -t$, indicating that $-t - (-\hat{t} + \hat{\gamma})\cos\theta \leq -t + t\cos\theta \leq 0$, hence $\mathbf{x}_2 \leq 0$ when $\mathbb{1}\{\mathcal{E}(\mathbf{w}, \hat{t}), \mathbf{x}_2 \leq -t/\sin\theta - (-\hat{t} + \hat{\gamma})\cot\theta\} = 1$. Therefore, we split the indicator $\mathbb{1}\{\mathcal{E}(\mathbf{w}, \hat{t}), \mathbf{x}_2 \sin\theta \leq -t - \mathbf{x}_1 \cos\theta\}$ into the indicators of three sub-events:

$$
\mathbb{1}\{\mathcal{E}(\mathbf{w}, \hat{t}), \mathbf{x}_2 \leq -t/\sin\theta - \mathbf{x}_1 \cot\theta\} = \mathbb{1}\{\mathcal{E}_1\} + \mathbb{1}\{\mathcal{E}_2\} + \mathbb{1}\{\mathcal{E}_3\},
$$

where

$$
\begin{aligned}
\mathcal{E}_1 &:= \mathcal{E}(\mathbf{w}, \hat{t}) \cap \{\mathbf{x}_2 \leq -t/\sin\theta - (-\hat{t} + \hat{\gamma})\cot\theta\} \\
\mathcal{E}_2 &:= \mathcal{E}(\mathbf{w}, \hat{t}) \cap \{0 \leq \mathbf{x}_2 \leq -t/\sin\theta - \mathbf{x}_1 \cot\theta\} \\
\mathcal{E}_3 &:= \mathcal{E}(\mathbf{w}, \hat{t}) \cap \{-t/\sin\theta - (-\hat{t} + \hat{\gamma})\cot\theta \leq \mathbf{x}_2 \leq \min\{-t/\sin\theta - \mathbf{x}_1 \cot\theta, 0\}\}.
\end{aligned}
$$

For $\mathbf{E}_{\mathbf{x} \sim \mathcal{D}_\mathbf{x}}[\mathbf{x}_2 \mathbb{1}\{\mathcal{E}_1\}]$, observe first that

$$
\begin{aligned}
\mathbb{1}\{\mathcal{E}_1\} &= \mathbb{1}\{\mathcal{E}(\mathbf{w}, \hat{t}), \mathbf{x}_2 \leq -t/\sin\theta + \hat{t}\cot\theta - \hat{\gamma}\cot\theta\} \\
&\geq \mathbb{1}\{\mathcal{E}(\mathbf{w}, \hat{t}), \mathbf{x}_2 \leq -t/\sin\theta + (t - \epsilon'^2/8)\cot\theta - \hat{\gamma}\cot\theta\}
\end{aligned}
$$

Since $\mathbf{x}_2 \leq 0$ under $\mathcal{E}_1$, it then holds

$$
\begin{aligned}
\mathop{\mathbf{E}}_{\mathbf{x} \sim \mathcal{D}_\mathbf{x}}[\mathbf{x}_2 \mathbb{1}\{\mathcal{E}_1\}] &\leq \mathop{\mathbf{E}}_{\mathbf{x} \sim \mathcal{D}_\mathbf{x}}[\mathbf{x}_2 \mathbb{1}\{\mathcal{E}(\mathbf{w}, \hat{t}), \mathbf{x}_2 \leq -t/\sin\theta + (t - \epsilon'^2/8)\cot\theta - \hat{\gamma}\cot\theta\}] \\
&= \mathop{\mathbf{E}}_{\mathbf{x} \sim \mathcal{D}_\mathbf{x}}[\mathbf{x}_2 \mathbb{1}\{\mathcal{E}(\mathbf{w}, \hat{t}), \mathbf{x}_2 \leq -t\tan(\theta/2) - \epsilon'^2/8\cot\theta - \hat{\gamma}\cot\theta\}] \\
&= -\frac{\mathbf{Pr}[\mathcal{E}(\mathbf{w}, \hat{t})]}{\sqrt{2\pi}}\exp\left(-\frac{1}{2}\left(t\tan\left(\frac{\theta}{2}\right) + \frac{\epsilon'^2}{8}\cot\theta + \hat{\gamma}\cot\theta\right)^2\right),
\end{aligned}
$$

following similar steps as in Claim B.3, Equation (5)–Equation (6). Again, note that we have assumed $\epsilon'\exp(t^2/2) \leq \theta \leq 1/(5t)$ and have chosen $\hat{\gamma} = \epsilon'\exp(t^2/2)/2$, thus, using the fact that $\tan(\theta/2) \leq \theta/2$, $\cot\theta \leq 1/\theta$, we further get:

$$
\mathop{\mathbf{E}}_{\mathbf{x} \sim \mathcal{D}_\mathbf{x}}[\mathbf{x}_2 \mathbb{1}\{\mathcal{E}_1\}] \leq -\frac{\mathbf{Pr}[\mathcal{E}(\mathbf{w}, \hat{t})]}{\sqrt{2\pi}}\exp\left(-\frac{1}{2}\left(\frac{t\theta}{2} + \frac{\epsilon'^2}{8\theta} + \frac{\hat{\gamma}}{\theta}\right)^2\right) \leq -\frac{2\mathbf{Pr}[\mathcal{E}(\mathbf{w}, \hat{t})]}{3\sqrt{2\pi}},
$$

using the same arguments as in Equation (7)–Equation (8). For $\mathbf{E}_{\mathbf{x}\sim\mathcal{D}_{\mathbf{x}}}[\mathbf{x}_2\mathbb{1}\{\mathcal{E}_3\}]$, note that $\mathbf{x}_2 \leq 0$ under $\mathcal{E}_3$, hence $\mathbf{E}_{\mathbf{x}\sim\mathcal{D}_{\mathbf{x}}}[\mathbf{x}_2\mathbb{1}\{\mathcal{E}_3\}] \leq 0$.

We now study $\mathbf{E}_{\mathbf{x}\sim\mathcal{D}_{\mathbf{x}}}[\mathbf{x}_2\mathbb{1}\{\mathcal{E}_2\}]$. Observe that

$$\mathbb{1}\{\mathcal{E}_2\} = \mathbb{1}\{\mathcal{E}(\mathbf{w},\hat{t}), 0 \leq \mathbf{x}_2 \leq -t/\sin\theta - \mathbf{x}_1\cot\theta\}$$
$$\leq \mathbb{1}\{\mathcal{E}(\mathbf{w},\hat{t}), 0 \leq \mathbf{x}_2 \leq -t/\sin\theta + \hat{t}\cot\theta\}$$
$$\leq \mathbb{1}\{\mathcal{E}(\mathbf{w},\hat{t}), 0 \leq \mathbf{x}_2 \leq |-t(1-\cos\theta)/\sin\theta + (\epsilon'^2/8)\cot\theta|\},$$

where the first inequality results from the condition that $\mathbf{x}_1 \geq -\hat{t}$. Hence, the expectation $\mathbf{E}_{\mathbf{x}\sim\mathcal{D}_{\mathbf{x}}}[\mathbf{x}_2\mathbb{1}\{\mathcal{E}_2\}]$ can be upper-bounded by

$$\mathop{\mathbf{E}}_{\mathbf{x}\sim\mathcal{D}_{\mathbf{x}}}[\mathbf{x}_2\mathbb{1}\{\mathcal{E}_2\}] \leq \mathop{\mathbf{E}}_{\mathbf{x}\sim\mathcal{D}_{\mathbf{x}}}[\mathbf{x}_2\mathbb{1}\{\mathcal{E}(\mathbf{w},\hat{t}), 0 \leq \mathbf{x}_2 \leq |-t\tan(\theta/2) + (\epsilon'^2/8)\cot\theta|\}]$$

$$= \mathbf{Pr}[\mathcal{E}(\mathbf{w},\hat{t})] \int_0^{|-t\tan(\theta/2)+(\epsilon'^2/8)\cot\theta|} \frac{x}{\sqrt{2\pi}}\exp(-x^2/2)\,\mathrm{d}x$$

$$\leq \mathbf{Pr}[\mathcal{E}(\mathbf{w},\hat{t})] \int_0^{|-t\tan(\theta/2)+(\epsilon'^2/8)\cot\theta|} \frac{x}{\sqrt{2\pi}}\,\mathrm{d}x$$

$$\leq \frac{\mathbf{Pr}[\mathcal{E}(\mathbf{w},\hat{t})]}{2\sqrt{2\pi}}(t\tan(\theta/2) - (\epsilon'^2/8)\cot\theta)^2.$$

We again use the fact that $\tan(\theta/2) \leq \theta$ and $\cot\theta \leq 1/\theta$, then recall that $\epsilon'\exp(t^2/2) \leq \theta \leq 1/(5t)$, $\epsilon'/\theta \leq 1$, thus, we get

$$\mathop{\mathbf{E}}_{\mathbf{x}\sim\mathcal{D}_{\mathbf{x}}}[\mathbf{x}_2\mathbb{1}\{\mathcal{E}_2\}] \leq \frac{\mathbf{Pr}[\mathcal{E}(\mathbf{w},\hat{t})]}{\sqrt{2\pi}}((t\theta)^2 + (\epsilon'^2/(8\theta))^2) \leq \frac{\mathbf{Pr}[\mathcal{E}(\mathbf{w},\hat{t})]}{\sqrt{2\pi}}(1/25 + \epsilon'^2/64) \leq \frac{\mathbf{Pr}[\mathcal{E}(\mathbf{w},t')]}{12\sqrt{2\pi}}.$$

Combining with the derived upper bounds on $\mathbf{E}_{\mathbf{x}\sim\mathcal{D}_{\mathbf{x}}}[\mathbf{x}_2\mathbb{1}\{\mathcal{E}_1\}]$ and $\mathbf{E}_{\mathbf{x}\sim\mathcal{D}_{\mathbf{x}}}[\mathbf{x}_2\mathbb{1}\{\mathcal{E}_3\}]$ completes the proof. $\qquad\square$

Combining Claim B.4 and Claim B.3, we have $\mathbf{E}_{\mathbf{x}\sim\mathcal{D}_{\mathbf{x}}}[\mathbf{x}_2\mathbb{1}\{\mathcal{E}(\mathbf{w},\hat{t}), \mathbf{x}_2\sin\theta \leq -t - \mathbf{x}_1\cos\theta\}] \leq -\frac{\mathbf{Pr}[\mathcal{E}(\mathbf{w},\hat{t})]}{2\sqrt{2\pi}}$. Thus, plugging this result back to Equation (4) and then combining with Equation (3), we complete the proof of the lemma. $\qquad\square$

Since, by construction, $\mathbf{g}(\mathbf{w})$ is orthogonal to $\mathbf{w}$ (see Line 5 in Algorithm 5), we can bound the norm of the expected gradient vector by bounding $\mathbf{g}(\mathbf{w}) \cdot \mathbf{u}$ for some unit vectors $\mathbf{u}$ that are orthogonal to $\mathbf{w}$ using similar techniques as in Lemma 2.3. To be specific, we have the following lemma.

**Lemma 2.4.** *Under the assumptions of Lemma 2.3,* $\left\|\mathbf{E}_{(\mathbf{x},y)\sim\mathcal{D}(\mathbf{w},\hat{t})}[\mathbf{g}(\mathbf{w};\mathbf{x},y)]\right\|_2 \leq \frac{(1-2\eta)}{\sqrt{2\pi}}$.

*Proof.* First, we show that for any vector that is orthogonal to both $\mathbf{w}$ and $\mathbf{w}^*$, the expected gradient of $\mathbf{g}(\mathbf{w}) = \mathbf{g}(\mathbf{w};\mathbf{x},y)$ is zero. As a consequence, $\mathbf{E}[\mathbf{g}(\mathbf{w})]$ must lie in the 2-dimensional space spanned by $\mathbf{w}$ and $\mathbf{w}^*$. To see that, observe first that

$$\mathop{\mathbf{E}}_{(\mathbf{x},y)\sim\mathcal{D}(\mathbf{w},\hat{t})}[\mathbf{g}(\mathbf{w})\cdot\mathbf{v}] = \frac{1-2\eta}{\mathbf{Pr}[\mathcal{E}(\mathbf{w},\hat{t})]}\mathop{\mathbf{E}}_{\mathbf{x}\sim\mathcal{D}_{\mathbf{x}}}[\mathbb{1}\{\mathcal{E}(\mathbf{w},\hat{t}), \mathbf{w}^*\cdot\mathbf{x}+t \leq 0\}\mathbf{v}\cdot\mathrm{proj}_{\mathbf{w}^\perp}(\mathbf{x})]$$

$$= \frac{1-2\eta}{\mathbf{Pr}[\mathcal{E}(\mathbf{w},\hat{t})]}\mathop{\mathbf{E}}_{\mathbf{x}\sim\mathcal{D}_{\mathbf{x}}}[\mathbb{1}\{\mathcal{E}(\mathbf{w},\hat{t}), \mathbf{w}^*\cdot\mathbf{x}+t \leq 0\}]\mathop{\mathbf{E}}_{\mathbf{x}\sim\mathcal{D}_{\mathbf{x}}}[\mathbf{v}\cdot\mathbf{x}] = 0,$$

where we used $\mathrm{proj}_{\mathbf{w}^\perp}(\mathbf{x}) = \mathbf{x} - (\mathbf{x}\cdot\mathbf{w})\mathbf{w}$ and $\mathbf{v}\cdot\mathbf{w} = 0$, the fact that $\mathbf{v}\cdot\mathbf{x}$ is independent of $\mathbf{w}\cdot\mathbf{x}$ and $\mathbf{w}^*\cdot\mathbf{x}$, and that $\mathbf{E}_{\mathbf{x}\sim\mathcal{D}_{\mathbf{x}}}[\mathbf{v}\cdot\mathbf{x}] = 0$. Furthermore, by construction, $\mathbf{g}(\mathbf{w})$ is orthogonal to $\mathbf{w}$. This indicates that $\mathbf{E}_{(\mathbf{x},y)\sim\mathcal{D}}[\mathbf{g}(\mathbf{w})]$ is parallel to $\mathbf{w}^*$, as both $\mathbf{w}$ and $\mathbf{w}^*$ are unit vectors. Since $\mathbf{x}\sim\mathcal{D}_{\mathbf{x}}$ is rotation invariant, we can assume $\mathbf{w} = \mathbf{e}_1$ and $\mathbf{w}^* = \cos\theta\mathbf{e}_1 + \sin\theta\mathbf{e}_2$. We thus only need to bound $|\mathbf{E}_{(\mathbf{x},y)\sim\mathcal{D}(\mathbf{w},\hat{t})}[\mathbf{g}(\mathbf{w})]\cdot\mathbf{e}_2|$. Recall that $\mathcal{D}(\mathbf{w},\hat{t})$ is the distribution conditioned on the band $\mathcal{E}(\mathbf{w},\hat{t}) = \{\mathbf{x} : -\hat{t} \leq \mathbf{w}\cdot\mathbf{x} \leq -\hat{t}+\hat{\gamma}\}$; hence, by the definition of $\mathbf{g}(\mathbf{w})$, we have

$$\left|\mathop{\mathbf{E}}_{(\mathbf{x},y)\sim\mathcal{D}(\mathbf{w},\hat{t})}[\mathbf{g}(\mathbf{w})\cdot\mathbf{e}_2]\right| = \frac{|\mathbf{E}_{(\mathbf{x},y)\sim\mathcal{D}}[\mathbf{g}(\mathbf{w})\cdot\mathbf{e}_2\mathbb{1}\{\mathcal{E}(\mathbf{w},\hat{t})\}]|}{\mathbf{Pr}[\mathcal{E}(\mathbf{w},\hat{t})]}$$

$$= \frac{(1-2\eta)}{\mathbf{Pr}[\mathcal{E}(\mathbf{w},\hat{t})]}\left|\mathop{\mathbf{E}}_{\mathbf{x}\sim\mathcal{D}_{\mathbf{x}}}[\mathbf{x}_2\mathbb{1}\{\mathcal{E}(\mathbf{w},\hat{t}), \sin\theta\mathbf{x}_2 \leq -t - \cos\theta\mathbf{x}_1\}]\right|. \quad (9)$$

To proceed, we discuss the cases where $t \leq 1$ and $t \geq 1$, following similar steps as in Lemma 2.3.

**Claim B.5.** *Under the assumptions of Lemma 2.3, if $t \geq 1$, then*

$$\left\| \mathop{\mathbf{E}}_{(\mathbf{x},y) \sim \mathcal{D}(\mathbf{w},\hat{t})} [\mathbf{g}(\mathbf{w}; \mathbf{x}, y)] \right\|_2 \leq \frac{1 - 2\eta}{\sqrt{2\pi}}.$$

*Proof.* As shown in the proof of Claim B.3, when $t \geq 1$ the condition $\mathbf{x}_2 \leq -t/\sin\theta - \mathbf{x}_1 \cot\theta$, $-\hat{t} \leq \mathbf{x}_1 \leq -\hat{t} + \hat{\gamma}$ implies that $\mathbf{x}_2 \leq 0$. Hence, in this case we have

$$\mathbb{1}\{\mathcal{E}(\mathbf{w}, \hat{t}), \mathbf{x}_2 \leq -t/\sin\theta - \mathbf{x}_1 \cot\theta\} \leq \mathbb{1}\{\mathcal{E}(\mathbf{w}, \hat{t}), \mathbf{x}_2 \leq 0\},$$

and we can further conclude that

$$\left| \mathop{\mathbf{E}}_{\mathbf{x} \sim \mathcal{D}_\mathbf{x}} [\mathbf{x}_2 \mathbb{1}\{\mathcal{E}(\mathbf{w}, \hat{t}), \, \mathbf{x}_2 \sin\theta \leq -t - \mathbf{x}_1 \cos\theta\}] \right| = \mathop{\mathbf{E}}_{\mathbf{x} \sim \mathcal{D}_\mathbf{x}} [-\mathbf{x}_2 \mathbb{1}\{\mathcal{E}(\mathbf{w}, \hat{t}), \mathbf{x}_2 \sin\theta \leq -t - \mathbf{x}_1 \cos\theta\}]$$

$$\leq \mathop{\mathbf{E}}_{\mathbf{x} \sim \mathcal{D}_\mathbf{x}} [-\mathbf{x}_2 \mathbb{1}\{\mathcal{E}(\mathbf{w}, \hat{t}), \mathbf{x}_2 \leq 0\}]$$

$$= \frac{\mathbf{Pr}[\mathcal{E}(\mathbf{w}, \hat{t})]}{\sqrt{2\pi}}.$$

Plugging this back into Equation (9) yields the claimed result. □

When $t \leq 1$, we use a slightly different decomposition

$$\mathbb{1}\{\mathcal{E}(\mathbf{w}, \hat{t}), \, \mathbf{x}_2 \leq -\hat{t}/\sin\theta - \mathbf{x}_1 \cot\theta\} = \mathbb{1}\{\mathcal{E}_1'\} + \mathbb{1}\{\mathcal{E}_2'\},$$

where

$$\mathcal{E}_1' = \mathcal{E}(\mathbf{w}, \hat{t}) \cap \{\mathbf{x}_2 \leq -t/\sin\theta - \mathbf{x}_1 \cot\theta, \mathbf{x}_2 \leq 0\},$$

$$\mathcal{E}_2' = \mathcal{E}(\mathbf{w}, \hat{t}) \cap \{0 \leq \mathbf{x}_2 \leq -t/\sin\theta - \mathbf{x}_1 \cot\theta\}.$$

By the definitions of these two events, we have

$$\mathop{\mathbf{E}}_{\mathcal{D}_\mathbf{x}} \left[ \mathbf{x}_2 \mathbb{1}\{\mathcal{E}_1'\} \right] \leq 0, \quad \mathop{\mathbf{E}}_{\mathcal{D}_\mathbf{x}} \left[ \mathbf{x}_2 \mathbb{1}\{\mathcal{E}_2'\} \right] \geq 0.$$

Since in the proof of Lemma 2.3 we have shown that

$$\mathop{\mathbf{E}}_{\mathcal{D}_\mathbf{x}} [\mathbf{x}_2 \mathbb{1}\{\mathcal{E}(\mathbf{w}, \hat{t}), \, \mathbf{x}_2 \leq -\hat{t}/\sin\theta - \mathbf{x}_1 \cot\theta\}] \leq 0,$$

it must hold that

$$\left| \mathop{\mathbf{E}}_{\mathbf{x} \sim \mathcal{D}_\mathbf{x}} [\mathbf{x}_2 \mathbb{1}\{\mathcal{E}(\mathbf{w}, \hat{t}), \mathbf{x}_2 \leq -\hat{t}/\sin\theta - \mathbf{x}_1 \cot\theta\}] \right| = \left| \mathop{\mathbf{E}}_{\mathbf{x} \sim \mathcal{D}_\mathbf{x}} [\mathbf{x}_2 \mathbb{1}\{\mathcal{E}_1'\}] + \mathop{\mathbf{E}}_{\mathbf{x} \sim \mathcal{D}_\mathbf{x}} [\mathbf{x}_2 \mathbb{1}\{\mathcal{E}_2'\}] \right|$$

$$\leq - \mathop{\mathbf{E}}_{\mathbf{x} \sim \mathcal{D}_\mathbf{x}} [\mathbf{x}_2 \mathbb{1}\{\mathcal{E}_1'\}].$$

Since $\mathbb{1}\{\mathcal{E}_1'\} \leq \mathbb{1}\{\mathcal{E}(\mathbf{w}, \hat{t}), \mathbf{x}_2 \leq 0\}$, we thus have

$$\left| \mathop{\mathbf{E}}_{\mathbf{x} \sim \mathcal{D}_\mathbf{x}} [\mathbf{x}_2 \mathbb{1}\{\mathcal{E}(\mathbf{w}, \hat{t}), \mathbf{x}_2 \leq -\hat{t}/\sin\theta - \mathbf{x}_1 \cot\theta\}] \right| \leq \frac{\mathbf{Pr}[\mathcal{E}(\mathbf{w}, \hat{t})]}{\sqrt{2\pi}}.$$

Plugging this back into Equation (9) completes the proof. □

The following lemma establishes a uniform convergence result for the empirical subgradient.

**Lemma B.6.** *Consider the learning problem defined in Definition 1.1. Fix $\epsilon', \delta \in (0, 1/2)$, and let $\alpha$ be any absolute constant in $(0, 1)$. Consider the following class of functions for $(\mathbf{x}, y) \sim \mathcal{D}$ :*

$$\mathcal{F} = \{\mathbf{g}'(\mathbf{w}) : \mathbf{w} \in \mathbb{R}^d, \|\mathbf{w}\|_2 = 1, \epsilon' \exp\left(t^2/2\right) \leq \theta(\mathbf{w}, \mathbf{w}^*) \leq 1/(5t)\}, \quad \text{where}$$

$$\mathbf{g}'(\mathbf{w}; \mathbf{x}, y) = \frac{1}{2}\left((1 - 2\eta)\mathrm{sign}(\mathbf{w} \cdot \mathbf{x} + \hat{t}) - y\right)\mathrm{proj}_{\mathbf{w}^\perp}(\mathbf{x}) \mathbb{1}\{\mathbf{w} \cdot \mathbf{x} \in [-\hat{t}, -\hat{t} + \hat{\gamma}]\},$$

*and where $\hat{t}$ satisfies $|\hat{t} - t| \leq \epsilon'^2/8$ and $\hat{\gamma} = (1/2)\epsilon' \exp(\hat{t}^2/2)$. Then, using $N = \widetilde{O}(\frac{d \log(1/(\delta))}{(1-2\eta)^2 \epsilon'})$ samples from $\mathcal{D}$ with probability at least $1 - \delta$, for any $\mathbf{g}'(\mathbf{w}, \hat{t}) \in \mathcal{F}$ it holds*

$$\left\| \frac{1}{N} \sum_{i=1}^N \mathbf{g}'(\mathbf{w}; \mathbf{x}^{(i)}, y^{(i)}) - \mathop{\mathbf{E}}_{(\mathbf{x},y) \sim \mathcal{D}} [\mathbf{g}'(\mathbf{w}; \mathbf{x}, y)] \right\|_2 \leq \alpha \left\| \mathop{\mathbf{E}}_{(\mathbf{x},y) \sim \mathcal{D}} [\mathbf{g}'(\mathbf{w}; \mathbf{x}, y)] \right\|_2 .$$

*Proof.* For simplicity, we will use $\mathbf{g}'(\mathbf{w})$ to denote $\mathbf{g}'(\mathbf{w}; \mathbf{x}, y)$ and we further define

$$\widehat{\mathbf{g}}'(\mathbf{w}) := \frac{1}{N} \sum_{i=1}^{N} \mathbf{g}'(\mathbf{w}; \mathbf{x}^{(i)}, y^{(i)}).$$

By definition, $\mathbf{g}'(\mathbf{w})$ is orthogonal to $\mathbf{w}$, hence so is $\widehat{\mathbf{g}}'(\mathbf{w})$. As already argued in the proof of Lemma 2.4, $\mathbf{E}_{(\mathbf{x},y)\sim\mathcal{D}}[\mathbf{g}'(\mathbf{w})]$ is also orthogonal to $\mathbf{w}$. Thus, $\|\widehat{\mathbf{g}}'(\mathbf{w}) - \mathbf{E}_{(\mathbf{x},y)\sim\mathcal{D}}[\mathbf{g}'(\mathbf{w})]\|_2$ is determined by $(\widehat{\mathbf{g}}'(\mathbf{w}) - \mathbf{E}_{(\mathbf{x},y)\sim\mathcal{D}}[\mathbf{g}'(\mathbf{w})]) \cdot \mathbf{w}'$, where $\mathbf{w}'$ is a unit vector that is orthogonal to $\mathbf{w}$.

Fix unit vectors $\mathbf{w}, \mathbf{w}' \in \mathbb{R}^d$ with $\mathbf{w} \cdot \mathbf{w}' = 0$. We are going to make use of the following variant of Bernstein's inequality (see, e.g., [God55]).

**Fact B.7.** *Let $X_1, \dots, X_N$ be zero mean i.i.d. random variables. Assume that for some positive reals $L, \sigma > 0$, it holds that $\mathbf{E}[|X_i|^k] \leq (1/2)\sigma^2 L^{k-2} k!$. Then, for any $x \in (0, \sqrt{N\sigma^2}/(2L))$,*

$$\mathbf{Pr}\left[\left|\sum_{i=1}^{N} X_i\right| \geq 2x\sqrt{N\sigma^2}\right] \leq \exp(-x^2).$$

We show that the random variable $\mathbf{g}'(\mathbf{w}) \cdot \mathbf{w}'$ satisfies the assumptions of Fact B.7. For any $k \geq 2$, using the definition of $\mathbf{g}'$ which enforces $\mathbf{g}'(\mathbf{w}; \mathbf{x}, y) = 0$ whenever $\mathbb{1}\{\mathbf{w} \cdot \mathbf{x} \in [-\hat{t}, -\hat{t}+\hat{\gamma}]\} = 0$, we have that

$$\mathop{\mathbf{E}}_{(\mathbf{x},y)\sim\mathcal{D}}[|\mathbf{g}'(\mathbf{w}) \cdot \mathbf{w}'|^k] \leq \mathop{\mathbf{E}}_{\mathbf{x}\sim\mathcal{D}_{\mathbf{x}}}[\mathbb{1}\{\mathbf{w} \cdot \mathbf{x} \in [-\hat{t}, -\hat{t}+\hat{\gamma}]\}] \mathop{\mathbf{E}}_{\mathbf{x}\sim\mathcal{D}_{\mathbf{x}}}[|\mathbf{w}' \cdot \mathbf{x}|^k]$$

$$\leq \mathbf{Pr}[\mathbf{w} \cdot \mathbf{x} \in [-\hat{t}, -\hat{t}+\hat{\gamma}]] C^{k-2} k!,$$

where the last inequality comes from the fact that for a $\sqrt{2}$-sub-Gaussian variable $z$ (e.g., a Gaussian random variable), it holds $\mathbf{E}[|z|^k] \leq (\sqrt{2}e\sqrt{k})^k \leq C_1 C_2^{k-2} k!$ for some absolute constants $C_1$ and $C_2$, and we choose $C$ to be a large enough multiple of $C_1, C_2$. Let $\sigma^2 = \mathbf{Pr}[\mathbf{w} \cdot \mathbf{x} \in [-\hat{t}, -\hat{t}+\hat{\gamma}]]$. Note that since $\hat{\gamma} = \frac{1}{2}\epsilon' \exp(\hat{t}^2/2)$, we have $\sigma^2 \geq \hat{\gamma}\exp(-\hat{t}^2/2) = \frac{1}{2}\epsilon'$. Then, the condition of Fact B.7 is satisfied with $\sigma^2 \geq \frac{1}{2}\epsilon'$.

Next, we show that we can bound $\|\mathbf{E}_{(\mathbf{x},y)\sim\mathcal{D}}[\mathbf{g}'(\mathbf{w})]\|_2$ from below. In Lemma 2.3 we showed that when $\epsilon' \exp(t^2/2) \leq \theta(\mathbf{w}, \mathbf{w}^*) \leq 1/(5t)$, $|\hat{t} - t| \leq \epsilon'^2/8$ and $\hat{\gamma} = \frac{1}{2}\epsilon' \exp(\hat{t}^2/2)$, it holds

$$\mathop{\mathbf{E}}_{(\mathbf{x},y)\sim\mathcal{D}(\mathbf{w},\hat{t})}[\mathbf{g}(\mathbf{w}) \cdot \mathbf{w}^*] = \frac{\mathbf{E}_{(\mathbf{x},y)\sim\mathcal{D}}[\mathbf{g}(\mathbf{w}) \cdot \mathbf{w}^* \mathbb{1}\{\mathbf{w} \cdot \mathbf{x} \in [-\hat{t}, -\hat{t}+\hat{\gamma}]\}]}{\mathbf{Pr}[\mathbf{w} \cdot \mathbf{x} \in [-\hat{t}, -\hat{t}+\hat{\gamma}]]}$$

$$= \frac{\mathbf{E}_{(\mathbf{x},y)\sim\mathcal{D}}[\mathbf{g}'(\mathbf{w}) \cdot \mathbf{w}^*\}]}{\sigma^2} \leq -\frac{(1-2\eta)\sin(\theta(\mathbf{w}, \mathbf{w}^*))}{2\sqrt{2\pi}}. \quad (10)$$

Let $\tilde{\mathbf{w}}$ be a unit vector orthogonal to $\mathbf{w}$ such that $\mathbf{w}^* = \mathbf{w}\cos(\theta(\mathbf{w}, \mathbf{w}^*)) + \tilde{\mathbf{w}}\sin(\theta(\mathbf{w}, \mathbf{w}^*))$. Then, recalling the fact that $\mathbf{E}_{(\mathbf{x},y)\sim\mathcal{D}}[\mathbf{g}'(\mathbf{w})]$ is also orthogonal to $\mathbf{w}$, we have:

$$\|\mathop{\mathbf{E}}_{(\mathbf{x},y)\sim\mathcal{D}}[\mathbf{g}'(\mathbf{w})]\|_2 = \sup_{\mathbf{v}\in\mathbb{R}^d : \|\mathbf{v}\|_2=1} \mathop{\mathbf{E}}_{(\mathbf{x},y)\sim\mathcal{D}}[\mathbf{g}'(\mathbf{w})] \cdot \mathbf{v}$$

$$\geq -\mathop{\mathbf{E}}_{(\mathbf{x},y)\sim\mathcal{D}}[\mathbf{g}'(\mathbf{w})] \cdot \tilde{\mathbf{w}}$$

$$= -\frac{\mathbf{E}_{(\mathbf{x},y)\sim\mathcal{D}}[\mathbf{g}'(\mathbf{w})] \cdot \mathbf{w}^*}{\sin\theta(\mathbf{w}, \mathbf{w}^*)}$$

$$\geq \frac{1-2\eta}{2\sqrt{2\pi}},$$

where in the last line we used Equation (10). Thus, as $\sigma^2 \leq 1$ (it is defined as a probability), we conclude that

$$\|\mathop{\mathbf{E}}_{(\mathbf{x},y)\sim\mathcal{D}}[\mathbf{g}'(\mathbf{w})]\|_2 \geq \frac{(1-2\eta)}{2\sqrt{2\pi}}\sigma^2. \quad (11)$$

Applying Fact B.7, we now get that for any $\mathbf{w}' \in \mathbb{R}^d$, $\|\mathbf{w}'\|_2 = 1$,

$$\mathbf{Pr}\left[\left|\widehat{\mathbf{g}}'(\mathbf{w}) \cdot \mathbf{w}' - \mathop{\mathbf{E}}_{(\mathbf{x},y)\sim\mathcal{D}}[\mathbf{g}'(\mathbf{w}) \cdot \mathbf{w}']\right| \geq 2x\sqrt{\sigma^2/N}\right] \leq \exp(-x^2).$$

Choosing $x = \frac{\alpha(1-2\eta)}{2\sqrt{2\pi}}\sqrt{\sigma^2 N}$, where $\alpha \in (0,1)$ is an absolute constant, yields

$$\mathbf{Pr}\left[\left|\widehat{\mathbf{g}}'(\mathbf{w}) \cdot \mathbf{w}' - \mathop{\mathbf{E}}_{(\mathbf{x},y)\sim\mathcal{D}}[\mathbf{g}'(\mathbf{w}) \cdot \mathbf{w}']\right| \geq \frac{\alpha(1-2\eta)}{2\sqrt{2\pi}}\sigma^2\right] \leq \exp(-N\alpha^2(1-2\eta)^2\sigma^2/(8\pi)) \ .$$

In particular, since the above inequality holds for any unit $\mathbf{w}'$, using Equation (11), it follows that

$$\mathbf{Pr}\left[\left|\widehat{\mathbf{g}}'(\mathbf{w}) \cdot \mathbf{w}' - \mathop{\mathbf{E}}_{(\mathbf{x},y)\sim\mathcal{D}}[\mathbf{g}'(\mathbf{w}) \cdot \mathbf{w}']\right| \geq \alpha\|\mathop{\mathbf{E}}_{(\mathbf{x},y)\sim\mathcal{D}}[\mathbf{g}'(\mathbf{w})]\|_2\right]$$
$$\leq \exp(-N\alpha^2(1-2\eta)^2\sigma^2/(8\pi)) \ . \tag{12}$$

It remains to show that Equation (12) holds for all functions in $\mathcal{F}$. To do that, we apply the union bound along all the directions $\mathbf{w}'$, all the hypothesis $\mathbf{w}$. A cover for these parameters will be of order $(1/\epsilon')^{O(d)}$. Hence, we have that for any unit vector $\mathbf{w} \in \mathbb{R}^d$ with $\epsilon'\exp(t^2/2) \leq \theta(\mathbf{w}, \mathbf{w}^*) \leq 1/(5t)$,

$$\mathbf{Pr}\left[\|\widehat{\mathbf{g}}'(\mathbf{w}) - \mathop{\mathbf{E}}_{(\mathbf{x},y)\sim\mathcal{D}}[\mathbf{g}'(\mathbf{w})]\|_2 \geq \alpha\|\mathop{\mathbf{E}}_{(\mathbf{x},y)\sim\mathcal{D}}[\mathbf{g}'(\mathbf{w})]\|_2\right]$$
$$\leq \exp(O(d\log(1/\epsilon')))\exp(-N\alpha^2(1-2\eta)^2\sigma^2/(8\pi)) \leq \delta \ ,$$

where in the last inequality, we used that $\sigma^2 \geq \frac{1}{2}\epsilon'$, and $N \geq \widetilde{O}(d\log(1/\delta)/(\epsilon'(1-2\eta)^2))$. $\qquad\square$

Recall that for some fixed $\mathbf{w}$ and $\hat{t}$, we have defined the empirical gradient vector in Line 6 as:

$$\widehat{\mathbf{g}}(\mathbf{w}) = \frac{1}{N_2 P(\hat{t}, \hat{\gamma})}\sum_{i=1}^{N_2}\mathbf{g}(\mathbf{w}; \mathbf{x}^{(i)}, y^{(i)})\mathbb{1}\{\mathbf{x}^{(i)} \in \mathcal{E}(\mathbf{w}, \hat{t})\},$$

where $P(\hat{t}, \hat{\gamma}) = \mathbf{Pr}_{z\sim\mathcal{N}}[z \in [-\hat{t}, -\hat{t}+\hat{\gamma}]] = \mathbf{Pr}[\mathcal{E}(\mathbf{w}, \hat{t})]$, since $\mathbf{w}$ is a unit vector and $\mathbf{w} \cdot \mathbf{x}$ follows standard Gaussian. Thus, $\widehat{\mathbf{g}}(\mathbf{w}) = \widehat{\mathbf{g}}'(\mathbf{w})/\mathbf{Pr}[\mathcal{E}(\mathbf{w}, \hat{t})]$. In addition, by definition we know that

$$\mathop{\mathbf{E}}_{(\mathbf{x},y)\sim\mathcal{D}(\mathbf{w},\hat{t})}[\mathbf{g}(\mathbf{w})] = \mathop{\mathbf{E}}_{(\mathbf{x},y)\sim\mathcal{D}}[\mathbf{g}(\mathbf{w})]/\mathbf{Pr}[\mathcal{E}(\mathbf{w}, \hat{t})],$$

and so Lemma B.6 immediately implies the following corollary.

**Corollary B.8.** *Consider the learning problem from Definition 1.1. Let $\epsilon', \delta, \hat{t}, \hat{\gamma}$ be parameters satisfying the condition of Lemma B.6 and choose $\alpha = 1/4$. Then using $\widetilde{O}(d\log(1/(\delta))/((1-2\eta)^2\epsilon'))$ samples to construct $\widehat{\mathbf{g}}$, for any unit vector $\mathbf{w}$ such that $\epsilon'\exp(t^2/2) \leq \theta(\mathbf{w}, \mathbf{w}^*) \leq 1/(5t)$, it holds with probability at least $1-\delta$: $\|\widehat{\mathbf{g}}(\mathbf{w}) - \mathbf{E}_{(\mathbf{x},y)\sim\mathcal{D}(\mathbf{w},\hat{t})}[\mathbf{g}(\mathbf{w})]\|_2 \leq (1/4)\|\mathbf{E}_{(\mathbf{x},y)\sim\mathcal{D}(\mathbf{w},\hat{t})}[\mathbf{g}(\mathbf{w})]\|_2$.*

We are now ready to present and prove our main algorithm-related result. A short roadmap for our proof is as follows. Since Algorithm 3 constructs a grid with grid-width $\epsilon'^2/8$ that covers all possible values of the true threshold $t$, there exists at least one guess $\hat{t}$ that is $\epsilon'^2$-close to the true threshold $t$. We first show that to get a halfspace with error at most $\epsilon'$, it suffices to use this $\hat{t}$ as the threshold and find a weight vector $\mathbf{w}$ such that the angle $\theta(\mathbf{w}, \mathbf{w}^*)$ is of the order $\epsilon'$, which is exactly what Algorithm 5 does. The connection between $\theta(\mathbf{w}, \mathbf{w}^*)$ and the error is conveyed by the following fact:

**Fact B.9** (see, e.g., Lemma 4.2 of [DKS18]). *Under the standard normal distribution, it holds:*

$$\mathbf{Pr}[\mathrm{sign}(\mathbf{w} \cdot \mathbf{x} + t) \neq \mathrm{sign}(\mathbf{w}^* \cdot \mathbf{x} + t)] \leq \frac{\theta(\mathbf{w}, \mathbf{w}^*)}{\pi}\exp(-t^2/2).$$

Let $\mathbf{w}_k$ be the parameter generated by Algorithm 5 at iteration $k$ for threshold $\hat{t}$. We show that $\theta(\mathbf{w}_k, \mathbf{w}^*)$ converges to zero at a linear rate. To this end, we prove that under our carefully devised step size $\mu_k$, there exists an upper bound on $\|\mathbf{w}_k - \mathbf{w}^*\|_2$, which contracts at each iteration. Note that since both $\mathbf{w}_k$ and $\mathbf{w}^*$ are on the unit sphere, we have $\|\mathbf{w}_k - \mathbf{w}^*\|_2 = 2\sin(\theta(\mathbf{w}_k, \mathbf{w}^*)/2)$. Essentially, this implies that Algorithm 5 produces a sequence of parameters $\mathbf{w}_k$ such that $\theta(\mathbf{w}_k, \mathbf{w}^*)$ converges to 0 linearly, under this threshold $\hat{t}$. Thus, we can conclude that there exists a halfspace among all halfspaces generated by Algorithm 3 that achieves $\epsilon'$ error with high probability.

**Theorem 2.6.** *Consider the learning problem from Definition 1.1. Fix any unit vector $\mathbf{w}_0 \in \mathbb{R}^d$ such that $\theta(\mathbf{w}_0, \mathbf{w}^*) \leq \min(1/(5t), \pi/2)$. Fix any $\epsilon, \delta > 0$. Let $\hat{t} > 0$ be a threshold such that $|\hat{t} - t| \leq \epsilon^2/(8(1 - 2\eta)^2)$, and let $\hat{\gamma} = \epsilon/(2(1 - 2\eta)) \exp(\hat{t}^2/2)$. Then Algorithm 2 uses $N_2 = \widetilde{O}\big(d/((1 - 2\eta)\epsilon) \log(1/\delta)\big)$ samples from $\mathcal{D}$, has runtime $\widetilde{O}(N_2 d)$, and outputs a weight vector $\mathbf{w}$ such that $h(\mathbf{x}) = \mathrm{sign}(\mathbf{w} \cdot \mathbf{x} + \hat{t})$ satisfies $\mathbf{Pr}[h(\mathbf{x}) \neq y] \leq \eta + \epsilon$ with probability at least $1 - \delta$.*

*Proof.* Let $\epsilon' = \frac{\epsilon}{1 - 2\eta}$, and denote $\mathbf{w}_k$ as the parameter produced by the algorithm at $k^{\text{th}}$ iteration under threshold $\hat{t}$. Observe that for any unit vector $\mathbf{w}$:

$$\mathbf{Pr}[\mathrm{sign}(\mathbf{w} \cdot \mathbf{x} + \hat{t}) \neq \mathrm{sign}(\mathbf{w}^* \cdot \mathbf{x} + t)]$$
$$= \mathbf{Pr}[\mathrm{sign}(\mathbf{w} \cdot \mathbf{x} + \hat{t}) \neq \mathrm{sign}(\mathbf{w} \cdot \mathbf{x} + t), \mathrm{sign}(\mathbf{w} \cdot \mathbf{x} + t) = \mathrm{sign}(\mathbf{w}^* \cdot \mathbf{x} + t)]$$
$$+ \mathbf{Pr}[\mathrm{sign}(\mathbf{w} \cdot \mathbf{x} + \hat{t}) = \mathrm{sign}(\mathbf{w} \cdot \mathbf{x} + t), \mathrm{sign}(\mathbf{w} \cdot \mathbf{x} + t) \neq \mathrm{sign}(\mathbf{w}^* \cdot \mathbf{x} + t)]$$
$$\leq \mathbf{Pr}[\mathrm{sign}(\mathbf{w} \cdot \mathbf{x} + \hat{t}) \neq \mathrm{sign}(\mathbf{w} \cdot \mathbf{x} + t)] + \mathbf{Pr}[\mathrm{sign}(\mathbf{w} \cdot \mathbf{x} + t) \neq \mathrm{sign}(\mathbf{w}^* \cdot \mathbf{x} + t)].$$

Since $|\hat{t} - t| \leq \epsilon'^2/8$, it holds

$$\mathbf{Pr}[\mathrm{sign}(\mathbf{w} \cdot \mathbf{x} + \hat{t}) \neq \mathrm{sign}(\mathbf{w} \cdot \mathbf{x} + t)] = \mathbf{Pr}[-\max\{t, \hat{t}\} \leq \mathbf{w} \cdot \mathbf{x} \leq -\min\{t, \hat{t}\}]$$
$$\leq \frac{2|\hat{t} - t|}{\sqrt{2\pi}} \leq \frac{\epsilon'^2}{4\sqrt{2\pi}}.$$

In addition, as shown in Fact B.9, $\mathbf{Pr}[\mathrm{sign}(\mathbf{w} \cdot \mathbf{x} + t) \neq \mathrm{sign}(\mathbf{w}^* \cdot \mathbf{x} + t)] \leq \frac{\theta(\mathbf{w}, \mathbf{w}^*)}{\pi} \exp(-t^2/2)$; thus,

$$\mathbf{Pr}[\mathrm{sign}(\mathbf{w} \cdot \mathbf{x} + \hat{t}) \neq \mathrm{sign}(\mathbf{w}^* \cdot \mathbf{x} + t)] \leq \frac{\epsilon'^2}{4\sqrt{2\pi}} + \frac{\theta(\mathbf{w}, \mathbf{w}^*)}{\pi} \exp(-t^2/2). \quad (13)$$

Therefore, it suffices to find a parameter $\mathbf{w}$ such that $\theta(\mathbf{w}, \mathbf{w}^*) \leq \pi\epsilon' \exp(t^2/2)$. Note that since both $\mathbf{w}$ and $\mathbf{w}^*$ are unit vectors, we have $\|\mathbf{w} - \mathbf{w}^*\|_2 = 2\sin(\theta/2)$, indicating that it suffices to minimize $\|\mathbf{w} - \mathbf{w}^*\|_2$ efficiently. As proved in Lemma 2.1 and Lemma 2.2, we can start with an initial vector $\mathbf{w}_0$ such that $\theta(\mathbf{w}_0, \mathbf{w}^*) \leq 1/(5t)$ by calling Algorithm 4. Starting from this $\mathbf{w}_0$, we show that $\|\mathbf{w}_k - \mathbf{w}^*\|_2$ contracts linearly whenever the angle between $\mathbf{w}_k$ and $\mathbf{w}^*$ is larger than $\epsilon' \exp(t^2/2)$, thus we reach the required upper bound for this angle within a logarithmic number of steps. Denote $\theta_k = \theta(\mathbf{w}_k, \mathbf{w}^*)$ and consider the case when $\theta_k \geq \epsilon' \exp(t^2/2)$.

**Claim 2.7.** *Let $C_1 := (1 - 2\eta)/\sqrt{2\pi}$. Drawing $N_2 = \widetilde{O}(d \log(1/\delta)/((1 - 2\eta)^2 \epsilon'))$ samples from distribution $\mathcal{D}$, we have that if $\theta_k \geq \epsilon' \exp(t^2/2)$ then with probability at least $1 - \delta$: $\|\mathbf{w}_{k+1} - \mathbf{w}^*\|_2^2 \leq \|\mathbf{w} - \mathbf{w}^*\|_2^2 - (C_1/2)\mu_k \sin\theta_k + 4C_1^2\mu_k^2$.*

*Proof.* Observe first that since $\widehat{\mathbf{g}}(\mathbf{w}_k)$ is orthogonal to $\mathbf{w}_k$, we have $\|\mathbf{w}_k - \mu_k\widehat{\mathbf{g}}(\mathbf{w}_k)\|_2^2 = \|\mathbf{w}_k\|_2^2 + \mu_k^2\|\widehat{\mathbf{g}}(\mathbf{w}_k)\|_2^2 \geq 1$, thus normalizing $\mathbf{w}_k - \mu\widehat{\mathbf{g}}(\mathbf{w}_k)$ is equivalent to projecting $\mathbf{w}_k - \mu_k\widehat{\mathbf{g}}(\mathbf{w}_k)$ to the unit ball $\mathbb{B}$. Since we have assumed $\mathbf{w}^* \in \mathbb{B}$, by the non-expansiveness of the projection operator and $\mathrm{proj}_{\mathbb{B}}(\mathbf{w}^*) = \mathbf{w}^*$, we have:

$$\|\mathbf{w}_{k+1} - \mathbf{w}^*\|_2^2 = \left\|\frac{\mathbf{w}_k - \mu_k\widehat{\mathbf{g}}(\mathbf{w}_k)}{\|\mathbf{w}_k - \mu_k\widehat{\mathbf{g}}(\mathbf{w}_k)\|_2} - \mathbf{w}^*\right\|_2^2 = \|\mathrm{proj}_{\mathbb{B}}(\mathbf{w}_k - \mu_k\widehat{\mathbf{g}}(\mathbf{w}_k)) - \mathbf{w}^*\|_2^2$$
$$\leq \|\mathbf{w}_k - \mu_k\widehat{\mathbf{g}}(\mathbf{w}_k) - \mathbf{w}^*\|_2^2.$$

Thus, expanding the squared norm on the right-hand side yields:

$$\|\mathbf{w}_{k+1} - \mathbf{w}^*\|_2^2 \leq \|\mathbf{w}_k - \mathbf{w}^*\|_2^2 - 2\mu_k\widehat{\mathbf{g}}(\mathbf{w}_k) \cdot (\mathbf{w}_k - \mathbf{w}^*) + \mu_k^2\|\widehat{\mathbf{g}}(\mathbf{w}_k)\|_2^2$$
$$= \|\mathbf{w}_k - \mathbf{w}^*\|_2^2 + 2\mu_k \mathop{\mathbf{E}}_{(\mathbf{x},y) \sim \mathcal{D}(\mathbf{w}_k, \hat{t})}[\mathbf{g}(\mathbf{w}_k) \cdot \mathbf{w}^*] \quad (14)$$
$$+ 2\mu_k\Big(\widehat{\mathbf{g}}(\mathbf{w}_k) - \mathop{\mathbf{E}}_{(\mathbf{x},y) \sim \mathcal{D}(\mathbf{w}_k, \hat{t})}[\mathbf{g}(\mathbf{w}_k)]\Big) \cdot \mathbf{w}^* + \mu_k^2\|\widehat{\mathbf{g}}(\mathbf{w}_k)\|_2^2$$

where in the first equality we used the fact that $\widehat{\mathbf{g}}(\mathbf{w}_k)$ and $\mathbf{E}_{(\mathbf{x},y) \sim \mathcal{D}(\mathbf{w}_k, \hat{t})}[\mathbf{g}(\mathbf{w}_k)]$ are both orthogonal to $\mathbf{w}_k$. Without loss of generality (because of the rotational invariance), assume $\mathbf{w}_k = \mathbf{e}_1$ and

$\mathbf{w}^* = \cos\theta\mathbf{e}_1 + \sin\theta\mathbf{e}_2$. Then, again by the fact that both $\widehat{\mathbf{g}}(\mathbf{w}_k)$ and $\mathbf{E}_{(\mathbf{x},y)\sim\mathcal{D}(\mathbf{w}_k,\hat{t})}[\mathbf{g}(\mathbf{w}_k)]$ are orthogonal to $\mathbf{w}_k$, we have

$$
\begin{aligned}
(\widehat{\mathbf{g}}(\mathbf{w}_k) - \mathop{\mathbf{E}}_{(\mathbf{x},y)\sim\mathcal{D}(\mathbf{w}_k,\hat{t})}[\mathbf{g}(\mathbf{w}_k)]) \cdot \mathbf{w}^* &= \sin\theta_k(\widehat{\mathbf{g}}(\mathbf{w}_k) - \mathop{\mathbf{E}}_{(\mathbf{x},y)\sim\mathcal{D}(\mathbf{w}_k,\hat{t})}[\mathbf{g}(\mathbf{w}_k)]) \cdot \mathbf{e}_2 \\
&\leq \sin\theta_k \big\|\widehat{\mathbf{g}}(\mathbf{w}_k) - \mathop{\mathbf{E}}_{(\mathbf{x},y)\sim\mathcal{D}(\mathbf{w}_k,\hat{t})}[\mathbf{g}(\mathbf{w}_k)]\big\|_2.
\end{aligned}
$$

Thus, further invoking Lemma 2.3, we have:

$$
\begin{aligned}
\|\mathbf{w}_{k+1} - \mathbf{w}^*\|_2^2 \leq \|\mathbf{w}_k - \mathbf{w}^*\|_2^2 &- 2\mu_k \frac{(1-2\eta)\sin\theta_k}{2\sqrt{2\pi}} \\
&+ 2\mu_k \sin\theta_k \big\|\widehat{\mathbf{g}}(\mathbf{w}_k) - \mathop{\mathbf{E}}_{(\mathbf{x},y)\sim\mathcal{D}(\mathbf{w}_k,\hat{t})}[\mathbf{g}(\mathbf{w}_k)]\big\|_2 + \mu_k^2\|\widehat{\mathbf{g}}(\mathbf{w}_k)\|_2^2. \quad (15)
\end{aligned}
$$

Corollary B.8 (or Lemma 2.5) implies that with $N_2 = \widetilde{O}(d\log(1/\delta)/((1-2\eta)^2\epsilon'))$ samples in total, for any unit vector $\mathbf{w}_k$ satisfying $\epsilon'\exp(t^2/2) \leq \theta_k \leq 1/(5t)$ with probability at least $1-\delta$, it holds:

$$
\big\|\widehat{\mathbf{g}}(\mathbf{w}_k) - \mathop{\mathbf{E}}_{(\mathbf{x},y)\sim\mathcal{D}(\mathbf{w}_k,\hat{t})}[\mathbf{g}(\mathbf{w}_k)]\big\|_2 \leq \frac{1}{4}\big\|\mathop{\mathbf{E}}_{(\mathbf{x},y)\sim\mathcal{D}(\mathbf{w}_k,\hat{t})}[\mathbf{g}(\mathbf{w}_k)]\big\|_2. \quad (16)
$$

Recall that we have shown in the proof of Lemma 2.4 that $\|\mathbf{E}_{(\mathbf{x},y)\sim\mathcal{D}(\mathbf{w}_k,\hat{t})}[\mathbf{g}(\mathbf{w}_k)]\|_2 \leq \frac{1-2\eta}{\sqrt{2\pi}}$; therefore, Equation (16) further gives that with probability at least $1-\delta$:

$$
\begin{aligned}
\|\widehat{\mathbf{g}}(\mathbf{w}_k)\|_2 &\leq \big\|\widehat{\mathbf{g}}(\mathbf{w}_k) - \mathop{\mathbf{E}}_{(\mathbf{x},y)\sim\mathcal{D}(\mathbf{w}_k,\hat{t})}[\mathbf{g}(\mathbf{w}_k)]\big\|_2 + \big\|\mathop{\mathbf{E}}_{(\mathbf{x},y)\sim\mathcal{D}(\mathbf{w}_k,\hat{t})}[\mathbf{g}(\mathbf{w}_k)]\big\|_2 \\
&\leq \frac{5}{4}\big\|\mathop{\mathbf{E}}_{(\mathbf{x},y)\sim\mathcal{D}(\mathbf{w}_k,\hat{t})}[\mathbf{g}(\mathbf{w}_k)]\big\|_2 \leq \frac{2(1-2\eta)}{\sqrt{2\pi}}. \quad (17)
\end{aligned}
$$

Thus, plugging Equation (16) and Equation (17) back into Equation (15), we get that with probability at least $1-\delta$,

$$
\begin{aligned}
\|\mathbf{w}_{k+1} - \mathbf{w}^*\|_2^2 &\leq \|\mathbf{w}_k - \mathbf{w}^*\|_2^2 - 2\mu_k\frac{1-2\eta}{2\sqrt{2\pi}}\sin\theta_k + 2\mu_k\frac{1-2\eta}{4\sqrt{2\pi}}\sin\theta_k + \mu_k^2\frac{2(1-2\eta)^2}{\pi} \\
&\leq \|\mathbf{w}_k - \mathbf{w}^*\|_2^2 - \mu_k\frac{1-2\eta}{2\sqrt{2\pi}}\sin\theta_k + \mu_k^2\frac{2(1-2\eta)^2}{\pi}. \quad (18)
\end{aligned}
$$

Let $C_1 := \frac{1-2\eta}{\sqrt{2\pi}}$. Then Equation (18) is simplified to:

$$
\|\mathbf{w}_{k+1} - \mathbf{w}^*\|_2^2 \leq \|\mathbf{w} - \mathbf{w}^*\|_2^2 - \frac{C_1}{2}\mu_k\sin\theta_k + 4C_1^2\mu_k^2, \quad (19)
$$

completing the proof of this claim. $\qquad\square$

It remains to choose the step size $\mu_k$ properly to get linear convergence. By carefully designing a shrinking step size, we are able to construct an upper-bound $\phi_k$ on the distance of $\|\mathbf{w}_{k+1} - \mathbf{w}_k\|_2$ using Claim 2.7. Importantly, by exploiting the property that both $\mathbf{w}$ and $\mathbf{w}^*$ are on the unit sphere, we show that the upper bound is contracting at each step, even though the distance $\|\mathbf{w}_{k+1} - \mathbf{w}_k\|_2$ could be increasing. Concretely, we have the following claim.

**Lemma 2.8.** *Let $\rho = 0.00098$ and $\phi_k = (1-\rho)^k$. Then, setting $\mu_k = (1-4\rho)\phi_k/(16C_1)$ it holds $\sin(\theta_k/2) \leq \phi_k$ for $k = 1, \cdots, K$.*

*Proof.* Let $\phi_k = (1-\rho)^k$ where $\rho = 0.00098$. This choice of $\rho$ ensures that $32\rho^2 + 1020\rho - 1 \leq 0$. We show by induction that choosing $\mu_k = (1-4\rho)\phi_k/(16C_1) = (1-\rho)^k(1-4\rho)/(16C_1)$, it holds $\sin(\theta_k/2) \leq \phi_k$. The condition certainly holds for $k = 1$ since $\theta_1 \in [0, \pi/2]$. Now suppose that $\sin(\theta_k/2) \leq \phi_k$ for some $k \geq 1$. We discuss the following 2 cases: $\phi_k \geq \sin(\theta_k/2) \geq \frac{3}{4}\phi_k$ and $\sin(\theta_k/2) \leq \frac{3}{4}\phi_k$.

First, suppose $\phi_k \geq \sin(\theta_k/2) \geq \frac{3}{4}\phi_k$. Since $\sin(\theta_k/2) \leq \sin\theta_k$, it also holds $\sin\theta_k \geq \frac{3}{4}\phi_k$. Bringing in the fact that $\|\mathbf{w}_{k+1} - \mathbf{w}^*\|_2 = 2\sin(\theta_{k+1}/2)$ and $\|\mathbf{w}_k - \mathbf{w}^*\|_2 = 2\sin(\theta_k/2)$, as well as the definition of $\mu_k$, Equation (19) becomes:

$$
\begin{aligned}
(2\sin(\theta_{k+1}/2))^2 &\leq (2\sin(\theta_k/2))^2 - \frac{C_1}{2}\mu_k \sin\theta_k + 4C_1^2 \frac{(1-4\rho)}{16C_1}\phi_k\mu_k \\
&\leq 4\phi_k^2 - \frac{C_1}{2}\mu_k \frac{3}{4}\phi_k + \frac{C_1(1-4\rho)}{4}\mu_k\phi_k \\
&= 4\phi_k^2 - \frac{C_1}{4}\left(\frac{1}{2} + 4\rho\right)\frac{1-4\rho}{16C_1}\phi_k^2 \\
&= 4\phi_k^2\left(1 - \frac{(1+8\rho)(1-4\rho)}{512}\right),
\end{aligned}
$$

where in the second line we used $\sin\theta_k \geq \frac{3}{4}\phi_k$ and in the third line we used the definition of $\mu_k$ by which $\mu_k = (1-4\rho)\phi_k/(16C_1)$. Since $\rho$ is chosen so that $32\rho^2 + 1020\rho - 1 \leq 0$, we have:

$$
\begin{aligned}
\sin(\theta_{k+1}/2) &\leq \phi_k\sqrt{1 - \frac{(1+8\rho)(1-4\rho)}{512}} \\
&\leq \phi_k\left(1 - \frac{(1+8\rho)(1-4\rho)}{1024}\right) \leq (1-\rho)\phi_k = (1-\rho)^{k+1},
\end{aligned}
$$

as desired.

Next, consider $\sin(\theta_k/2) \leq \frac{3}{4}\phi_k$. Recall that $\mathbf{w}_{k+1} = \text{proj}_{\mathbb{B}}(\mathbf{w}_k - \mu_k\widehat{\mathbf{g}}(\mathbf{w}_k))$ and $\mathbf{w}_k \in \mathbb{B}$; therefore, $\|\mathbf{w}_{k+1} - \mathbf{w}_k\|_2 \leq \|\mathbf{w}_k - \mu_k\widehat{\mathbf{g}}(\mathbf{w}) - \mathbf{w}_k\|_2 = \mu_k\|\widehat{\mathbf{g}}(\mathbf{w}_k)\|_2$ by the non-expansiveness of the projection operator. Furthermore, applying Equation (17), we have $\|\widehat{\mathbf{g}}(\mathbf{w}_k)\|_2 \leq 2C_1$; therefore, $\|\mathbf{w}_{k+1} - \mathbf{w}_k\|_2 \leq 2\mu_k C_1$, which indicates that:

$$
2(\sin(\theta_{k+1}/2) - \sin(\theta_k/2)) = \|\mathbf{w}_{k+1} - \mathbf{w}^*\|_2 - \|\mathbf{w}_k - \mathbf{w}^*\|_2 \leq \|\mathbf{w}_{k+1} - \mathbf{w}_k\|_2 \leq 2\mu_k C_1.
$$

Since we have assumed $\sin(\theta_k/2) \leq \frac{3}{4}\phi_k$, then it holds:

$$
\begin{aligned}
\phi_{k+1} - \sin(\theta_{k+1}/2) &\geq (1-\rho)\phi_k - \phi_k + \phi_k - \sin(\theta_k/2) - \mu_k C_1 \\
&\geq -\rho\phi_k + \frac{1}{4}\phi_k - \frac{1-4\rho}{16}\phi_k = \frac{3(1-4\rho)}{16}\phi_k > 0,
\end{aligned}
$$

since we have chosen $\mu_k = (1-4\rho)\phi_k/(16C_1)$. Hence, it also holds that $\sin(\theta_{k+1}/2) \leq \phi_{k+1}$. $\qquad\square$

Lemma 2.8 shows that $\sin(\theta_k/2)$ converges to 0 linearly. Therefore, using $N_2 = \widetilde{O}(d\log(1/\delta)/((1-2\eta)^2\epsilon'))$ samples, after $K = O(\frac{1}{\rho}\log(1/(\exp(t^2/2)\epsilon'))) = O(\log(1/(p\epsilon')))$ iterations, we get a $\mathbf{w}_K$ such that $\theta_K \leq 2\sin(\theta_K/2) \leq \epsilon'\exp(t^2/2)$. Let $h(\mathbf{x}) := \text{sign}(\mathbf{w}_K \cdot \mathbf{x} + \hat{t})$. Equation (13) then implies that the disagreement of $h(\mathbf{x})$ and $f(\mathbf{x})$ is bounded by:

$$
\mathbf{Pr}[h(\mathbf{x}) \neq f(\mathbf{x})] \leq \frac{\epsilon'^2}{4\sqrt{2\pi}} + \frac{\epsilon'}{\pi} \leq \epsilon'.
$$

Furthermore, for any boolean function $h : \mathbb{R}^d \mapsto \{\pm 1\}$ it holds

$$
\text{err}_{0-1}^{\mathcal{D}}(h) = \Pr_{(\mathbf{x},y)\sim\mathcal{D}}[h(\mathbf{x}) \neq y] = \eta + (1-2\eta)\Pr_{\mathbf{x}\sim\mathcal{D}_{\mathbf{x}}}[h(\mathbf{x}) \neq \text{sign}(\mathbf{w}^* \cdot \mathbf{x} + t)].
$$

Thus, to get misclassification error at most $\eta + \epsilon$ (with respect to the $y$), we only need to use $\epsilon' = \epsilon/(1-2\eta)$, and we finally get that $\mathbf{Pr}_{(\mathbf{x},y)\sim\mathcal{D}}[\text{sign}(\mathbf{w}_K \cdot \mathbf{x} + \hat{t}) \neq y] \leq \eta + \epsilon$, using $N_2 = \widetilde{O}(d\log(1/\delta)/((1-2\eta)\epsilon))$ samples. Since the algorithm runs for $O(\log(1/\epsilon))$ iterations, the overall runtime is $\widetilde{O}(N_2 d)$. This completes the proof of Theorem 2.6. $\qquad\square$

*Proof of Theorem 1.3.* From Lemma 2.2, we get that with $\widetilde{O}(d\log(1/\delta)/((1-2\eta)^2 p^2))$ samples Algorithm 4 produces a unit vector $\mathbf{w}_0$ so that $\theta(\mathbf{w}_0, \mathbf{w}^*) \leq \min(1/(5t), \pi/2)$.

Since our guesses of the threshold $t_m$, $m \in [M]$ form a grid of $(\epsilon^2/(8(1-2\eta)^2)$-separated values on the interval $[\sqrt{2\log(1/\hat{p})}, \sqrt{2\log(4/\hat{p})}] \ni t$, which covers all possible values of the true threshold $t$, there exists a $\bar{m} \in [M]$ such that $|t_{\bar{m}} - t| \leq \epsilon^2/(8(1-2\eta)^2)$. Thus, the condition of Theorem 2.6 is satisfied by at least one input threshold. Given $\mathbf{w}_0$, let $\widehat{\mathbf{w}}_m$, $m \in [M]$ be the weight vector produced by Algorithm 5 at call $m = 1, \cdots, M$. From Theorem 2.6, we know that with $\widetilde{O}(d\log(1/\delta)/((1-2\eta)\epsilon))$ samples, with probability at least $1 - \delta$ we get a list of halfspaces $\{h_m(\mathbf{x}) : h_m(\mathbf{x}) = \text{sign}(\widehat{\mathbf{w}}_m + t_m), m = 1, \cdots, M\}$ so that

$$\min_{h_m, m \in [M]} \Pr_{(\mathbf{x}, y) \sim \mathcal{D}}[h_m(\mathbf{x}) \neq y] \leq \eta + \epsilon .$$

Finally, to pick the optimal hypothesis from the list, we utilize the following fact.

**Fact B.10** (Equation (7) in [MN06]). *Let $\mathcal{D}$ be a distribution over $\mathbb{R}^d \times \{\pm 1\}$. Let $\mathcal{F}$ be a concept set of boolean functions with VC dimension at most d. Let $\widehat{\mathcal{D}}$ be the empirical distribution obtained by drawing $\widetilde{O}(d\log(1/\delta)/((1-2\eta)\epsilon))$ samples from $\mathcal{D}$. Then it holds that*

$$\min_{f \in \mathcal{F}} \Pr_{(\mathbf{x}, y) \sim \widehat{\mathcal{D}}}[f(\mathbf{x}) \neq y] \leq \min_{f \in \mathcal{F}} \Pr_{(\mathbf{x}, y) \sim \mathcal{D}}[f(\mathbf{x}) \neq y] + \epsilon .$$

Using the above fact and a sample size of $N_3 = \widetilde{O}(d\log(1/\delta)/((1-2\eta)\epsilon))$ from $\mathcal{D}$, we output the hypothesis with the minimum empirical error. By Fact B.10, we have that this will introduce an error at most $\epsilon$ with probability at least $1 - \delta$. Since $M = O(1/\epsilon^2)$, the total number of calls of Algorithm 5 in Algorithm 3 is $O(1/\epsilon^2)$, and the runtime is $\widetilde{O}(Nd/\epsilon^2)$. $\qquad \square$

---

**Algorithm 6** Testing Procedure

---

**Input**: Hypothesis weight vectors $\widehat{\mathbf{w}}_1, \widehat{\mathbf{w}}_2, \ldots, \widehat{\mathbf{w}}_m$ and thresholds $t_1, t_2, \ldots, t_m$
Draw $N_3$ samples $\{(\mathbf{x}^{(i)}, y^{(i)})\}_{i=1}^{N_3}$ from $\mathcal{D}$
Calculate the test error (the fraction of misclassified points) for $h_m(\mathbf{x}) = \text{sign}(\widehat{\mathbf{w}}_m \cdot \mathbf{x} + t_m)$, $m \in [M]$, using $\{(\mathbf{x}^{(i)}, y^{(i)})\}_{i=1}^{N_3}$
Let $h_{\bar{m}}(\mathbf{x}) = \text{sign}(\widehat{\mathbf{w}}_{\bar{m}} \cdot \mathbf{x} + t_{\bar{m}})$, $\bar{m} \in [M]$ be the halfspace with smallest empirical error.
**return** $\widehat{\mathbf{w}}_{\bar{m}}, t_{\bar{m}}$

---

### B.3 The Case Where Both $\eta$ and $p$ are Unknown

Throughout this section, we carried out the analysis assuming knowledge of the noise parameter $\eta$. We now show how to relax this requirement, without changing the sample complexity (up to constant factors) and only affecting the algorithm runtime by a factor $1/\epsilon$. In the following lemma, we show that with $\widetilde{O}(d\log(1/\delta)/(p^2(1-2\eta)^2))$ samples we can compute constant factor estimates of the values of $p$ and $1 - 2\eta$, which suffice for determining the correct number of samples to draw in all three subprocedures of our main algorithm (i.e., we can correctly determine $N_1$, $N_2$, and $N_3$).

**Lemma B.11.** *There is an algorithm that uses $\widetilde{O}(d\log(1/\delta)/(p^2(1-2\eta)^2))$ samples, and with probability at least $1 - \delta$ outputs estimates $\hat{p}, \hat{\eta}$, so that $C\hat{p} \geq p \geq \hat{p}$ and $C(1-2\hat{\eta}) \geq (1-2\eta) \geq (1-2\hat{\eta})$, where $C > 0$ is a sufficient large absolute constant.*

*Proof Sketch.* We note that Algorithm 4, can, in fact, be used to get an estimate of $(1-2\eta)p$ instead of only $p$. Therefore, Algorithm 4 outputs $\hat{z}$ so that $2\hat{z} \geq (1-2\eta)p \geq \hat{z}$.

We assume for simplicity that $f(\mathbf{x})$ is positively biased, i.e., $\mathbf{E}_{\mathbf{x} \sim \mathcal{N}}[f(\mathbf{x})] \geq 0$. Note that $\mathbf{E}_{(\mathbf{x}, y) \sim \mathcal{D}}[\mathbb{1}\{y = b\}] = (1-2\eta)\Pr_{\mathbf{x} \sim \mathcal{N}}[f(\mathbf{x}) = b] + \eta$, where $b \in \{\pm 1\}$. Because $f(\mathbf{x})$ is positively biased, we have that $\Pr_{\mathbf{x} \sim \mathcal{N}}[f(\mathbf{x}) = 1] \geq 1/2$. Denote the random variable $Z$ as $Z = \mathbb{1}\{y = 1\} - 1/2$. Note that $\mathbf{E}[Z] \geq (1/2)(1-2\eta)(\Pr[f(\mathbf{x}) = 1] - 1/2)$. Note that if $p$ is less than a sufficiently small constant, then $\mathbf{E}[Z] \geq (1/4)(1-2\eta)$, whereas if $p = 1/2$, this expectation does not give any useful information. Note that by standard Chernoff bounds, using $O(N\log(1/\delta))$ samples, where $N$ is a parameter, we can get estimates $\widehat{Z}$, so that $\mathbf{E}[Z] \geq \widehat{Z} - \sqrt{1/N}$. By letting $N = O(1/\hat{z})$, we can distinguish between the cases that $(1-2\eta)(\Pr[f(\mathbf{x}) = 1] - 1/2) \leq \hat{z}$, in

which case we have that $p$ is close to $1/2$, therefore $\hat{z}$ is an estimate of $1 - 2\eta$ that satisfies our requirements. Otherwise, we are in the case where $\mathbf{E}[Z] \geq (1/4)(1 - 2\eta)$. Therefore, we run the following algorithm: In each round $s$, we draw $N_s = O(2^s \log(\log(1/\epsilon)/\delta))$ samples, and we check whether $\widehat{Z}_s - \sqrt{1/N_s} \geq 1/2\widehat{Z}_s$. If $\widehat{Z}_s - \sqrt{1/N_s} < 1/2\widehat{Z}_s$, we continue, otherwise we stop and return $1/2\widehat{Z}_s$, which is an effective lower bound of $(1 - 2\eta)$. The number of rounds is at most $\log(1/\epsilon)$, so by union bound, the probability of success is at least $1 - \delta$. After we have estimated an effective lower bound for $(1 - 2\eta)$, we can get an estimate for the value of $p$, using the estimator $\hat{z}$ (recalling that $2\hat{z} \geq (1 - 2\eta)p \geq \hat{z}$). $\qquad\square$

While Lemma B.11 is sufficient for ensuring that the number of samples our algorithm draws is not higher than when assuming the knowledge of $\eta$, it is not sufficient for correctly translating the 0-1 error and guaranteeing that it is bounded by $\eta + O(\epsilon)$. However, it is not hard to verify that if we run the Optimization procedure for an estimate $\hat{\eta}$ that is within $\pm\epsilon$ of $\eta$, then the correct 0-1 error bound of $\eta + O(\epsilon)$ would follow. This is resolved by simply running the entire optimization component of the algorithm (including all calls to Algorithm 2) for a grid of $\epsilon$-separated values of $\eta$ in the range $(0, 1/2)$, which must contain the true value of $\eta$. It is immediate that this increases the runtime (and the number of hypothesis halfspaces) by a factor $O(1/\epsilon)$. Yet the same number of samples suffices for the optimization and testing, as all that we require is that at least one hypothesis is constructed using estimates of $\eta$ and $t$ that are sufficiently close to their true values (by order-$\epsilon$ and order-$\epsilon^2$, respectively, as discussed before).

## C    Omitted Content from Section 3

### C.1    Background on Hermite Polynomials

We define the standard $L^p$ norms with respect to the Gaussian measure, i.e., $\|g\|_{L^p} = (\mathbf{E}_{\mathbf{x}\sim\mathcal{N}}[|g(\mathbf{x})|^p])^{1/p}$. We denote by $L^2(\mathcal{N})$ the vector space of all functions $f : \mathbb{R}^d \to \mathbb{R}$ such that $\mathbf{E}_{\mathbf{x}\sim\mathcal{N}}[f^2(x)] < \infty$. The usual inner product for this space is $\mathbf{E}_{\mathbf{x}\sim\mathcal{N}}[f(\mathbf{x})g(\mathbf{x})]$. While, usually one considers the probabilist's or physicist's Hermite polynomials, in this work we define the *normalized* Hermite polynomial of degree $i$ to be $\mathrm{He}_0(x) = 1, \mathrm{He}_1(x) = x, \mathrm{He}_2(x) = \frac{x^2-1}{\sqrt{2}}, \ldots, \mathrm{He}_i(x) = \frac{\widehat{\mathrm{He}}_i(x)}{\sqrt{i!}}, \ldots$ where by $\widehat{\mathrm{He}}_i(x)$ we denote the probabilist's Hermite polynomial of degree $i$. The unnormalized Hermite polynomials are defined as $\widehat{\mathrm{He}}_i(z)\exp(-z^2/2) = (-1)^i \frac{\mathrm{d}^i \exp(-z^2/2)}{\mathrm{d}z^i}$. The normalized Hermite polynomials $\mathrm{He}_1, \mathrm{He}_2, \ldots, \mathrm{He}_i, \ldots$ form a complete orthonormal basis for the single dimensional version of the inner product space defined above. To get an orthonormal basis for $L^2(\mathcal{N})$, we use a multi-index $V \in \mathbb{N}^d$ to define the $d$-variate normalized Hermite polynomial as $\mathrm{He}_V(\mathbf{x}) = \prod_{i=1}^d \mathrm{He}_{v_i}(x_i)$. The total degree of $\mathrm{He}_V$ is $|V| = \sum_{v_i \in V} v_i$. Given a function $f \in L^2$, we compute its Hermite coefficients as $\hat{f}(V) = \mathbf{E}_{\mathbf{x}\sim\mathcal{N}}[f(\mathbf{x})\mathrm{He}_V(\mathbf{x})]$ and express it uniquely as $\sum_{V \in \mathbb{N}^d} \hat{f}(V)\mathrm{He}_V(\mathbf{x})$.

### C.2    Additional Background on the SQ Model

To define the SQ dimension, we need the following definition.

**Definition C.1** (Pairwise Correlation). *The pairwise correlation of two distributions with probability density functions (pdfs) $D_1, D_2 : \mathcal{X} \to \mathbb{R}_+$ with respect to a distribution with pdf $D : \mathcal{X} \to \mathbb{R}_+$, where the support of $D$ contains the supports of $D_1$ and $D_2$, is defined as $\chi_D(D_1, D_2) + 1 := \int_{x\in\mathcal{X}} D_1(x)D_2(x)/D(x)\mathrm{d}x$. We say that a collection of $s$ distributions $\mathfrak{D} = \{D_1, \ldots, D_s\}$ over $\mathcal{X}$ is $(\gamma, \beta)$-correlated relative to a distribution $D$ if $|\chi_D(D_i, D_j)| \leq \gamma$ for all $i \neq j$, and $|\chi_D(D_i, D_j)| \leq \beta$ for $i = j$.*

The following notion of dimension effectively characterizes the difficulty of the decision problem.

**Definition C.2** (SQ Dimension). *For $\gamma, \beta > 0$, a decision problem $\mathcal{B}(\mathfrak{D}, D)$, where $D$ is fixed and $\mathfrak{D}$ is a family of distributions over $\mathcal{X}$, let $s$ be the maximum integer such that there exists $\mathfrak{D}_D \subseteq \mathfrak{D}$ such that $\mathfrak{D}_D$ is $(\gamma, \beta)$-correlated relative to $D$ and $|\mathfrak{D}_D| \geq s$. We define the* Statistical Query dimension with pairwise correlations $(\gamma, \beta)$ *of $\mathcal{B}$ to be $s$ and denote it by $\mathrm{SD}(\mathcal{B}, \gamma, \beta)$.*

The connection between SQ dimension and lower bounds is captured by the following lemma.

**Lemma C.3** ([FGR+17]). *Let $\mathcal{B}(\mathfrak{D}, D)$ be a decision problem, where $D$ is the reference distribution and $\mathfrak{D}$ is a class of distributions over $\mathcal{X}$. For $\gamma, \beta > 0$, let $s = \mathrm{SD}(\mathcal{B}, \gamma, \beta)$. Any SQ algorithm that solves $\mathcal{B}$ with probability at least $2/3$ requires at least $s \cdot \gamma/\beta$ queries to the $\mathrm{VSTAT}(1/\gamma)$ oracle.*

In order to construct a large set of nearly uncorrelated hypotheses, we need the following fact:

**Fact C.4** (see, e.g., [DKS17]). *Let $d \in \mathbb{Z}_+$. Let $0 < c < 1/2$. There exists a collection $\mathcal{S}$ of $2^{\Omega(d^c)}$ unit vectors in $\mathbb{R}^d$, such that any pair $\mathbf{v}, \mathbf{u} \in \mathcal{S}$, with $\mathbf{v} \neq \mathbf{u}$, satisfies $|\mathbf{v} \cdot \mathbf{u}| < d^{-1/2+c}$.*

### C.3 Omitted Details from the Proof of Theorem 3.2

**Claim C.5.** *Let $f_{\mathbf{v}}(\mathbf{x}) = \mathrm{sign}(\mathbf{v} \cdot \mathbf{x} - t)$ and $f_{\mathbf{u}}(\mathbf{x}) = \mathrm{sign}(\mathbf{u} \cdot \mathbf{x} - t)$ and let $c_i$ be the Hermite coefficient of $\mathrm{He}_i$. Then, it holds $\mathbf{E}_{\mathbf{x} \sim \mathcal{N}}[f_{\mathbf{v}}(\mathbf{x}) f_{\mathbf{u}}(\mathbf{x})] = \sum_{i=0}^{\infty} \cos^i \theta \, c_i^2$ , where $\theta$ is the angle between $\mathbf{v}$ and $\mathbf{u}$.*

*Proof.* We first need the following standard fact about the rotations of Hermite polynomials (see, e.g., Fact D.1 in [DKS17]):

**Fact C.6.** *For $\theta \in \mathbb{R}$, it holds $\mathrm{He}_i(x \cos(\theta) + y \sin \theta) = \sum_{j=0}^{i} \binom{i}{j} \cos^j \theta \sin^{i-j} \theta \mathrm{He}_j(x) \mathrm{He}_{i-j}(y)$ .*

We have that

$$
\mathop{\mathbf{E}}_{\mathbf{x} \sim \mathcal{N}}[f_{\mathbf{v}}(\mathbf{x}) f_{\mathbf{u}}(\mathbf{x})] = \mathop{\mathbf{E}}_{\mathbf{x}_1, \mathbf{x}_2 \sim \mathcal{N}}[\mathrm{sign}(\mathbf{x}_1 - t)\mathrm{sign}(\cos\theta\mathbf{x}_1 + \sin\theta\mathbf{x}_2 - t)]
$$

$$
= \mathop{\mathbf{E}}_{\mathbf{x}_1, \mathbf{x}_2 \sim \mathcal{N}}\left[\left(\sum_{i=0}^{\infty} c_i \mathrm{He}_i(\mathbf{x}_1)\right)\left(\sum_{i=0}^{\infty} c_i \mathrm{He}_i(\cos\theta\mathbf{x}_1 + \sin\theta\mathbf{x}_2)\right)\right]
$$

$$
= \mathop{\mathbf{E}}_{\mathbf{x}_1 \sim \mathcal{N}}\left[\left(\sum_{i=0}^{\infty} c_i \mathrm{He}_i(\mathbf{x}_1)\right) \mathop{\mathbf{E}}_{\mathbf{x}_2 \sim \mathcal{N}}\left[\left(\sum_{i=0}^{\infty} c_i \mathrm{He}_i(\cos\theta\mathbf{x}_1 + \sin\theta\mathbf{x}_2)\right)\right]\right]
$$

$$
= \mathop{\mathbf{E}}_{\mathbf{x}_1 \sim \mathcal{N}}\left[\left(\sum_{i=0}^{\infty} c_i \mathrm{He}_i(\mathbf{x}_1)\right)\left(\sum_{i=0}^{\infty} \cos^i \theta c_i \mathrm{He}_i(\mathbf{x}_1)\right)\right] = \sum_{i=0}^{\infty} \cos^i \theta \, c_i^2 ,
$$

where in the third equality, we used Fact C.6 and the orthogonality of the Hermite polynomials with respect to the Gaussian. $\qquad \square$

We prove the following.

**Claim C.7.** *It holds that $\mathbf{E}_{z \sim \mathcal{N}}[\mathrm{sign}(z - t)\mathrm{He}_i(z)] = 2(i)^{-1/2}\mathrm{He}_{i-1}(t) \exp(-t^2/2)$ .*

*Proof.* Denote as $\widehat{\mathrm{He}}_i(z)$ the non-normalized Hermite polynomial of order $d$. The Hermite polynomials are defined as follows:

$$
\widehat{\mathrm{He}}_i(z) \exp(-z^2/2) = (-1)^i \frac{\mathrm{d}^i \exp(-z^2/2)}{\mathrm{d}z^i} .
$$

By taking the derivative over $z$ (which exists as $\mathrm{He}_i$ is a polynomial and $\exp(-z^2/2)$ is differentiable), we have that $\int(\widehat{\mathrm{He}}_i(z) \exp(-z^2/2)) = -\widehat{\mathrm{He}}_{i-1}(z)$. Therefore, we have that

$$
\mathop{\mathbf{E}}_{z \sim \mathcal{N}}[\mathrm{sign}(z - t)\widehat{\mathrm{He}}_i(z)] = \int_{z \in \mathbb{R}} \mathrm{sign}(z - t)\widehat{\mathrm{He}}_i(z)G(z)\mathrm{d}z
$$

$$
= 2 \int_t^{\infty} \widehat{\mathrm{He}}_i(z)G(z)\mathrm{d}z ,
$$

where we used that $\int_{z \in \mathbb{R}} \widehat{\mathrm{He}}_i(z)G(z)\mathrm{d}z = 0$ by the orthogonality of the Hermite Polynomials with respect to the Gaussian measure. Furthermore, using that $\int(\widehat{\mathrm{He}}_i(z) \exp(-z^2/2)) = -\widehat{\mathrm{He}}_{i-1}(z)$ we get that

$$
\mathop{\mathbf{E}}_{z \sim \mathcal{N}}[\mathrm{sign}(z - t)\widehat{\mathrm{He}}_i(z)] = 2 \int_t^{\infty} (-\widehat{\mathrm{He}}_{i-1}(z)G(z))'\mathrm{d}z
$$

$$
= 2\widehat{\mathrm{He}}_{i-1}(t)G(t) .
$$

By normalizing the Hermite polynomial, we complete the proof of Claim C.7. $\qquad\square$

## C.4 Proof of Lemma 3.6

We restate and prove the following lemma.

**Lemma C.8.** *Let $D_0$ be a product distribution over $\mathcal{N} \times \{\pm 1\}$, where $\mathbf{Pr}_{(\mathbf{x},y)\sim D_0}[y = 1] = \mathbf{Pr}_{(\mathbf{x},y)\sim D_\mathbf{v}}[y = 1] = p$. We have $\chi_{D_0}(D_\mathbf{v}, D_\mathbf{u}) \leq 2(1-2\eta)(\mathbf{E}[f_\mathbf{v}(\mathbf{x})f_\mathbf{u}(\mathbf{x})] - \mathbf{E}[f_\mathbf{v}(\mathbf{x})]\,\mathbf{E}[f_\mathbf{u}(\mathbf{x})])$ and $\chi^2(D_\mathbf{v}, D_0) \leq (1 - 2\eta)(\mathbf{E}[f_\mathbf{v}(\mathbf{x})] - \mathbf{E}[f_\mathbf{v}(\mathbf{x})]^2)$.*

*Proof.* We have that
$$\chi_{D_0}(D_\mathbf{v}, D_\mathbf{u}) = \mathop{\mathbf{Pr}}_{(\mathbf{x},y)\sim D_\mathbf{v}}[y = 1]\chi_\mathcal{N}(A_\mathbf{v}, A_\mathbf{u}) + \mathop{\mathbf{Pr}}_{(\mathbf{x},y)\sim D_\mathbf{v}}[y = 0]\chi_\mathcal{N}(B_\mathbf{v}, B_\mathbf{u})$$
$$= (1/p)\chi_\mathcal{N}(A_\mathbf{v}, A_\mathbf{u}) + (1/(1 - p))\chi_\mathcal{N}(B_\mathbf{v}, B_\mathbf{u}) \ .$$

We bound each term. Note that by construction $A_\mathbf{v}(\mathbf{x}) = G(\mathbf{x})(\eta + (1 - 2\eta)\mathbb{1}\{f_\mathbf{v}(\mathbf{x}) > 0\})/(\eta + (1 - 2\eta)\,\mathbf{E}_{\mathbf{x}\sim\mathcal{N}}[\mathbb{1}\{f_\mathbf{v}(\mathbf{x}) > 0\}])$. Note that $\mathbb{1}\{f_\mathbf{v}(\mathbf{x}) > 0\} = (f(\mathbf{x}) + 1)/2$, therefore $A_\mathbf{v}(\mathbf{x}) = G(\mathbf{x})(1 + (1 - 2\eta)f_\mathbf{v}(\mathbf{x}))/(1 + (1 - 2\eta)\,\mathbf{E}_{\mathbf{x}\sim\mathcal{N}}[f_\mathbf{v}(\mathbf{x})])$. Therefore,
$$\frac{A_\mathbf{v}(\mathbf{x})}{G(\mathbf{x})} - 1 = \frac{(1 - 2\eta)}{p}\left(f_\mathbf{v}(\mathbf{x}) - \mathop{\mathbf{E}}_{\mathbf{x}\sim\mathcal{N}}[f_\mathbf{v}(\mathbf{x})]\right) \ .$$

Using the above, we get that $\chi_\mathcal{N}(A_\mathbf{v}, A_\mathbf{u}) = (1 - 2\eta)/p(\mathbf{E}[f_\mathbf{v}(\mathbf{x})f_\mathbf{u}(\mathbf{x})] - \mathbf{E}[f_\mathbf{v}(\mathbf{x})]\,\mathbf{E}[f_\mathbf{u}(\mathbf{x})])$. Similarly, we get that $\chi_\mathcal{N}(B_\mathbf{v}, B_\mathbf{u}) = (1 - 2\eta)/(1 - p)(\mathbf{E}[f_\mathbf{v}(\mathbf{x})f_\mathbf{u}(\mathbf{x})] - \mathbf{E}[f_\mathbf{v}(\mathbf{x})]\,\mathbf{E}[f_\mathbf{u}(\mathbf{x})])$. It remains to bound $\chi^2(D_\mathbf{v}, D_0)$. Note that $\chi^2(D_\mathbf{v}, D_0) = \chi_{D_0}(D_\mathbf{v}, D_\mathbf{v})$, hence, $\chi^2(D_\mathbf{v}, D_0) \leq (1 - 2\eta)\,\mathbf{E}_{\mathbf{x}\sim\mathcal{N}}([f_\mathbf{v}(\mathbf{x})] - \mathbf{E}_{\mathbf{x}\sim\mathcal{N}}[f_\mathbf{v}(\mathbf{x})]^2)$. $\qquad\square$

## C.5 Reduction of Testing to Learning

**Lemma C.9** (Reduction of Testing to Learning). *Any algorithm that learns halfspaces with $\eta = 1/3$ RCN noise can be used to solve the decision problem of Theorem 3.2.*

*Proof.* Assume that there is an algorithm $\mathcal{A}$ which given $\epsilon > 0$ and distribution $D$ with Gaussian $\mathbf{x}$- marginals and corrupted with $\eta = 1/3$ random classification noise, outputs a hypothesis $h$ with $\mathbf{Pr}_{(\mathbf{x},y)\sim D}[h(\mathbf{x}) \neq y] \leq \eta + \epsilon$. We can use $\mathcal{A}$ to solve the decision problem $\mathcal{B}(D_0, \mathfrak{D})$. Note that if the distribution were $D_0$, then any hypothesis would get error at least $\eta + (1 - 2\eta)p$ (as $y$ is independent of $\mathbf{x}$). If the distribution were one in the set $\mathfrak{D}$, then the algorithm for $\epsilon = (1 - 2\eta)p/2$ would give a hypothesis such that $\mathbf{Pr}_{(\mathbf{x},y)\sim D}[h(\mathbf{x}) \neq y] \leq \eta + (1 - 2\eta)p/2$. So making one additional query of tolerance $(1 - 2\eta)p$ would be able to solve the decision problem. This completes the proof. $\qquad\square$

## C.6 Solving the Decision Problem Efficiently

In this section, we show that our SQ lower bound (Theorem 3.2) for the testing problem is, in fact, tight. We prove the following:

**Theorem C.10** (Efficient Algorithm for Testing). *Let $d \in \mathbb{N}$ and $\epsilon \in (0, 1)$ and let $\mathcal{D}$ be a distribution supported on $\mathbb{R}^d \times \{\pm 1\}$ such that $\mathcal{D}_\mathbf{x}$ is the standard Gaussian on $\mathbb{R}^d$. There exists an algorithm that, given $N = C\sqrt{d}/(\epsilon^2 \log(1/\epsilon))$ samples from $\mathcal{D}$, where $C > 0$ is a sufficiently large absolute constant, distinguishes between the following cases with probability of error at most 1/3:*

*1. $\mathbf{x}$ is independent of $y$, where $(\mathbf{x}, y) \sim \mathcal{D}$.*

*2. $y$ is $f(\mathbf{x})$ corrupted with RCN for $\eta = 1/3$, where $f$ is an LTF with $\mathbf{Pr}[f(\mathbf{x}) = 1] = \epsilon$.*

*Proof.* Let $\mathbf{Z} = y\mathbf{x}$ be the random variable where $(\mathbf{x}, y) \sim \mathcal{D}$ and let $\mathbf{Z}_N = (1/N)\sum_{i=1}^N \mathbf{x}^{(i)}y^{(i)}$ be the random variables where $(\mathbf{x}^{(i)}, y^{(i)})$, for $i \in [N]$ are samples drawn from $\mathcal{D}$. The tester works as follows: If $\|\mathbf{Z}_N\|_2^2 > d/N + c\epsilon^2 \log(1/\epsilon)$, where $c > 0$ is a sufficiently small universal constant, we answer that we are in Case 2, otherwise that we are in the Case 1. We prove the correctness of the algorithm below.

We note that if we are in the Case 1, then $Z$ follows the standard Gaussian distribution. To see that, we show the following simple claim:

**Claim C.11.** *Let $\mathbf{x}$ be distributed as standard normal and let $y$ supported in $\{\pm 1\}$ be a random variable independent of $\mathbf{x}$, then $y\mathbf{x}$ is distributed as standard normal.*

*Proof.* We assume that $y = 1$ with probability $1-\eta$ for some $\eta \in (0,1)$. Let $\phi_A(t) = \mathbf{E}[\exp(itA)]$ be the characteristic function of $A$. Then, we have that $\phi_{y\mathbf{x}}(t) = \mathbf{E}[\exp(ity\mathbf{x})] = (1-\eta)\mathbf{E}[\exp(it\mathbf{x})] + \eta\mathbf{E}[\exp(-it\mathbf{x})] = (1-\eta)\phi_{\mathbf{x}}(t) + \eta\phi_{-\mathbf{x}}(t) = \phi_{\mathbf{x}}(t)$, where in the last equality we used that $\phi_{-\mathbf{x}}(t) = \phi_{\mathbf{x}}(t)$ as the standard normal distribution is symmetric. Therefore, $\phi_{y\mathbf{x}}(t) = \phi_{\mathbf{x}}(t)$, hence the distribution of $y\mathbf{x}$ and $\mathbf{x}$ is the same. $\qquad\square$

From Claim C.11, we have that $\mathbf{Z}$ follows standard normal distribution if we are in Case 1 and therefore $\mathbf{Z}_N$ follows $\mathcal{N}(\mathbf{0}, \mathbf{I}/N)$. Hence, $\|\mathbf{Z}_N\|_2^2$ has mean $d/N$ and standard deviation $O(\sqrt{d}/N)$. By Chebyshev's inequality, we answer correctly in this case with a probability of at least $2/3$.

We next analyze the case where $Z$ is in Case 2. Let $f(\mathbf{x}) = \text{sign}(\mathbf{v} \cdot \mathbf{x} + t)$ be the defining halfspace. Then for any $\mathbf{u}$ orthogonal to $\mathbf{v}$, we have that the random variables $y$ and $(\mathbf{x} \cdot \mathbf{u})$ are independent as $y$ only depends on $\mathbf{x} \cdot \mathbf{v}$. Therefore, by Claim C.11, we have that $y(\mathbf{x} \cdot \mathbf{u})$ is standard normal, therefore $y\mathbf{x}^{\perp \mathbf{v}}$ is standard $(d-1)$-dimensional normal. Furthermore, note that from Equation (1) that $\mathbf{E}[y\mathbf{x} \cdot \mathbf{v}] = \Theta(\epsilon\sqrt{\log(1/\epsilon)})$ and $\text{Var}[y\mathbf{x} \cdot \mathbf{v}] = \Theta(1)$. Therefore, we can write $\|\mathbf{Z}_N\|_2^2 = (1/N)(\sum_{i=1}^{d-1} \mathbf{g}_i^2 + \mathbf{E}^2)$, where $\mathbf{g}_i$ are distributed as standard normal and $\mathbf{E}^2$ is the contribution due to the noise. We show that with probability at least $2/3$, it holds that $\|\mathbf{Z}_N\|_2^2 \geq d/N + c\epsilon^2 \log(1/\epsilon)$. Note that with probability at least $9/10$, we have that $|\mathbf{E}| = \Theta(\epsilon\sqrt{\log(1/\epsilon)})$ by Chebyshev's inequality. Due to the independence of the directions, the random variables $\mathbf{g}_i$ for $i = 1, \ldots, d-1$ are independent of $\mathbf{E}$. Hence, conditioned on the event that $|\mathbf{E}| = \Theta(\epsilon\sqrt{\log(1/\epsilon)})$, we have that $\|\mathbf{Z}_n\|_2^2$ has mean $(d-1)/N + \Theta(\epsilon^2 \log(1/\epsilon))$ and standard deviation $O(\sqrt{d}/N)$. Hence, again the tester would succeed with a probability of at least $2/3$. $\qquad\square$

### C.7 SQ Algorithm for Learning Halfspaces with RCN with Exponential Number of Queries

Here we show that there exists a query-inefficient SQ algorithm that can be simulated with near-optimal sample complexity of $\tilde{O}(d/\epsilon)$.

**Lemma C.12** (Inefficient SQ Algorithm). *There is an SQ algorithm that makes $2^{O(d)\text{polylog}(1/\epsilon)}$ queries to $\text{VSTAT}(1/\epsilon)$, and learns the class of halfspaces on $\mathbb{R}^d$ in the presence of RCN with $\eta = 1/3$ with error at most $\eta + \epsilon$.*

*Proof Sketch.* We note that it is always possible to design an exponential query SQ algorithm that achieves the optimal sample complexity of $\tilde{O}(d/\epsilon)$ (if we were to simulate it with samples). One approach is to generate an $\epsilon$-cover $\mathcal{G}$ that encompasses all hypotheses in $\mathbb{R}^d$ (with size roughly $(1/\epsilon)^d$), and then utilize the query function $f(\mathbf{x}, y) = \mathbb{1}\{h_1(\mathbf{x}) \neq y\} - \mathbb{1}\{h_2(\mathbf{x}) \neq y\}$ for any $h_1, h_2 \in \mathcal{G}$. It can be readily observed that the variance of $f$ is at most the probability that $h_1(\mathbf{x}) \neq h_2(\mathbf{x})$, which means that $\text{VSTAT}(1/\epsilon)$ can distinguish which hypothesis, $h_1$ or $h_2$, yields a smaller error. Note that if we were to simulate this SQ algorithm using samples, we would also need to do a union bound over the set of all hypotheses. $\qquad\square$

## D  Lower Bound for Low-Degree Polynomial Testing

### D.1  Preliminaries: Low-Degree Method

We begin by recording the necessary notation, definitions, and facts. This section mostly follows [BBH+20].

**Low-Degree Polynomials**  A function $f : \mathbb{R}^a \to \mathbb{R}^b$ is a polynomial of degree at most $k$ if it can be written in the form

$$f(x) = (f_1(x), f_2(x), \ldots, f_b(x)) \,,$$

where each $f_i : \mathbb{R}^a \to \mathbb{R}$ is a polynomial of degree at most $k$. We allow polynomials to have random coefficients as long as they are independent of the input $x$. When considering *list-decodable estimation* problems, an algorithm in this model of computation is a polynomial $f : \mathbb{R}^{d_1 \times n} \to \mathbb{R}^{d_2 \times \ell}$, where $d_1$ is the dimension of each sample, $n$ is the number of samples, $d_2$ is the dimension of the output hypotheses, and $\ell$ is the number of hypotheses returned. On the other hand, [BBH$^+$20] focuses on *binary hypothesis testing* problems defined in Definition D.2.

A degree-$k$ polynomial test for Definition D.2 is a degree-$k$ polynomial $f : \mathbb{R}^{d \times n} \to \mathbb{R}$ and a threshold $t \in \mathbb{R}$. The corresponding algorithm consists of evaluating $f$ on the input $x_1, \ldots, x_n$ and returning $H_0$ if and only if $f(x_1, \ldots, x_n) > t$.

**Definition D.1** ($n$-sample $\epsilon$-good distinguisher)**.** *We say that the polynomial $p : \mathbb{R}^{d \times n} \mapsto \mathbb{R}$ is an $n$-sample $\epsilon$-distinguisher for the hypothesis testing problem in Definition D.2 if*

$$\left| \mathop{\mathbf{E}}_{X \sim D_0^{\otimes n}} [p(X)] - \mathop{\mathbf{E}}_{u \sim \mu} \mathop{\mathbf{E}}_{X \sim D_u^{\otimes n}} [p(X)] \right| \geq \epsilon \sqrt{\mathrm{Var}_{X \sim D_0^{\otimes n}} [p(X)]}.$$

*We call $\epsilon$ the* advantage *of the distinguisher.*

Let $\mathcal{C}$ be the linear space of polynomials with a degree at most $k$. The best possible advantage is given by the *low-degree likelihood ratio*

$$\max_{\substack{p \in \mathcal{C} \\ \mathbf{E}_{X \sim D_0^{\otimes n}}[p^2(X)] \leq 1}} \left| \mathop{\mathbf{E}}_{u \sim \mu} \mathop{\mathbf{E}}_{X \sim D_u^{\otimes n}} [p(X)] - \mathop{\mathbf{E}}_{X \sim D_0^{\otimes n}} [p(X)] \right| = \left\| \mathop{\mathbf{E}}_{u \sim \mu} \left[ (\bar{D}_u^{\otimes n})^{\leq k} \right] - 1 \right\|_{D_0^{\otimes n}},$$

where we denote $\bar{D}_u = D_u / D_0$ and the notation $f^{\leq k}$ denotes the orthogonal projection of $f$ to $\mathcal{C}$.

Another notation we will use regarding a finer notion of degrees is the following: We say that the polynomial $f(x_1, \ldots, x_n) : \mathbb{R}^{d \times n} \to \mathbb{R}$ has *samplewise degree* $(r, k)$ if it is a polynomial, where each monomial uses at most $k$ different samples from $x_1, \ldots, x_n$ and uses degree at most $r$ for each of them. In analogy to what was stated for the best degree-$k$ distinguisher, the best distinguisher of samplewise degree $(r, k)$-achieves advantage $\left\| \mathbf{E}_{u \sim \mu}[(\bar{D}_u^{\otimes n})^{\leq r, k}] - 1 \right\|_{D_0^{\otimes n}}$ the notation $f^{\leq r, k}$ now means the orthogonal projection of $f$ to the space of all samplewise degree-$(r, k)$ polynomials with unit norm.

We begin by formally defining a hypothesis problem.

**Definition D.2** (Hypothesis testing)**.** *Let $D_0$ be a distribution and $\mathcal{S} = \{D_u\}_{u \in S}$ be a set of distributions on $\mathcal{X}$. Let $\mu$ be a prior distribution on the indices $S$ of that family. We are given access (via i.i.d. samples or oracle) to an* underlying *distribution where one of the two is true:*

- *$H_0$: The underlying distribution is $D_0$.*

- *$H_1$: First $u$ is drawn from $\mu$ and then the underlying distribution is set to be $D_u$.*

*We say that a (randomized) algorithm solves the hypothesis testing problem if it succeeds with non-trivial probability (i.e., greater than $0.9$).*

**Definition D.3.** *Let $D_0$ be the joint distribution over the pairs $(\mathbf{x}, y) \in \mathbb{R}^d \times \{\pm 1\}$ where $\mathbf{x} \sim \mathcal{N}$ and $y \sim D_0(y)$ independently of $\mathbf{x}$. Let $D_\mathbf{v}$ be the joint distribution over pairs $(\mathbf{x}, y) \in \mathbb{R}^d \times \{\pm 1\}$ where the marginal on $y$ is again $D_0(y)$ but the conditional distribution $E_\mathbf{v}(\mathbf{x}|1)$ is of the form $A_\mathbf{v}$ (as in Theorem 3.2) and the conditional distribution $E_\mathbf{v}(\mathbf{x}| - 1)$ is of the form $B_\mathbf{v}$ . Define $\mathcal{S} = \{E_\mathbf{v}\}_{\mathbf{v} \in S}$ for $S$ being the set of $d$-dimensional nearly orthogonal vectors from Fact C.4 and let the hypothesis testing problem be distinguishing between $D_0$ vs. $\mathcal{S}$ with prior $\mu$ being the uniform distribution on $S$.*

In this section, we prove the following:

**Theorem D.4.** *Let $0 < c < 1/2$. Consider the hypothesis testing problem of Definition D.3. For $d \in \mathbb{Z}_+$ with $d$ larger than an absolute constant, any $n \leq \Omega(d)^{1/2-c}/p^2$ and any even integer $k < d^{c/4}$, we have that*

$$\left\| \mathop{\mathbf{E}}_{\mathbf{v} \sim \mu} \left[ (\bar{E}_\mathbf{v}^{\otimes n})^{\leq \infty, \Omega(k)} \right] - 1 \right\|_{D_0^{\otimes n}}^2 \leq 1.$$

We need the following variant of the statistical dimension from [BBH+20], which is closely related to the hypothesis testing problems considered in this section. Since this is a slightly different definition from the statistical dimension (SD) used so far, we will assign the distinct notation (SDA) for it.

**Notation** For $f : \mathbb{R} \to \mathbb{R}$, $g : \mathbb{R} \to \mathbb{R}$ and a distribution $D$, we define the inner product $\langle f, g \rangle_D = \mathbf{E}_{X \sim D}[f(X)g(X)]$ and the norm $\|f\|_D = \sqrt{\langle f, f \rangle_D}$.

**Definition D.5** (Statistical Dimension). *For the hypothesis testing problem of Definition D.2, we define the* statistical dimension $\mathrm{SDA}(\mathcal{S}, \mu, n)$ *as follows:*

$$\mathrm{SDA}(\mathcal{S}, \mu, n) = \max \left\{ q \in \mathbb{N} : \mathop{\mathbf{E}}_{u,v \sim \mu}[|\langle \bar{D}_u, \bar{D}_v \rangle_{D_0} - 1| \mid E] \leq \frac{1}{n} \text{ for all events } E \text{ s.t. } \mathop{\mathbf{Pr}}_{u,v \sim \mu}[E] \geq \frac{1}{q^2} \right\} .$$

*We will omit writing $\mu$ when it is clear from the context.*

## D.2 Proof of Theorem D.4

To prove Theorem D.4, we first need to bound the SDA of our setting. The following lemma translates the $(\gamma, \beta)$-correlation of $\mathcal{S}$ to a lower bound for the statistical dimension of the hypothesis testing problem. The proof is very similar to that of Corollary 8.28 of [BBH+20] but it is given below for completeness.

**Lemma D.6.** *Let $0 < c < 1/2$ and $d, m \in \mathbb{Z}_+$. Consider the hypothesis testing problem of Definition D.3. Then, for any $q \geq 1$,*

$$\mathrm{SDA}\left(\mathcal{D}, \left(\frac{p^{-1}\Omega(d)^{1/2-c}}{p(q^2/2^{\Omega(d^c/2)} + 1)}\right)\right) \geq q .$$

*Proof.* The first part is to calculate the correlation of the set $\mathcal{S}$. By Theorem 3.2, we know that the set $\mathcal{S}$ is $(\gamma, \beta)$-correlated with $\gamma = p^2 \Omega(d)^{c-1/2}$ and $\beta = 4p$.

We next calculate the SDA according to Definition D.5. We denote by $\bar{E}_\mathbf{v}$ the ratios of the density of $E_\mathbf{v}$ to the density of $R$. Note that the quantity $\langle \bar{E}_\mathbf{u}, \bar{E}_\mathbf{v} \rangle - 1$ used there is equal to $\langle \bar{E}_\mathbf{u} - 1, \bar{E}_\mathbf{v} - 1 \rangle$. Let $E$ be an event that has $\mathbf{Pr}_{\mathbf{u},\mathbf{v} \sim \mu}[E] \geq 1/q^2$. For $d$ sufficiently large we have that

$$\mathop{\mathbf{E}}_{u,v \sim \mu}[|\langle \bar{E}_\mathbf{u}, \bar{E}_\mathbf{v} \rangle - 1| \, | E] \leq \min\left(1, \frac{1}{|\mathcal{S}| \, \mathbf{Pr}[E]}\right) \beta + \max\left(0, 1 - \frac{1}{|\mathcal{S}| \, \mathbf{Pr}[E]}\right) \gamma$$

$$\leq p\left(\frac{q^2}{2^{\Omega(d^c)}} + \frac{p}{\Omega(d)^{1/2-c}}\right) = p\left(\frac{p^{-1}\Omega(d)^{1/2-c}}{q^2/2^{\Omega(d^c/2)} + 1}\right)^{-1} ,$$

where the first inequality uses that $\mathbf{Pr}[\mathbf{u} = \mathbf{v}|E] = \mathbf{Pr}[\mathbf{u} = \mathbf{v}, E]/\mathbf{Pr}[E]$ and bounds the numerator in two different ways: $\mathbf{Pr}[\mathbf{u} = \mathbf{v}, E]/\mathbf{Pr}[E] \leq \mathbf{Pr}[\mathbf{u} = \mathbf{v}]/\mathbf{Pr}[E] = 1/(|\mathcal{S}| \, \mathbf{Pr}[E])$ and $\mathbf{Pr}[\mathbf{u} = \mathbf{v}, E]/\mathbf{Pr}[E] \leq \mathbf{Pr}[E]/\mathbf{Pr}[E] = 1$. $\square$

In [BBH+20], the following relation between SDA and low-degree likelihood ratio is established.

**Fact D.7** (Theorem 4.1 of [BBH+20]). *Let $\mathcal{D}$ be a hypothesis testing problem on $\mathbb{R}^d$ with respect to null hypothesis $D_0$. Let $n, k \in \mathbb{N}$ with $k$ even. Suppose that for all $0 \leq n' \leq n$, $\mathrm{SDA}(\mathcal{S}, n') \geq 100^k (n/n')^k$. Then, for all $r$, $\left\|\mathbf{E}_{u \sim \mu}\left[(\bar{D}_u^{\otimes n})^{\leq r, \Omega(k)}\right] - 1\right\|_{D_0^{\otimes n}}^2 \leq 1$.*

In Lemma D.6 we set $n = \Omega(d)^{1/2-c}/p^2$ and $q = \sqrt{2^{\Omega(d^c/2)}(n/n')}$. Then, $\mathrm{SDA}(\mathcal{S}, n') \geq \sqrt{2^{\Omega(d^c/2)}(n/n')} \geq (100n/n')^k$ for $k < d^c/4$ and then we apply the theorem above.

