# OpenReview forum: "Near-Optimal Bounds for Learning Gaussian Halfspaces with Random Classification Noise"
_NeurIPS.cc/2023/Conference — NeurIPS 2023 poster_

### Official Review · Reviewer_imug · 2023-07-03

**Soundness:** 4 excellent
**Presentation:** 3 good
**Contribution:** 3 good
**Rating:** 7
**Confidence:** 3

**Summary:**

The paper considers the problem of learning $d$-dimensional halfspaces $h(x) = \mathrm{sign}(w \cdot x + t)$ over Gaussian marginals under random classification noise (where with probability $\eta$, a sample is given the incorrect label). While the homogeneous case (where $t = 0$ and the bias $p = \frac12$) by prior works, the paper focuses on the in-homogenous case where $t$ is unknown. The results give a nearly tight characterization of this regime:
* Theorem 1.3 shows that there exists a polynomial-time algorithm for learning in-homogeneous halfspaces that uses $N = \tilde{O}(d / ((1 - \eta)\epsilon) + d / \max(p(1 - 2\eta), \epsilon)^2) \log \frac1\delta$ samples. The algorithm (given and analyzed in Section 2 and Appendix B) works as follows:
  * The Initialization procedure chooses a weight vector $w_0$ and predicts the bias $\hat{p}$ by iteratively averaging together signed samples until the norm of their average is sufficiently large.
  * The Optimization procedure locally updates $w_0$ by running Riemannian subgradient descent on a band of inputs on unit sphere of width $\hat{t}$, for each $\hat{t}$ belonging in an $\epsilon$-net over all thresholds $t$.
  * The Testing procedure identifies which threshold gives the lowest error on a new sample and outputs such a classifier.
* Theorem 1.5 shows the near-optimality of the upper bound by showing that SQ-learning $p$-biased halfspaces under constant RCN requires either an exponential number of queries or at queries to an oracle of accuracy $d^{1/2 - c} / p^2$, whose dependence on $p$ matches the upper bounds.


**Strengths:**

The paper is well-written, the bounds are mathematically interesting and have clear proofs, and the matching dependence on $1 / p^2$ in the upper and lower bounds is a strong argument that these bounds are nearly optimal. While I did not review all of the details of the proofs in the appendices, I didn't find any technical red flags. Theorems and lemmas are well-stated and easy to follow.

**Weaknesses:**

A minor note: The pseudocode for Algorithm 1 could be written in way that makes it more easily digestible for readers. Some helpful details (like the sample complexity for Initialization and Testing) could be included, while terms like $M$ could probably be given asymptotically for simplicity.

**Questions:**

The upper and lower bounds have a substantial gap in $d$. Do the authors believe that this gap could be closed by a future work? And does a similar gap exist in the literature for learning homogeneous halfspaces?

Likewise, do you expect it to be possible to prove a lower bound with a more sensitive dependence on the classification noise $\eta$? It would be interesting to see if the problem gets much harder as the noise level approaches $\frac12$.

I don't exactly see why the Optimization procedure is subgradient descent on a leaky ReLU rather than just a scaled ReLU; if it were leaky, shouldn't there be another additive term in the gradient computation multiplied by $\eta$, rather than just one term in the band multiplied by $(1 - \eta)$?

**Limitations:**

All limitations are well-documented in the assumptions underlying the theoretical results.

---

> ### Author Rebuttal · Authors · 2023-08-10
>
> We thank the reviewer for their time and effort and their positive assessment. Below we reply to the questions in detail.
>
> 1. (**Question 1**) We would like to remark that the optimal lower bound should have a linear dependence on $d$, since even for the simpler realizable case the sample complexity of PAC learning a halfspace has a linear dependence on $d$ (actually, the sample complexity is $\tilde{O}(d/\epsilon))$. However, note that a lower bound of $\Omega(d/\epsilon + \sqrt{d}/\mathrm{max}(\epsilon,p)^2)$ follows from our results, where the first linear term $d/\epsilon$ comes from the information-theoretic bound. Therefore it is the second term that should be potentially strengthened from $\sqrt{d}$ to $d$. Improving our lower bound requires new techniques and it is an interesting future direction. For learning homogeneous halfspaces (with RCN under Gaussian) there is no statistical-computational tradeoff. There is an efficient algorithm using the information-theoretically optimal sample size (within $\log$ factors). This also follows from our upper bound when $p=1/2$.  The conceptually interesting contribution of our work is that an information-computation gap appears for non-homogeneous halfspaces.
>
> 2. (**Question 2**) Regarding question 2 about a lower bound that is more sensitive with $\eta$: it is known that (even for computationally inefficient algorithms) there exists a sample complexity lower bound of order $\Omega(d/((1-2\eta)\epsilon))$ for RCN (see line 36).
>
> 3. (**Question 3**) Regarding question 3, we thank the reviewer for pointing this out. There is a typo in line 5 in the optimization subroutine, which we have fixed in the final version. The correct equation should contain $(1-2\eta)$ instead of $1-\eta$.

---

> > ### Comment · Reviewer_imug · 2023-08-16
> >
> > Thank you for addressing my questions and offering to edit the paper for clarity. I continue to believe that the paper is a useful contribution, and my score continues to reflect that.

---

### Official Review · Reviewer_qpWY · 2023-07-06

**Soundness:** 3 good
**Presentation:** 3 good
**Contribution:** 2 fair
**Rating:** 6
**Confidence:** 2

**Summary:**

This paper studies the problem of learning Gaussian half-spaces with random classification noise. Given labeled data $(x,y)\sim D$ where $x$ follows the standard Gaussian distribution,  a halfspace function $f$  such that $y=f(x)$ with probability $1-\eta$ and $y=-f(x)$ with probability $\eta$ and a precision parameter $\epsilon$. The goal is to construct a halfspace function $h$ such that for  $(x,y)\sim D$, $h(x)=y$ with probability at least $1-\eta-\epsilon$.

This paper provides lower and upper bounds on the sample/time complexity in the general case where $f$ can be biased: $p=\min(Pr[f(x)=1],  Pr[f(x)=-1])\neq 1/2 $. An efficient algorithm is proposed with a sample complexity $\tilde{O}(d/\epsilon + d/\max^2(p,\epsilon) )$. A lower bound on the sample complexity of $\Omega(\sqrt{d}/ \max^2(p,\epsilon) )$  is proved for SQ efficient algorithms.

**Strengths:**

Generalizing the problem of learning Gaussian halfspaces with RCN to not necessarily homogeneous halfspaces is a good contribution. Moreover, the bad term of the sample complexity $\tilde{O}( d/\max^2(p,\epsilon) )$ could not be improved in terms of the dependency in  $\max(p,\epsilon)$ thanks to the lower bound proved.

The algorithm is not a trivial generalization of the one for homogeneous halfspaces with RCN.

**Weaknesses:**

One weakness is that the lower bound has does not match the upper bound in terms of the dependency in $d$.

**Questions:**

Do you think your results could be generalized to (some) distributions $D$ with non Gaussian marginal $D_x$?

---

> ### Author Rebuttal · Authors · 2023-08-10
>
> We thank the reviewer for the time and effort. We address specific questions/comments by the reviewer below.
>
> 1. (**Weakness 1**) Gap between upper and lower bound (as a function of $d$): We refer the reviewer to bullet 4 in the response to reviewer m9mP for a detailed discussion. Our work showed an SQ lower bound of $\Omega(d^{1/2}/\mathrm{max}(p,\epsilon)^2)$ for the testing version of the halfspace learning problem; we also provided a matching upper bound (see Appendix C.6). We conjecture that our learning algorithm attains the optimal sample complexity as a function of $d$ as well (within the class of polynomial time algorithms). It remains an interesting question to develop a lower bound with the correct dependence on $d$ for the learning problem.
>
> 2. (**Question 1**) Regarding the reviewer’s question about an extension to non-Gaussian marginals: we believe that our algorithmic approach can be generalized to some log-concave distributions; this is left as an interesting direction for future work. We note that such an extension would require non-trivial extensions to our current analysis.

---

### Official Review · Reviewer_2pgi · 2023-07-07

**Soundness:** 4 excellent
**Presentation:** 4 excellent
**Contribution:** 3 good
**Rating:** 7
**Confidence:** 2

**Summary:**

The authors provide an algorithm to learn d-dimensional halfspaces under random classification noise upto error \eps. The algorithm has time complexity O(dN/\eps^2), where N is the number of samples; and sample complexity ~ \tilde{O}(d/\eps + d/\eps^2). They also prove a lower bound, in the statistical query model, that requires \Omega(\sqrt{d}/\eps^2) samples, unless the model makes high accuracy queries.

**Strengths:**

- The problem of learning half spaces under noise is an important and fundamental problem in machine learning.
- The results are tight in the statistical query model up to a factor \sqrt{d}.
- Clear and lucid technical overview..

**Weaknesses:**

No significant weaknesses.

**Questions:**

Some of the writing in the proofs can be simplified, constants like 0.000098 can be jarring.

**Limitations:**

Yes, adequately discussed.

---

> ### Author Rebuttal · Authors · 2023-08-10
>
> We thank the reviewer for the time and effort in providing feedback. We will make sure to polish the final version of the paper and simplify the statements where possible, as suggested.

---

### Official Review · Reviewer_QgYm · 2023-07-07

**Soundness:** 3 good
**Presentation:** 2 fair
**Contribution:** 3 good
**Rating:** 5
**Confidence:** 2

**Summary:**

This paper studies the PAC learning complexity of half spaces (or linear threshold functions), when the labels are flipped randomly with some probability $\eta$.

For realizable hypothesis classes, it is known that $O(d/\epsilon)$ samples are enough to learn a halfspace with $\epsilon$ 0-1 error. Under random classification error, the problem becomes much more challenging. This paper makes two contributions to this problem:
- First, it presents an efficient algorithm with comparable scaling to $\tilde O_{\eta}(d/\epsilon + d / (\max(p, \epsilon))^2 )$.
- Second, it proves statistical query lower bounds that match the above upper bound under certain regimes of $\eta$.

The upper bound involves three subroutines, including a warm start initialization that returns a vector close enough to the target. Second, it issues queries near the threshold, akin to learning a leaky ReLU loss. Lastly, it uses a simple hypothesis testing procedure, which draws a fresh sample and selects a hypothesis with the lowest test error.

Ther SQ lowe bound establishes the existence of a large set of distributions whose pariwise correlations are small. The construction builds on prior work of Diakonikolas, Kane and Steward (FOCS'17).

**Strengths:**

S1) The paper makes two contributions to learning halfspaces with random classification noise--this would be a nice contribution to this literature. The results seem solid and technically challenging.

S2) The authors make an effort to explain their results at a high level, which is helpful.

**Weaknesses:**

W1) My main concern with this paper is the presentation of their technical results; in particular, there is a large number of notations, many of which are not clearly explained or are difficult to follow. This would limit the potential audience of their work within the broader NeurIPS community.

W2) There are some inconsistencies between the statements, in particular, line 10 and line 46, which should be fixed or better explained.

**Questions:**

- Is the assumption of $p$-biasedness necessary for showing the sample complexity guarantee?

- The comparison with a concurrent work of Diakonikolas et al. (2023) needs to be stated more clearly.

- How critical are the results inherent to Gaussian inputs?

**Limitations:**

The authors have not discussed the limitations of their work in the main text.

The paper is of a highly technical nature, so potential negative societal impacts of their work would be limited.

---

> ### Author Rebuttal · Authors · 2023-08-10
>
> We thank the reviewer for their time and effort in providing feedback. Below, we provide a response to the comments and questions raised by the reviewer.
>
> 1. (**Weakness 1**) Regarding the reviewer’s comment: ‘main concern of this paper is the presentation of their technical result…’: We would like to argue that the definitions and relevant notation are already given clearly in the text and the algorithm. For example, the definition of $p$ (bias) and $t$ (threshold) are given in the introduction. In the main algorithm, $M$ is simply a parameter that denotes the total number of grid points we construct in $t$, and $\gamma_m$ denotes the width of the band on which we conditioned the gradients, as displayed in line 4 of the optimization subroutine. We believe these notations and parameters are necessary for a clear and succinct description of our algorithm.
>
> 2. (**Weakness 2**) Regarding the reviewer’s comment: `there are some inconsistencies between the statements, in particular, line 10 and line 46…’, we are afraid that the reviewer might have misunderstood our statements. In line 10, we presented our lower bound for learning general halfspaces under Gaussian with RCN, which is $\Omega(d^{1/2}/\mathrm{max}(p,\epsilon)^2)$; whereas in line 46 we provided the sample complexity of our algorithm, which is $\tilde{O}(d/\epsilon + d/\mathrm{max}(p,\epsilon)^2)$, i.e., an upper bound for this problem. There is a gap in the dependence of $d$ between our upper bound and lower bound, and we believe it is an interesting question to obtain a matching lower bound as a function of $d$ as well.
>
> 3. (**Question 1**) Regarding the reviewer’s first question (assumption on bias $p$): we emphasize that we do not make any assumptions on the bias $p$. The bias $p$ is an unknown parameter of the target halfspace (a number between $0$ to $1$). Our algorithm works for all possible values of $p$ and outputs a halfspace achieving error $\eta + \epsilon$. It turns out that the sample complexity of our algorithm depends on $p$. Importantly, our algorithm is adaptive in the sense that it does not need to know $p$ in advance. Finally, we reiterate that the sample complexity of our algorithm is near-optimal as a function of $p$ and $\epsilon$ within the class of efficient SQ algorithms.
>
> 4. (**Question 2**) Regarding the reviewer’s second question: A detailed comparison to the prior work of Diakonikolas et al. (2023) appears in Appendix A. The key points are that neither their algorithm nor their lower bound have any implications to our Gaussian setting. We will move this comparison to the main body in the revised version of the paper.
>
> 5. (**Question 3**) Regarding the reviewer’s third question (on Gaussian distribution): We start by pointing out that the Gaussian assumption makes our SQ lower bound stronger, as it holds even for the basic case that the feature vectors are Gaussian distributed. The analysis of our algorithm is currently specific to the Gaussian distribution, as this was the goal of our work (understanding sample-time tradeoffs for learning general halfspaces with RCN in the Gaussian setting). That said, we believe that our algorithmic approach can be modified to give near-optimal algorithms for more general marginal distributions (such as log-concave distributions). We leave this as an interesting extension for future work.

---

### Official Review · Reviewer_BT2g · 2023-07-26

**Soundness:** 4 excellent
**Presentation:** 3 good
**Contribution:** 4 excellent
**Rating:** 7
**Confidence:** 3

**Summary:**

This paper gives new positive and negative results for the problem of learning general (nonhomogenous) Gaussian halfspaces under the random classification noise (RCN) model. The motivating question is whether there exists a polynomial time algorithm nearly achieving the (known) minimal sample complexity required for solving the problem.

Positive result -- The authors give an efficient algorithm that solves this problem; this algorithm works with a sample complexity that is only off of the known optimal sample complexity by logarithmic factors. Specifically, the runtime of the algorithm is roughly $O(dN/\varepsilon^2)$, where $N$ (sample complexity) is roughly $O(d/\varepsilon^2 \cdot \log(1/\delta))$ and $\varepsilon$ is the target suboptimality. (note that I have omitted the dependences on the noise rate $\eta$ and the bias $p$ of the target function -- both are probably optimal)

To obtain the positive result, there are three main algorithmic contributions. The first is an initialization step, which returns a vector whose correlation with the true weight vector is pretty high. The second is an optimization procedure that takes as input a guess for the threshold of the target halfspace and returns a halfspace of low error with that fixed threshold. The third is a procedure that tests several hypotheses and selects the one with the lowest error.

Negative result -- The authors give a Statistical Query (SQ) model lower bound on the sample complexity of any algorithm solving the above problem -- either the statistical queries have to be very accurate, or there need to be an exponential number of queries made in order to recover a linear separator with nontrivial suboptimality. The authors also use a recent result almost equating SQ algorithms and low-degree tests to give a lower bound in the low-degree testing model. The exact statement is a bit technical, so I'll omit it here. It can easily be found in Theorem 1.5.

EDIT 2023-09-01 -- As mentioned in my response, my review stands. Thank you for answering the questions!

**Strengths:**

The paper almost (up to pesky logarithmic factors) resolves the motivating question of whether the optimal sample complexity can be achieved by an efficient algorithm for this problem. The algorithmic primitives are natural and easy to understand. The problem is pretty fundamental.

Finally, the paper is clearly presented.

**Weaknesses:**

There are minor typographic issues that can be cleaned up (e.g. spelling mistakes, long math strings, etc.) -- of course this is very minor.

Although the optimality of the sample complexity is discussed, there appears to not be as much emphasis on the optimality of the runtime (in particular, the worst case dependence on $\varepsilon$ in the runtime is $\varepsilon^{-4}$ -- is this unavoidable?)

**Questions:**

What do you think are interesting future directions? e.g. do you think it is possible to obtain an even faster algorithm for this problem? If so, do you think extensions of the ideas you presented are likely to work, or do you think a totally different algorithmic approach is necessary?

I probably missed this in the paper, but what is known about this problem when the input distribution is uniform over the vertices of the hypercube (i.e., $\mathsf{Unif}(\{\pm 1\})^d$? Do the techniques you give in this work readily transfer to this setting? It feels to me as though the Boolean setting is a more basic variant of the problem you study, but I would believe that the two are morally very similar.

**Limitations:**

Yes

---

> ### Author Rebuttal · Authors · 2023-08-09
>
> We thank the reviewer for the effort and the positive assessment.
>
> We address specific comments and questions by the reviewer below.
>
> 1. (**Weakness 2**) Regarding the reviewer’s comment ‘...not be as much emphasis on the optimality of the runtime…’: The focus of our work was to develop the first polynomial-time algorithm with near-optimal sample complexity (within the class of computationally efficient algorithms). We believe that a near-linear time algorithm  (i.e., an algorithm with runtime $\tilde{O}(Nd)$) exists, and it is an interesting direction for future work. The bottleneck to achieve this with our current algorithm is estimating the unknown threshold to the desired accuracy.
>
> 2. (**Question 1**) Future directions: A number of interesting future directions remain. An immediate open question is to strengthen the SQ lower bound to match our upper bound as a function of $d$ as well. We believe that our upper bound is tight up to log factors. Another direction concerns generalizing our algorithm to succeed for more general marginal distributions (e.g., isotropic log-concave) with the right sample complexity and understanding information-computation tradeoffs under such more general distributions.
>
> 3. (**Question 2**) Uniform Distribution on Hypercube: We start by pointing out that we do not think our methods can be modified to work for the uniform distribution over the hypercube. The main reason is that the anti-concentration property, which played a vital role in our analysis, is no longer possessed under the uniform distribution on the hypercube. Ergo we cannot compute the probability mass of an $\gamma_m$-width band (as we did in the Gaussian case), hence we are unable to utilize the gradient conditioned on the $\gamma_m$-width bands. To the best of our knowledge, the only known algorithm for learning halfspaces with RCN over the hypercube involves using a distribution-free (RCN tolerant) PAC learner. As a result, the sample and computational complexity is polynomial, but the degree of the polynomial is potentially not optimal.

---

> > ### Comment · Reviewer_BT2g · 2023-08-11
> >
> > Thank you for answering the questions! Seems like the hypercube case could be a nice problem...
> >
> > In any case, my assessment stands :)

---

### Official Review · Reviewer_m9mP · 2023-07-27

**Soundness:** 3 good
**Presentation:** 3 good
**Contribution:** 3 good
**Rating:** 6
**Confidence:** 3

**Summary:**

This paper studies the problem of learning non-homogeneous halfspaces in the presence of Random Classification Noise, where the marginal distribution is standard Gaussian in $d$ dimensions.

On the upper bound side, this work provides an efficient algorithm that achieves learning an $\epsilon$-optimal halfspace with sample complexity $\tilde{O} (\frac{d}{\epsilon} +  \frac{d}{\max (p,\epsilon)^2})$ where $p$ is the bias of the target halfspace. The algorithmic idea involves Initialization, Optimization, and final Testing subroutines. The algorithm runs for $O(1/\epsilon^2)$ guesses of the threshold $t$ and selects the best output halfspace.

On the lower bound side, this work establishes a nearly matching lower bound that no efficient SQ algorithm can learn a Gaussian halfspace with $\eta = 1/3$ with less than $\Omega( \frac{\sqrt d}{\max (p,\epsilon)^2} )$ samples.

**Strengths:**

1. This work studies learning halfspace with RCN, which is a fundamental problem in machine learning theory. Although it has been known that halfspaces are efficiently PAC learnable with RCN, this work refines the sample complexity of the previous works in the Gaussian marginal distribution setting. Moreover, it provides formal evidence that the quadratic dependence on $ \frac{1}{\max (p,\epsilon)}$ is the best one can hope for an efficient algorithm.

2. This paper is technically strong.

3. Along with the formal technical statements and proofs, this work exhibits some valuable informal explanations and implications of the results, which makes it easier for the readers to understand and appreciate the significance of the results in this paper.

4. The main algorithm consists of several subroutines, each coming with a nice explanation and theoretical performance guarantees.

**Weaknesses:**

1. It could be nice to have a more comprehensive review of the related works. E.g., how is the sample complexity in this work related to the other papers on learning halfspaces with RCN with comparable settings?

2. It could be good to include in the main algorithm the case where the threshold $t$ is significant.

**Questions:**

1. Could the main algorithm be modified a bit so that it could directly tell if the true threshold $t$ is significant enough and a constant hypothesis suffices?

2. In Algorithm 1, the Optimization procedure is invoked for $M = O(1/\epsilon^2)$ times, and $N_2 = \tilde{O} (\frac{d}{\epsilon} )$ samples are drawn in each iteration. Could you please explain a bit more about why the total sample complexity (the dependence on $\epsilon$) is less than $ \tilde{O} (\frac{1}{\epsilon^3} )$?

**Limitations:**

There are no obvious limitations to be addressed in my opinion.

One question that might be interesting to study is if it is possible to attain matching bounds in $d$.

---

> ### Author Rebuttal · Authors · 2023-08-09
>
> We thank the reviewer for the effort and the positive feedback. Below we respond to the reviewer’s questions in detail.
>
> 1. (**Weakness 1**) Regarding the reviewer’s comment on the related works, we have added more detailed comparisons with prior works in the related work section and Appendix A in the final version. Importantly, we remark that there is no other work that provides an efficient algorithm with the optimal sample complexity under Gaussian marginals for **general** halfspaces in the presence of RCN. Most prior works (e.g., [1], [2]) mainly aim to find computationally efficient algorithms for **homogeneous** halfspaces with RCN. In Appendix A, we also compared our work to a very recent paper [3], where the authors studied the setting of RCN corrupted halfspaces with margin assumptions.
>
> 2. (**Weakness 2 & Question 1**) Regarding the reviewer’s comment ‘could be good to include … the case where the threshold t is significant’ and the question about determining the case that a constant hypothesis suffices: In the realizable setting (or when $\eta=O(\epsilon)$), it is possible to distinguish between these cases by drawing ~$1/\epsilon$ samples and checking if there are different labels. In the case where $\eta=\Omega(1)$, a constant fraction of the samples will have positive and negative labels, therefore this naive approach will not work. In our case for determining whether a constant hypothesis suffices, we point out that this can be done by looking at the value of $p$, instead of $t$. Indeed, as mentioned in lines 204-205, when $p=\Theta(\epsilon)$, a constant hypothesis would suffice.
>
> 3. (**Question 2**) Regarding the reviewer’s question about why we only need $\tilde{O}(d/\epsilon)$ samples for the optimization subroutine rather than $\tilde{O}(d/\epsilon^3)$ samples, we would like to clarify that this is because we only draw one batch of samples before we start the optimization subroutine (line 4 in the main algorithm), rather than drawing fresh samples at each iteration. This is because our analysis leverages the uniform convergence of the empirical gradient estimation to the population gradient (under Gaussian), as manifested in Lemma B.6. The total sample complexity consists of the samples needed for the initialization subroutine, which is $\tilde{O}(d/\mathrm{max}(p,\epsilon)^2)$, plus the samples needed for the optimization subroutine, which in total is of the order $\tilde{O}(d/\epsilon + d/\mathrm{max}(p,\epsilon)^2)$.
>
> 4. (**Limitation**) Regarding the dependence on $d$: Our SQ lower bound applies to a natural testing (decision) version of our learning (search) problem (see Definition 3.1), which reduces to learning but is not necessarily equivalent to it. We show that any efficient SQ algorithm for this testing problem requires at least $\Omega(\sqrt{d}/\mathrm{max}(p,\epsilon)^2)$ sample complexity. Interestingly, we also show a matching upper bound for the testing problem (see Appendix C.6). It remains an interesting open question to develop a stronger lower bound technique that gives the correct dependence on $d$ for the learning problem. We conjecture that the sample complexity of our algorithm is optimal as a function of $d$ as well (within the class of all polynomial-time algorithms).
>
> References:
>
> [1] C. Zhang and Y. Li. Improved algorithms for efficient active learning halfspaces with massart and tsybakov noise. In Proceedings of The 34th Conference on Learning Theory, COLT, 2021.
>
> [2] C. Zhang, J. Shen, and P. Awasthi. Efficient active learning of sparse halfspaces with arbitrary bounded noise. In Advances in Neural Information Processing Systems, NeurIPS, 2020.
>
> [3] I. Diakonikolas, J. Diakonikolas, D. M. Kane, P. Wang, and N. Zarifis. Information-computation tradeoffs for learning margin halfspaces with random classification noise. CoRR, abs/2306.16352, 2023. Conference version in COLT’23.

---

> > ### Comment · Reviewer_m9mP · 2023-08-21
> >
> > Thank you for the responses.
> >
> > I am still a bit confused about Question 1, as $p$ is not required to be known ahead of time.
> >
> > My other questions are addressed. I will keep my rating.

---

### Author Rebuttal · Authors · 2023-08-09

We thank all the reviewers for their time and effort in reading and reviewing our paper. In particular, we are encouraged by the positive feedback and that our paper is appreciated by the reviewers in the following aspects: (i) **clear presentation and lucidity** (m9mp, BT2g, QgYm, 2pgi, Imug) (ii) **technical solidness** (m9mp, QgYm), (iii) **significance of solving a fundamental problem in machine learning** (m9mp, BT2g, 2pgi, qpWY).

---

### Decision · Program_Chairs · 2023-09-21

**Decision:**

Accept (poster)

**Comment:**

The paper nearly resolves the problem of learning general halfspaces under random classification noise with Gaussian marginals which is a fundamental problem in learning theory. The work shows a new statistical-computation gap for this problem under the SQ/low-degree testing framework by showing a computational lower bound of $\Omega(\sqrt{d}/\max(p, \epsilon)^2)$ compared to the known information theoretic bound of $\Theta(d/\epsilon)$. The authors further give a nearly matching (up to a $\sqrt{d}$ + log factors) upper bound.

All reviewers agree that this is a good mathematical contribution and adds to the long line of work on this problem and its variants. Therefore, I recommend accepting the paper. I encourage the authors to include the feedback from the reviewers as they prepare the camera-ready. In particular, it would help to clean up the typos/inconsistencies in theorem statements, improve the exposition of the main algorithm, and move the related work discussion to the main paper.